# Amyloid-β aggregates activate peripheral monocytes in mild cognitive impairment

Kristian Juul-Madsen [1,2], Peter Parbo [3], Rola Ismail[4], Peter L. Ovesen[2], Vanessa Schmidt[2], Lasse S. Madsen [5,6,7], Jacob Thyrsted[1], Sarah Gierl[1], Mihaela Breum[1], Agnete Larsen[1], Morten N. Andersen [1,6,8], Marina Romero-Ramos [1,9], Christian K. Holm [1], Gregers R. Andersen [10], Huaying Zhao [11], Peter Schuck [11], Jens V. Nygaard [12], Duncan S. Sutherland [13,14], Simon F. Eskildsen [6,7], Thomas E. Willnow[1,2], David J. Brooks[5,15,16] & Thomas Vorup-Jensen [1,9,13] ✉

The peripheral immune system is important in neurodegenerative diseases, both in protecting and inflaming the brain, but the underlying mechanisms remain elusive. Alzheimer's Disease is commonly preceded by a prodromal period. Here, we report the presence of large Aβ aggregates in plasma from patients with mild cognitive impairment ($n = 38$). The aggregates are associated with low level Alzheimer's Disease-like brain pathology as observed by [11]C-PiB PET and [18]F-FTP PET and lowered CD18-rich monocytes. We characterize complement receptor 4 as a strong binder of amyloids and show Aβ aggregates are preferentially phagocytosed and stimulate lysosomal activity through this receptor in stem cell-derived microglia. KIM127 integrin activation in monocytes promotes size selective phagocytosis of Aβ. Hydrodynamic calculations suggest Aβ aggregates associate with vessel walls of the cortical capillaries. In turn, we hypothesize aggregates may provide an adhesion substrate for recruiting CD18-rich monocytes into the cortex. Our results support a role for complement receptor 4 in regulating amyloid homeostasis.

Alzheimer's disease (AD) dementia usually follows after a long prodromal period, except for some rare familial forms[1,2]. This prodromal period offers opportunity for therapeutic intervention[3]. Significant progress in unraveling the clinical path to fulminant AD was made with the mild cognitive impairment (MCI) criteria[4]. MCI is characterized by a decline in amnestic abilities but lacking significant interference with the routines of daily life. Despite the innocuous symptoms, investigating the pathophysiology in MCI is essential because each year 10–20% of these patients develop dementia. Most MCI patients with AD-like brain pathology as shown by imaging techniques also develop dementia within 5 years[4]. In MCI patients, cortical amyloid-β (Aβ) plaques detectable with positron emission tomography (PET) imaging are associated with glial activation, which can be inflammatory[5] or may initially be protective to axons in early AD[6]. However, studies integrating

MCI patient's clinical characteristics with immune functioning at the cellular and molecular levels are still missing.

A hallmark of AD pathology is the formation of large protein deposits in the brain containing Aβ[42] polypeptide or hyperphosphorylated tau proteins. In their aggregated forms, both polypeptides are visualized by PET tracers as part of clinical procedures[7]. Soluble and aggregated forms of Aβ are also found in the cerebrospinal fluid (CSF) and in blood[8–10], but characterization in these samples has proven challenging. Progress was recently made by De et al. using super-resolution microscopy[11]. In surface-coated CSF samples, the Aβ aggregates from AD patients were structurally more elongated than in samples from MCI patients. In aqueous extracts from AD brains, Stern et al. also reported on large, soluble Aβ aggregates[12]. These findings encourage the hypothesis that more disease-relevant information on

neuropathological processes can be extracted from comparing size distributions of amyloid particles in patients.

A recent analysis of more than 100,000 individuals in the general population showed low monocyte counts in blood associated with an increased risk of developing AD[13]. In AD patients, the invasion of monocytes is a significant addition to the cerebral phagocytes[14–16]. Together with the epidemiological data, it highlights monocyte extravasation as a possible early step in this disease. It is well-accepted that the microvascular anatomy places constraints on the inflammatory responses in the highly vascularized cerebral cortex. Leukocyte extravasation occurs through the penetrating post capillary venules, which receive blood supply from a fine mesh of capillaries with luminal diameters of ∼6 µm[17,18]. Animal models of AD show intraluminal adhesion by monocytes to vascular deposits of Aβ, extravasation[19,20], and phagocytic clearance[21]. This is initiated by recognition of Aβ as a danger-associated molecular pattern (DAMP), which elicits critical signaling in myeloid cells[1]. Several receptors play a role in these processes, including the receptor for advanced glycation end products (RAGE), Toll-like 4, and triggering receptor expressed on myeloid cells 2 (TREM2)[1]. However, these receptors are only small molecules with no strong connection to the cytoskeleton precluding them as adhesion molecules halting leukocytes under shear forces in the blood vessels[22,23]. Intercellular adhesion molecule (ICAM)−1 has likewise been proposed to support monocyte migration across the blood-brain barrier through binding of lymphocyte function-associated antigen (LFA)−1, also known as CD11a/CD18 or integrin $\alpha_L\beta_2$ (*ITGAL/ITGB2*)[20]. However, the sole activation of this mechanism would not easily explain the relatively select depletion of monocytes in prodromal stages of AD[13] because leukocytes broadly use ICAM-1/LFA-1-mediated endothelial adhesion for extravasation[22]. It remains elusive which adhesion substrates and receptors would effectively enable primarily human monocytes to adhere to the Aβ-decorated cerebral blood vessels enough for consequences in MCI and AD.

Both peripheral monocytes and microglia express complement receptor (CR)3, also known as Mac-1, CD11b/CD18 or integrin $\alpha_M\beta_2$ (*ITGAM/ITGB2*), and CR4 (CD11c/CD18, integrin $\alpha_X\beta_2$, or *ITGAX/ITGB2*)[22]. They are highly expressed in maturated myeloid cells, with CR4 being the most abundant receptor in the murine dendritic cell membrane[24]. CR4 is co-expressed with CR3, and both receptors couple phagocytosis to the cytoskeleton, delivering sufficient mechanical strength for supporting cellular adhesion under blood flow and handling the uptake of large particles. Their function is highly regulated by conformational changes in the ectodomain from alterations in the cytoskeleton[24]. In humans, the CR4 binds polyanionic motifs through the alpha chain I-domain, which the CR3 does not[24,25]. Ablation of CR3-positive cells in a mouse AD model showed that peripheral monocytes play a crucial role in the brain clearance of amyloid plaques[26]. Genetic ablation of CR3 itself increased the formation of Aβ deposits[27,28]. Less is known about the role of CR4 in neural inflammation. For α-synuclein, another amyloidogenic protein of relevance in Parkinson's diseases, CR4 directly differentiates the amyloid from its monomeric form by binding ladders of anionic side chains, e.g., from glutamate, made by the characteristic stacking of parallel beta sheets[25,29]. These ladders exist widely in amyloids, including Aβ; however, it is unknown if CR4 recognizes other amyloids than α-synuclein and what role such recognition may play in the immunopathology of human neurological disease. CR4 is highly expressed in microglia, and the expression is further enhanced in the brains of AD patients[30]. CD18 integrins enable phagolysosomal activity via conformational changes in the integrin ectodomains[29]. However, the link between microglia scavenging of Aβ, phagolysosomal activity, and CD18 integrins is unclear.

In this work, with samples from a cohort of MCI patients, we identify a type of large amyloid Aβ aggregates in plasma with apparent hydrodynamic diameters ($D_H$) ranging from 600−900 nm. In patients, the concentration of aggregates correlates with depletion from the blood of CD18 integrin high-expressing monocyte subsets. The large

plasma Aβ aggregates characterize MCI patients with no detectable AD-like brain pathology, while a lack of aggregates is found in patients with increased AD-like brain pathology. As shown by hydrodynamic calculations, the behavior of these aggregates in a fluid prompts hard collision with the endothelial cell wall in the thin vessels of the brain. We hypothesize the aggregates are strong substrates for monocytic adherence through CR4 binding. This may enable monocytes to extravasate to clear amyloid aggregates in the highly vascularized cortex, explaining the connection between lack of brain pathology and large Aβ plasma aggregates observed in our study. Our findings bridge the activation of the peripheral immune system by Aβ aggregates and the regulation of amyloid AD-like brain pathology in MCI.

## Results

### Characterization of large Aβ aggregates in plasma from MCI patients versus AD mouse models

For the detection of large Aβ aggregates in human plasma, we used a recently developed methodology as single-particle reporter system based on nanoparticle tracking analysis (NTA) and quantum dots (QDs) conjugated with detecting antibodies (Ab) (Fig. 1a)[31]. An aducanumab biosimilar monoclonal Ab, specific for amyloid Aβ, was coupled to QDs using glycosylation sites in the Fc-region. We validated the fluorescence-detection mode (FDM)-NTA with regard to aggregate specificity and size determination. Specificity for in vitro-aggregated (A-)Aβ was probed by spiking healthy human control (HC) plasma (Fig. 1b−e and Supplementary Figs. 1 and 2). FDM-NTA robustly quantified A-Aβ aggregates with hydrodynamic diameters ($D_H$) of 600−900 nm (Fig. 1d) down to a total concentration of 1 ng/ml of spiked A-Aβ (Fig. 1e). Smaller aggregates, mainly with diameters from 50 to 200 nm, were more abundantly detected (Fig. 1c). No major background was found with isotype control Ab coupled to the QDs (Fig. 1b−e).

To understand how the FDM-NTA system reports sizes compared to other orthogonal techniques, we analyzed A-Aβ, both with sedimentation velocity analytical ultracentrifugation (SV-AUC) and scanning electron microscopy (SEM). In SV-AUC, samples were applied at total protein concentrations of 62.5, 250, and 1000 µg/ml (Fig. 1f, g and Supplementary Fig. 3). In a plot of the aggregate abundance as a function of sedimentation coefficient, a major peak showed at 20−30 S. There was a wide distribution of aggregates >100 S with no discrete peaks, indicating the presence of large particles with various sizes and shapes (Fig. 1g). The data showed a slight concentration dependence as the signal-weighted average sedimentation coefficient ($s_w$) of the 250-µg/ml sample is two-fold higher than $s_w$ of the 62.5-µg/ml sample. However, the total signal from A-Aβ stock was less than expected, suggesting that large particles sedimented to the cell bottom before the first scan. There was virtually no diffusion information in the data, due to the large size of aggregates and experimental timescale, which prevents an unambiguous calculation of $D_H$s from the data set. To obtain such information, we matched the distinct peak at ∼25 S in the sedimentation coefficient profile (Fig. 1g) with reported findings on large, highly anisotropic and polydisperse oligomers of mannan-binding lectin (MBL)[31,32]. The A-Aβ peak at ∼25 S (Fig. 1g) matched a 25-S MBL oligomer[31]. The diameter of the MBL oligomer was calculated to ∼108 nm from small-angle X-ray scattering data[32]. This corresponded well to a ∼120-nm peak in the distribution for A-Aβ when aggregates were analyzed without QDs using NTA in light scattering mode (Supplementary Fig. 4). In the data from spiked plasma analyzed by FDM-NTA, we also able to clearly discern this peak at ∼125 nm (Fig. 1c). Taken together, these orthogonal approaches confirm the NTA methodology agrees with the SV-AUC analysis for large complexes.

As a further validation, we also used SEM. In the micrographs, it was likewise easy to appreciate the presence of A-Aβ aggregates with a long axis of 600 nm or more (Fig. 1h and Supplementary Fig. 5). The ability of these aggregates to capture multiple QD-aducanumab

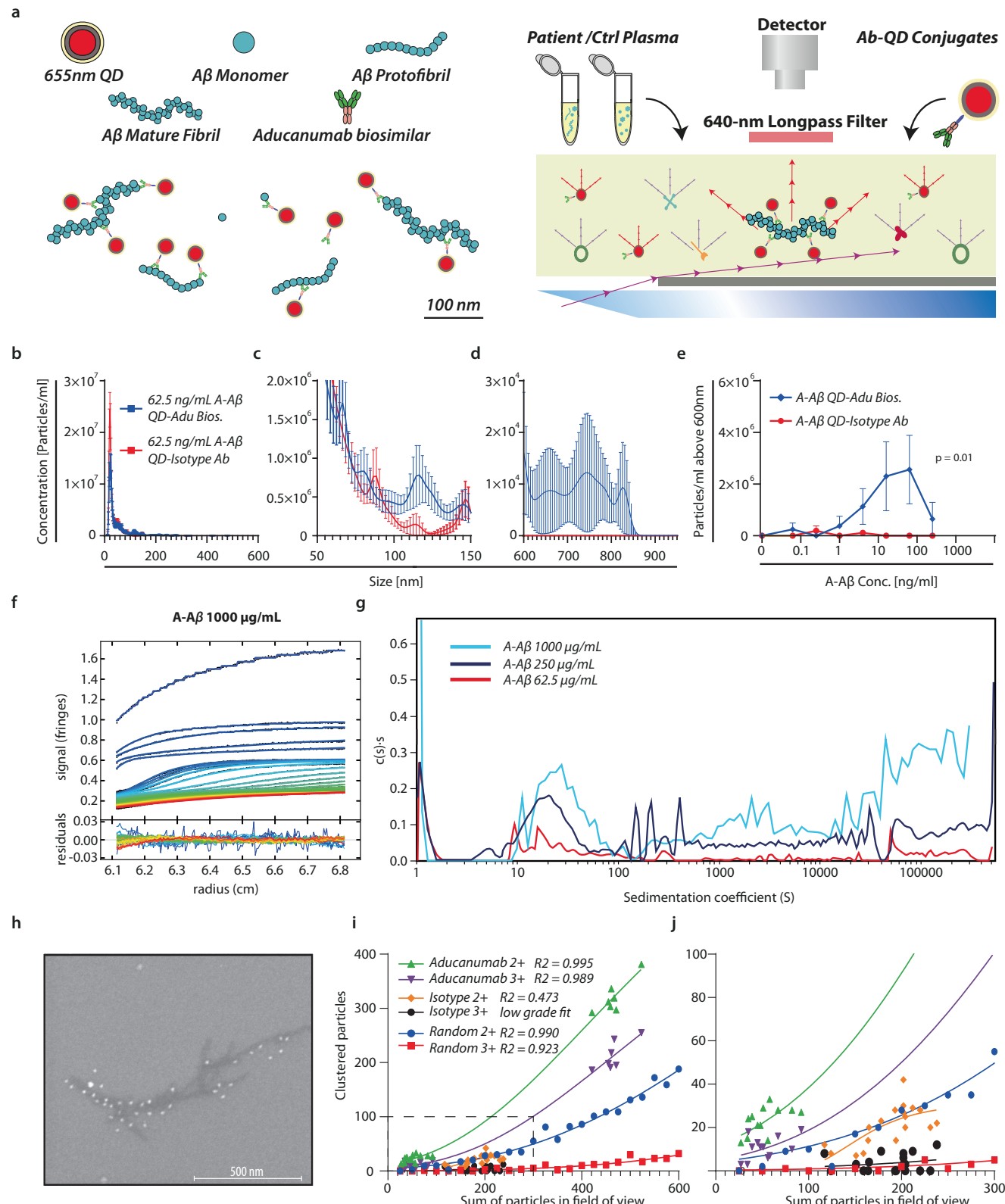

conjugates undoubtedly aided the sensitivity of the FDM-NTA for these particles (Fig. 1d, e). As for the FDM-NTA, the specificity of the QD-aducanumab conjugates was confirmed: multiple QD-aducanumab conjugates clustered with Aβ aggregates more frequently than seen by in silico-made random clustering. QDs coupled with isotype control Ab only showed clustering similar to that of the randomly generated data (Fig. 1i, j and Supplementary Fig. 5).

Samples were analyzed from a cohort of 38 MCI patients and 17 aged-matched HC by FDM-NTA (Fig. 2a–c). Below an aggregate diameter of 600 nm, no differences were found between MCI and HCs, although aggregates with diameters between 50–150 nm dominated the samples of both MCI and HC plasma (Fig. 2a, b) as also reported by others[11]. A subgroup of MCI patients displayed a low concentration of Aβ aggregates above 600 nm (Fig. 2c), and the concentration of

**Fig. 1 | Detection of Aβ aggregates using fluorescence detection nanoparticle tracking analysis, sedimentation velocity analytical ultracentrifugation, and scanning electron microscopy. a** QD-based reporter system[31] for detecting aggregated Aβ in complex bioliquid using 20-nm QDs with covalent coupling to aducanumab biosimilar (Adu Bios.) or isotypic control Ab. **b–d** Adu Bios. QD-conjugate specificity towards A-Aβ vs. isotype control Ab QD-conjugate (2 biologically independent experiments with 5 technical replicates; mean ± SEM). FDM-NTA size profiles for 1:10 HC plasma spiked with 62.5 ng/ml A-Aβ in the interval 0–600 (**b**) nm, 50–150 nm (**c**) and 600–900 nm (**d**). In (**e**), the sum of aggregates >600 in plasma spiked with 0–250 ng/mL of A-Aβ is shown as a function of the A-Aβ concentration (2 biologically independent experiments with 5 technical replicates were analyzed using a two-way ANOVA mixed effects model with p-value indicating column effect; mean ± SEM). **f, g** Rayleigh interference optical sedimentation

boundaries of A-Aβ. Best-fit from the c(s) model at 1000 μg/mL, sedimenting at 3000, 10,000, and 40,000 rpm (-700 × $g$, -7800 × $g$ and -125,000 × $g$, respectively). Residuals are shown in the lower panel. Each scan is shown in a color temperature indicating the evolution of time (**f**). SV-AUC of Aβ with the normalized abundance of aggregates, c(s), plotted as a function of the sedimentation coefficient (**g**). **h** ~700-nm Aβ-fibril binding multiple Ab-QD conjugates visualized by SEM. **i, j** Cluster analysis of experimentally generated surfaces and simulated surfaces with random positioning of particles in an equal size field of view. Data were fitted to a Gompertz growth model using Graphpad Prism for clustering of two or more particles (2+) and three or more particles (3+). Data represent three biologically independent experiments with 6 images from each sample. Source data are provided as a Source Data file.

aggregates was significantly different even between the combined group of patients and the HC group (Fig. 2d). Within the MCI group, the aggregates could not be explained by differences in *APOE* genotype (Supplementary Fig. 6).

We wanted to know how the presence of the large Aβ plasma aggregates connects with AD-like brain pathology and tested 52-week-old APP(V717F) transgenic mice expressing a human gene variant (V717F) causative of familial AD[33]. Marked manifestation of AD-like pathology in these mice was confirmed by quantification of soluble and insoluble Aβ$_{40}$ and Aβ$_{42}$ species in the cortex and hippocampus with a significant reduction in the ratio of soluble Aβ$_{42}$ to Aβ$_{40}$ in both regions and corresponding an increased proportion of the insoluble peptide variant (Fig. 2e, d). Plasma samples from 11 mice were analyzed with the QD-aducanumab reporter system used for the human samples. Despite the pronounced amyloid peptide burden seen in the brain of these mice, no Aβ aggregates above 600 nm were found in the plasma (Fig. 2g, h).

### Large >600 nm Aβ plasma aggregates associate with monocyte maturation and integrin activation in MCI patients

To determine the effect of these Aβ plasma aggregates on the peripheral immune system, we divided the MCI patient cohort into two groups, namely large >600 nm Aβ aggregate-positive (Agg [+]) (*n* = 10) and large >600 nm Aβ aggregate-negative (Agg [−]) (*n* = 28) patients. Peripheral blood mononuclear cells (PBMCs) from both MCI groups and HC were analyzed using flow cytometry. Monocytes were gated into classical (CD16-CD14++), intermediate (CD16+CD14++), non-classical (CD16++CD14+), and unclassified CD16-CD14- monocytes, the latter associated with the dendritic cells[34] (Supplementary Fig. 7). Both MCI groups displayed a higher fraction of total monocytes among all PBMCs than the HC group (Fig. 2i). In the large Agg [-] group, this increase was primarily in the immature classical monocyte subset, while unchanged in the large Agg [+] group (Fig. 2j). No significant differences were observed in the intermediate subset and the changes in the large Agg [+] group were driven by the expansion of maturated non-classical monocytes. This subset was unchanged in the large Agg [−] group compared to control (Fig. 2k, l). Both MCI groups had decreased ratios of the unclassified subset (Fig. 2m). Non-classical monocytes are particularly active in complement-mediated phagocytosis and transendothelial migration[15,34], functions regulated by the CD18 integrins LFA-1, CR3, and CR4[24]. We determined CD18 expression in the monocyte subsets. The concentration of Aβ plasma aggregates in the Agg [+] patients displayed a negative correlation with CD18 expression (Fig. 2n, o). The ligand binding activity of CD18 integrins is strongly regulated by the receptor ectodomain conformation[24]. We assessed the conformational change of CD18 integrins in large Agg [+] monocytes using the Ab KIM127, which binds an epitope exposed in activated CD18[24]. A positive correlation was found between the activation of CD18 integrins on the surface of unclassified monocytes and the concentration of Aβ plasma aggregates in large Agg [+] patients (Fig. 2p).

### MCI patients with large >600 nm Aβ plasma aggregates have less AD-like brain pathology

Previous studies have shown that brain inflammation accompanies amyloid formation in MCI[5]; however, a mechanistic explanation is missing for how inflammation develops and how the peripheral immune system is involved. To evaluate the link between Aβ plasma aggregates and AD-like brain pathology, we compared PET scans of MCI patients and HCs at baseline (where plasma samples were drawn) and two years after (Fig. 3). Presence of large >600 nm Aβ plasma aggregate was used to stratify the patients. Areas with AD-related pathology were quantified with the [18]F-Flourtaucipir ([18]F-FTP) PET tracer, which binds to neurofibrillary tau tangles, and the [11]C-Pittsburgh Compound-B ([11]C-PiB) PET tracer, which binds to amyloid Aβ. The large >600 nm Agg [−] MCI patients had an apparent increase in the retention of both [18]F-FTP and [11]C-PiB, in agreement with the observations from the PDAPP mouse model (Fig. 2g, h), and suggests the absence of large Aβ plasma aggregate correlate with an increased risk of developing AD. In comparison, the large >600 nm Agg [+] MCI patients had only minor changes in the retention of these tracers at a level similar to that of the HC group (Fig. 3a, b). We analyzed the distribution of tracer retention in the large Agg [+] group at a two-year follow-up. It showed significantly lower [18]F-FTP tracer retention in the temporal lobe and precuneus compared to the large >600 nm Agg [−] MCI group, even when compared to the HC group (Fig. 3c). Additionally, significantly lower [11]C-PiB retention was found across the entire cortex of the brain for large >600 nm Agg [+] MCI patients compared to the large >600 nm Agg [−] MCI group (Fig. 3d).

### Large, but not small, Aβ plasma aggregates form impact zones in thin vessels

With the observation that the presence of large Aβ plasma aggregates influence subsets of monocyte often involved in extravasation, a question pertains to if these aggregates are presented on relevant parts of the endothelium in blood vessels of the brain. To understand the likely behavior of Aβ in the bloodstream, we performed in silico simulations of the aggregate hydrodynamics under conditions mimicking the environment of brain capillaries. Compared to experimental investigations using engineered particles with $D_{HS}$ - 140 nm[18], our calculation permitted investigations on the roles of vessel diameter and particle size in vascular distribution.

Neutrally buoyant particles in a flow channel converge to specific locations in the channel cross-section under laminar flow. For a cylindrical pipe, the equilibrium position is about 0.6 times the pipe radius. This is known as the Segrè-Silberberg effect forming a ring of particles[35]. Initially, we considered channels with a diameter of 4 or 8 μm, low blood viscosity at 1.2 cP, normal density at 0.994 g/ml, and a high blood flow (velocity) of 3.2 mm/s for dispersion of particles with a density of 1.988 g/ml, twice the density of blood, and diameters of 30–900 nm (Fig. 4a). Wall-dependent lift and drag forces are exerted on the particles as they are carried along the channel by a parabolic fluid velocity profile. As larger particles, with diameters 675–900 nm,

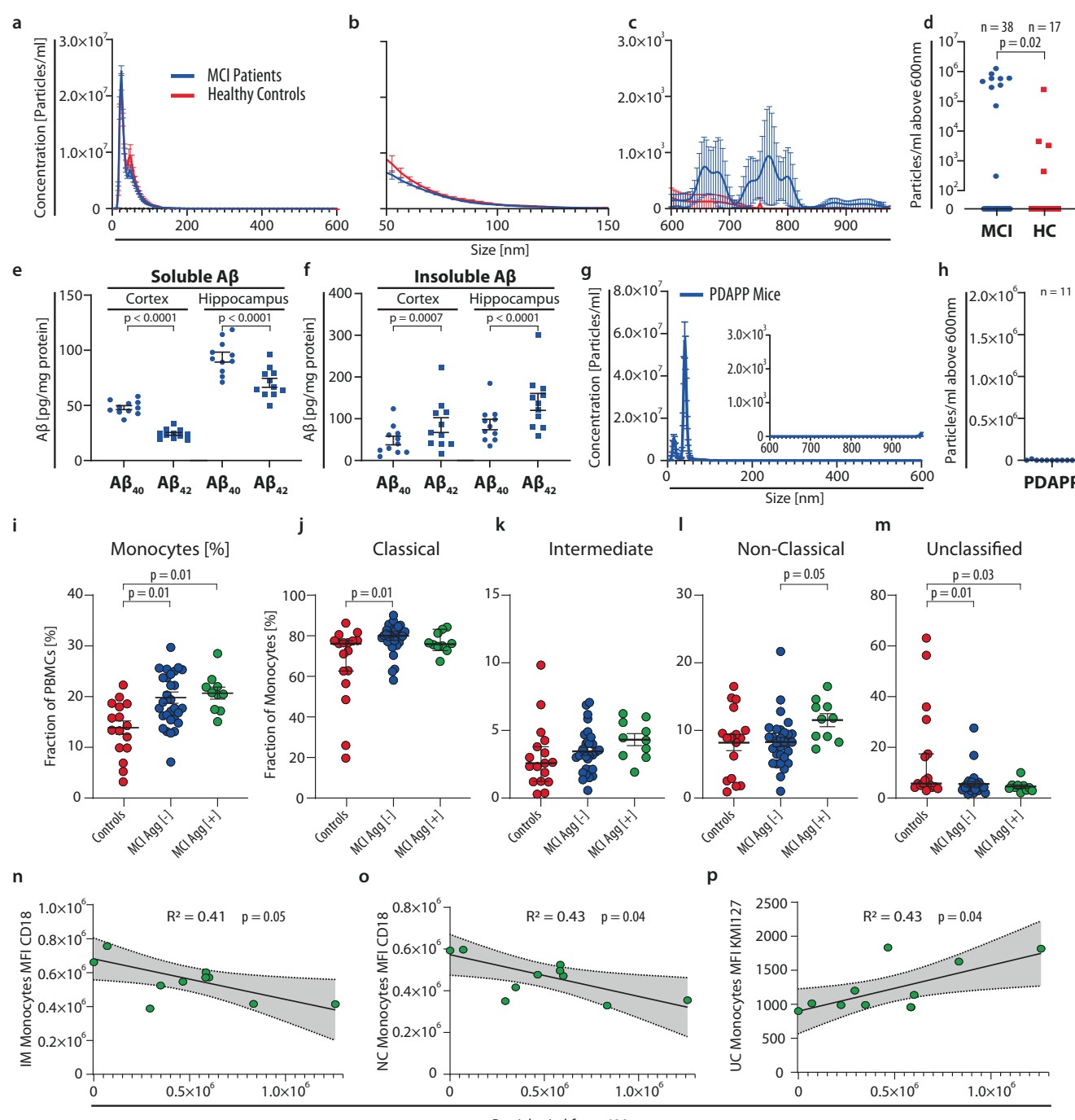

**Fig. 2 | Aβ aggregates in plasma larger than 600 nm from healthy controls (HC), mild cognitive impairment (MCI) patients, or aged PDAPP mice.** NTA size distribution from HC and MCI plasma incubated with Adu bios (MCI; $n = 38$, HC; $n = 17$; mean ± SEM). QDs in the size range 0–600 nm (**a**), 50–150 nm (**b**), and 600–975 nm (**c**). **d** Quantification of Aβ aggregates >600 nm from HC and MCI. Significance was tested in an unpaired two-sided *t*-test with Welch correction for unequal variances (MCI; $n = 38$, HC; $n = 17$; mean ± SEM). **e, f** Quantification of soluble and insoluble Aβ40 and Aβ42 from the cortex and hippocampus of PDAPP mice. Significance was tested using a two-sided paired *t*-test ($n = 11$; 5 male, 6 female; mean ± SEM). **g** NTA size distribution from PDAPP mice and magnification of 600–975 nm ($n = 11$; 5 male, 6 female; mean ± SEM). **h** Quantification of Aβ aggregates >600 nm from PDAPP mice ($n = 11$; 5 male, 6 female; mean ± SEM). **i–p** Flow cytometric analysis of monocyte subsets from MCI patient and HC plasma. Percentage of all monocytes

(**i**) classical monocytes (CD14++ CD16−) (**j**), intermediate monocytes (CD14++ CD16+) (**k**), non-classical monocytes (CD14+ CD16++) (**l**), and unclassified monocytes (CD14− CD16−) (**m**) of total monocytes. The MCI cohort were stratified into groups of >600 nm aggregate-positive (Agg [+]) and aggregate-negative (Agg [−]) patients. Statistical analyses in I-M were made with Kruskal–Wallis and Dunn's multiple comparisons test (Controls; $n = 17$, MCI Agg [−]; $n = 28$, MCI Agg [+]; $n = 10$; mean ± SEM). (**n, o**) Correlation between the concentration of aggregates Agg [+] MCI patients and membrane expression of CD18 (MFI) in intermediate (**n**) and non-classical monocytes (**o**). **p** Correlation between unclassified monocyte expression of activated (KIM127+) CD18 and concentration of aggregates in Agg [+] MCI patients. In (**n–p**), Spearman's correlation coefficient ($R^2$) was calculated together linear regression and a two-sided test of the slope to be non-zero ($n = 10$; error bars indicate SE). Source data are provided as a Source Data file.

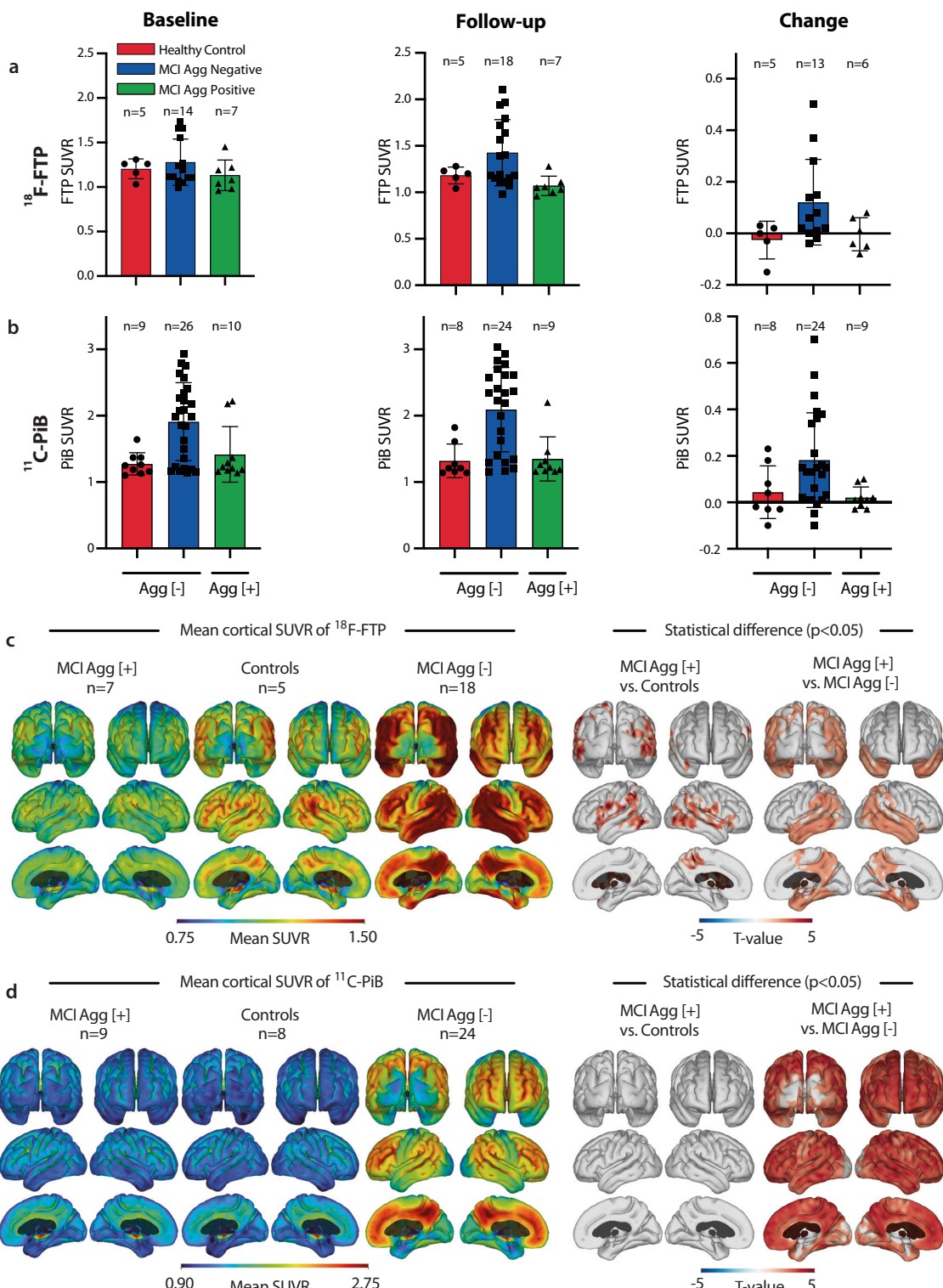

**Fig. 3 | Large Aβ Agg [+] MCI patients display lower AD-like brain pathology than large Aβ Agg [-] MCI. a** [18]F-FTP PET uptake in each group at baseline, two-year follow-up, and longitudinal change. The mean was calculated in a ROI comprising the entorhinal, amygdala, parahippocampal, fusiform, inferior temporal, and middle temporal cortical regions; mean ± SD. **b** Mean [11]C-PiB PET uptake in each group at baseline, two-year follow-up, and longitudinal change. The mean uptake was calculated in a ROI comprising the prefrontal, orbitofrontal, anterior and posterior cingulate, precuneus, parietal, and temporal cortical regions; mean ± SD. **c** Mean cortical [18]F-FTP PET uptake in each group at two-year follow-up (left panel)

and statistical results from a two-sided unpaired *t*-test between large Aβ Agg [+] MCI patients vs. controls and large Aβ Agg [-] MCI patients, respectively (right panel). **d** Mean cortical [11]C-PiB PET uptake in each group at two-year follow-up (left panel) and statistical results from a two-sided unpaired *t*-test between large Aβ Agg [+] MCI patients vs. controls and large Aβ Agg [−] MCI patients, respectively (right panel). Positive *t*-values (red) indicate significantly lower uptake in the large Aβ Agg [+] MCI group. Statistical cortical maps were familywise error rate corrected (α = 0.05) using cluster-extent-based thresholding with a primary cluster-defining threshold of *p* < 0.05. Source data are provided as a Source Data file.

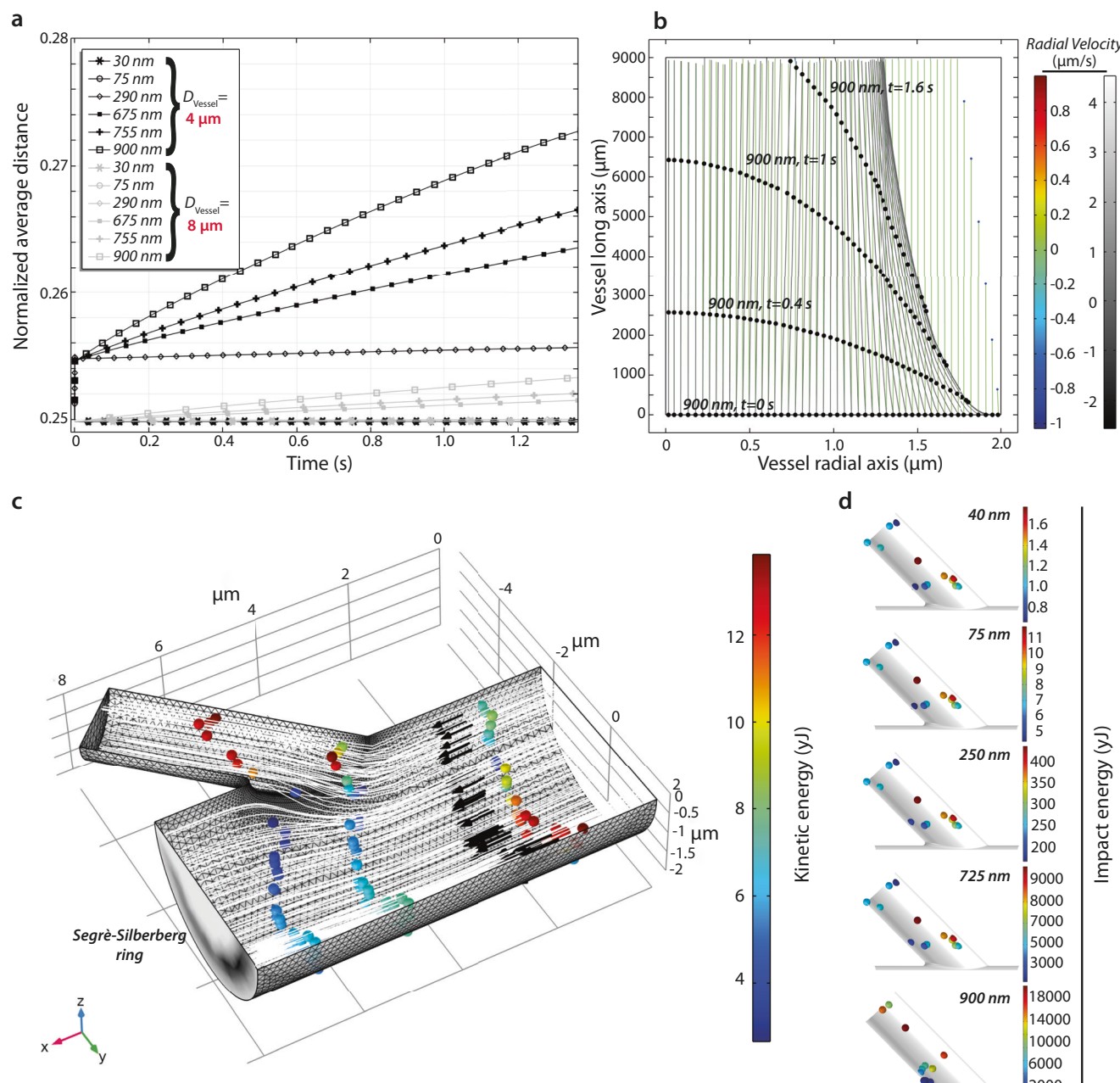

**Fig. 4 | Hydrodynamic focusing and endothelial impact of blood-born particles.** **a** Simulation of particle behavior in straight vessels with diameters of 4 (black) and 8 (gray) μm. Initially (t = 0 s), all particles are evenly distributed. The plot shows the normalized average distance in the radial direction of all particles as a function of time. According to the applied model, particles redistribute giving rise to inertial focusing. This mechanism is seen as a shift in average distance primarily in smaller vessels and for larger particles. **b** Trajectories of 900 (black dots) and 75 (blue dots) nm sized particles in a 4-μm in diameter vessel 9 mm long. The small, 75-nm particles shown to the right realize no inertial focusing. The larger, 900-nm particles (black dots) are depicted at time equal to 0, 0.4, 1, and 1.6 s. Immediate inertial focusing of these larger particles, following the trajectory of gray lines, is realized after a few ms as seen in the lower right corner of the plot. The bars in color and greyscale show the particle velocity in the radial direction. The larger particles move inwards at the lower right corner with a velocity of −2 μm/s. **c**, **d** Particle behavior in a branched 3D vessel. **c** Trajectories of 75-nm sized particles at 0.3, 0.4, 1.0, and 1.4 ms after being release at the inlet. The arrows indicate the flow direction of 50 out of 20.000 particles being modelled. The color indicates the kinetic energy of particles in units of yocto (10⁻²⁴×) Joules. The grayscale shown on the outlets plot the spatial distribution of inertial focused particles. **d** Particle impact on the endothelial surface detected by the model with calculation of the kinetic energy. Sizes from 40 to 900 nm were investigated and variations in impact pattern and energy levels are clearly demonstrated. Larger particles showed more localized impact zones and a 10,000-fold increase in endothelium impact energy.

were carried through the channel, inertial forces caused them to reach equilibrium positions at distances from the center of $0.3\,D$, where $D$ is the diameter of the channel (Fig. 4a, b). These equilibrium positions are consistent with the Segrè–Silberberg effect. Smaller particles did not significantly alter position, while the thinner channel diameter of 4 μm enhanced the inertial focusing compared to 8 μm.

Next, we asked how inertial focusing may affect contact of particles with the blood vessel wall when the morphology of the vessel alters. Boundary conditions were sampled from previous work on the carotid arteries[36–38], but maintaining other variables as above. In Fig. 4c, trajectories for 50 particles in a bifurcating vessel are plotted to illustrate their principal downstream behavior in the direction of the

black arrows. Their cross sectional distribution from inertial focusing can be seen at the outlet (black-white). The color of the particles indicates their velocity, with those particles closer to the wall (blue) traveling slower. Based on conditions noted under stenosis[39], a small pressure difference of 40 Pa was applied between the smaller (1 μm) and larger outlet (4 μm) outlet. This was sufficient to increase the average particle velocity in the smaller channel and cause wall impacts. The flow behavior of 20,000 particles of sizes of 40 to 900 nm in diameter were simulated. Impact zones were plotted after 20,000 particles passed through (Fig. 4d). Smaller particles affected a slightly larger area. For these boundary conditions, the most affected zones were primarily on the particle starboard side. Still, also a less impacted site was seen on the opposing surface. Calculation of collision energy found even more marked differences between small and large particles. On average, the large 900-nm particles had a 10,000-times higher impact energy than the smaller 40-nm particles. Locations of impact sites were similar for all particles (Fig. 4e).

### Aggregated Aβ is a strong adhesive substrate for CR4

The Aβ$_{42}$ fibrils found in the brain of AD patients[40] form highly stable beta-sheet structures that stack Asp-1, Glu-3, and Asp-7 in ladders (Fig. 5a). Such polyanionic stretches resemble the pattern recognized by CR4 in α-synuclein[29]. Open-conformation ligand-binding active CR4 binds better Glu than Asp side chains[24]. In the Aβ$_{42}$ fibrils, the Glu-3 ladder is accessible for contact with the positively charged top-face of the CR4 ligand recognition I domain (Fig. 5b). A ladder stretch filling the CR4 I-domain requires approximately five stacked Aβ$_{42}$ polypeptides, producing a low expected stoichiometry of I-domain per polypeptide. Further, the twisted fibril structure limits receptors engaging multiple binding sites. The distance is 43 nm between two binding sites facing the receptor-presenting surface with the same orientation (Fig. 5c). In this model, only long fibrillar aggregates would enable multiple CR4 molecules to sustain a high-avidity interaction, which plays a role in strong adhesion of integrins to substrates[41].

To determine the binding strength between amyloid Aβ and intact CR4, we applied a cell adhesion assay with parental K562 cells expressing no CD18 integrins or engineered to express CR3 or CR4 (Fig. 5d–f). Centrifugal force mimicked the shear stress in blood vessels[42], thereby simulating conditions of adhesion to vascular deposits of Aβ. When Mn$^{2+}$ ions were added to activate integrins[24], K562 cells expressing CR4 adhered more strongly than parental cells to surfaces coated with A-Aβ (Fig. 5f). Dissolution of the aggregates by prior treatment of the surfaces with guanidinium hydrochloride (Gu·HCl) reduced adhesion, supporting CR4 recognition preferentially of large amyloid aggregates[29] (Fig. 5c, f). K562 cells expressing CR3 showed the same level of adhesion as the parental K562 cells (Fig. 5d, e).

The CD18 integrin-activating Ab, KIM127[24], increased monocyte cell adhesion (Fig. 5g). CD18 is the beta-chain of both CR3 and CR4, but only inhibitory Ab to CR4 (3.9) reduced the adhesion significantly. At the same time, function-blocking Ab to CR3 (ICRF44) or an isotype control IgG1 did not (Fig. 5h), showing CR4 is a main factor behind A-Aβ recognition in monocytes.

At the biochemical level, we used surface plasmon resonance (SPR) and 2D analysis of binding kinetics[43,44] (Fig. 5i–l). We found no interaction between CR4 and synthetic, predominantly monomeric (M-)Aβ, even at high concentrations of the CR4 I-domain (Fig. 5k). In contrast, CR4 had multiple types of binding sites for A-Aβ, either with a $K_D$ ~ 1.6 × 10$^{-11}$ M and a dissociation rate $k_{off}$ ~ 2.5 × 10$^{-4}$ s$^{-1}$ or with a more moderate affinity, with $K_D$ ~ 1.0 × 10$^{-6}$ M and $k_{off}$ ~ 5.0 × 10$^{-5}$ s$^{-1}$ (Fig. 5l). CR4 binding to A-Aβ was at least as strong as to complement iC3b, a well-characterized ligand for CR4 (Supplementary Fig. 8)[24,45]. CR3 showed weak binding to Aβ monomer, with a $K_D$ ~ 7.9 × 10$^{-5}$ M and a $k_{off}$ at 1.1 × 10$^{-1}$ s$^{-1}$ (Fig. 5i). Aβ aggregation only moderately strengthened the binding by CR3 to a $K_D$ ~ 3.2 × 10$^{-6}$ M and a $k_{off}$ ~

1.6 × 10$^{-2}$ s$^{-1}$ (Fig. 5j). As directly observable from the sensorgram and the calculated total binding capacity of the surfaces (∬SdkdK), the CR4 bound with much lower stoichiometry to A-Aβ (Fig. 5J) than CR3 (Fig. 5l).

### CD18 integrins support phagocytic activity and depletion of large, but not small, Aβ aggregates in human monocytes

Myeloid cells are important scavengers of decayed macromolecules[21]. The contribution of CD18 integrins in Aβ scavenging was studied with blood-derived, primary monocytes incubated with fluorescent-labeled M- or A-Aβ. Again, CD18 integrins were activated with KIM127 Ab or, as a control, incubated without. Function-blocking antibody to CR4 (Ab 3.9) was also included (Fig. 6a and Supplementary Fig. 9). KIM127-activated CD18 integrin enabled efficient phagocytosis of A-Aβ, but not M-Aβ, while control IgG had no impact on the uptake. The CR4-blocking Ab 3.9 abolished the phagocytosis (Fig. 6b). Together, these data suggested monocytes use CD18 integrins, especially CR4, for clearance of A-Aβ agreeing well with the ligand binding preferences determined earlier by cell adhesion and SPR experiments (Fig. 5).

We also analyzed the concentration of remaining particles in the supernatant under conditions with and without CD18 integrin activation (Fig. 6c–h). To this end, M- or A-Aβ was labeled with QDs and incubated with monocytes under conditions permitting phagocytosis (Fig. 6c) as shown earlier for α-synuclein[29]. In terms of size-selective clearance, the supernatants were analyzed in FDM-NTA. The supernatants were dominated by QDs with a $D_H$ ~30 nm), indicating a limited protein coupling for these particles (Fig. 6d). When the normalized particle distribution was plotted as cumulative distributions, the critical size for cellular QD-Aβ depletion emerged more clearly. Interestingly, in agreement with the general preference for binding large A-Aβ (Fig. 5f, l) and the predicted structural requirements for fibrillar aggregates for avid CR4 binding (Fig. 5c), the difference in depletion mainly seemed to involve particles with diameters ~60–200 nm (Fig. 6e, f). The same result could also be derived from analyzing the remaining concentration of particles in the supernatant when a size-cut at $D_H$ > 50 nm to exclude particles with a limited protein coupling from the analysis (Fig. 6g, h). More of these particles were depleted from the supernatant when conjugated with A-Aβ than if they carried M-Aβ on the conditions of KIM127 integrin activation (Fig. 6h), implicating CD18 integrins in the process.

### Aβ aggregates are preferentially phagocytosed and stimulate lysosomal activity through CR4 in stem cell-derived microglia

The function of CR4 in microglial cells was not extensively studied in the past. We wanted to shed light on its role in phagocytosis and the formation of phagolysosomes when exposed to Aβ. To address this question, induced pluripotent stem cells were differentiated into induced microglial cells (iMGs) (Fig. 7a and Supplementary Fig. 10). The cells were incubated with fluorophore-labeled, M- or A-Aβ$_{42}$ and KIM127 or control Ab (Fig. 7b and Supplementary Fig. 11). Similar to the monocytes, iMGs preferentially phagocytosed A-Aβ in a process enhanced by KIM127 Ab activation of CD18 integrin (Fig. 7c). These findings were even more evident for the lysosomal activity. A-Aβ strongly induced lysosomal activity, but M-Aβ showed no such ability, which was not altered by adding KIM127 Ab. The presence of both A-Aβ and KIM127 stimulated a significantly higher activity than KIM127 Ab or A-Aβ alone (Fig. 7d).

For narrowing down the CD18 integrins involved, the membrane expression of the CR3 (CD11b) and CR4 (CD11c) alpha chains was quantified in the iMGs under stimulations with ligands and KIM127 integrin activating Ab (Fig. 7e, f). For the CD11c, the presence of aggregated rather than M-Aβ significantly enhanced the expression. It was significantly increased if KIM127 Ab was added rather than isotype control Ab. For the M-Aβ, KIM127 Ab did not influence the CD11c expression. By striking contrast, the CD11b expression was not affected

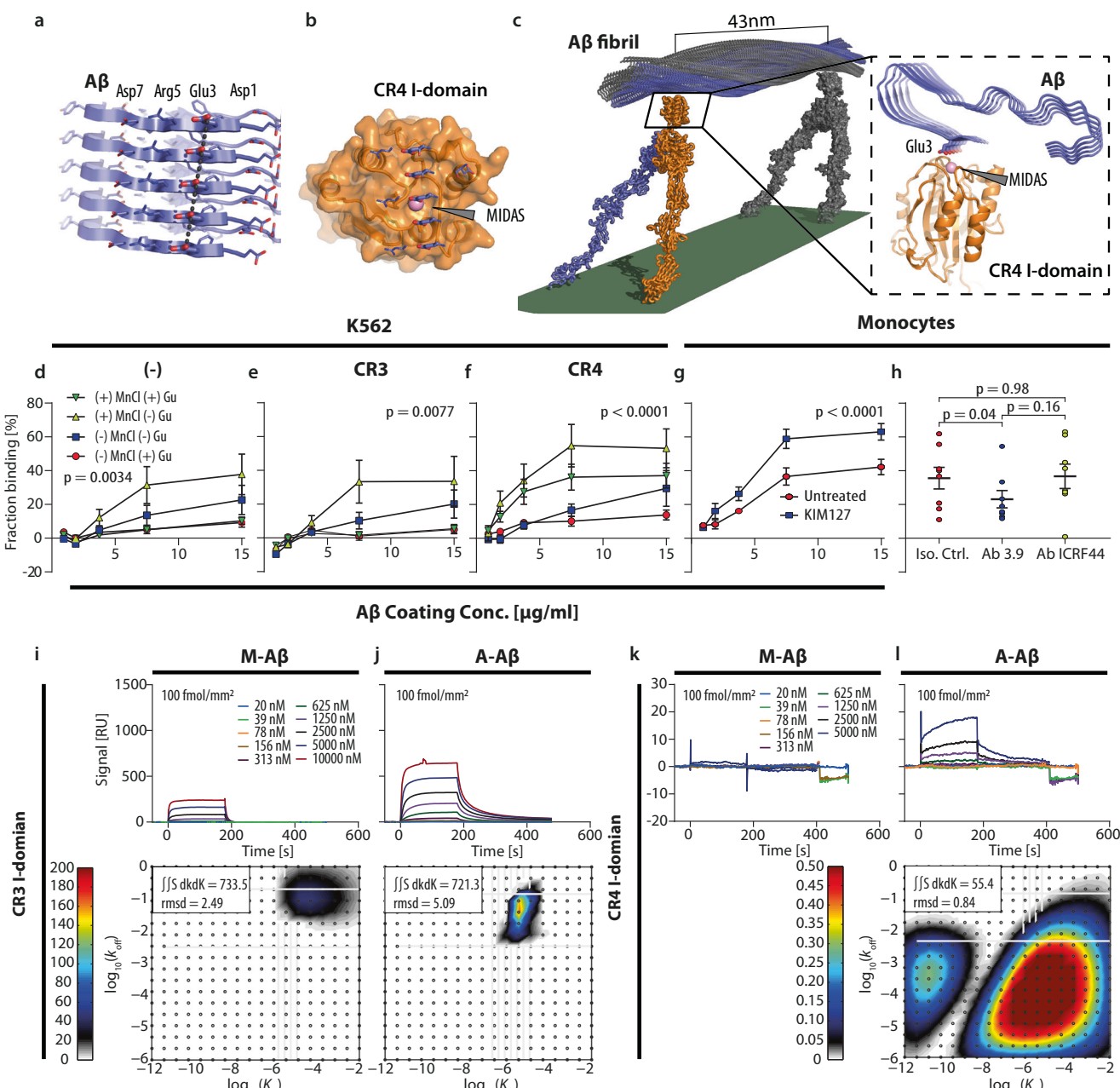

**Fig. 5 | CR4 binds preferentially aggregated Aβ. a** Stacking of 5 parallel beta sheet strands of an Aβ amyloid generated from the PDB entry #6SHS. **b** The ligand binding αₓI domain of CR4 (PDB #4NEH) was docked onto one Glu-3 side chain from the fibril to coordinate the Mg²⁺ ion (in blue) in the metal ion-dependent adhesion site of the I domain. The other negative charge of the Glu-3 ladder fitted a mainly positively charged groove in the I domain. **c** Model of the CR4 ectodomain on a cell surface in contact with a segment of a mature Aβ fibril repeating the orientation of CR4 I domain binding site with a periodicity of ~43 nm. Adhesion to surfaces coated with Aβ by parental K562 cells (**d**) or with recombinant CR3 (**e**) or CR4 (**f**) expression. Surfaces were treated with Gu·HCl or left untreated *n* = 5 biologically independent experiments; mean ± SEM. Statistical analyses were made in a two-way ANOVA test with Geisser–Greenhouse correction. **g, h** Monocyte adhesion to aggregated Aβ. **g** Adhesion was compared in the presence or absence of CD18 integrin-activating KIM127 Ab. Monocytes from *n* = 3 donors used in biologically independent experiments; mean ± SEM, were using a two-way ANOVA test with

Geisser–Greenhouse correction. **h** Adhesion of CR3 and CR4 was tested using function-blocking Abs ICRF44 and Ab 3.9 to CR3 and CR4, respectively, with iso-typic IgG1 Ab as control. Surfaces were coated with Aβ at 2 µg/mL. Monocytes from *n* = 8 donors were used in biologically independent experiments and analyzed in repeated-measures one-way-ANOVA with Turkey's correction for multiple comparison; mean ± SEM. SPR analysis of CR3 (**i, j**) and CR4 I-domain (**k, l**) binding to aggregated and monomeric Aβ coated at a density of 100 fmol Aβ42/mm². All sensorgrams were fitted using EVILFIT. The ensemble of 1:1 interactions was distributed according to equilibrium dissociation constant ($K_D$) and dissociation rate ($k_{off}$) with the abundance in arbitrary resonance units (RU) shown in colors. The total sum of binding ($\iint dk_{off} dK_D$) and root-mean-square deviation (rmsd) between the experimental data and the model, both in RU, are indicated for each analysis. All results are representative of three biologically independent experiments. Source data are provided as a Source Data file.

by the incubation with A-Aβ or M-Aβ. Further, unlike for the expression of CD11c, KIM127 lowered significantly the CD11b expression (Fig. 7f). At the transcriptional level no difference was observed for *ITGAM* or *ITGAX*, neither with low nor high uptake of Aβ (Supplementary Fig. 12).

We then investigated the role of CR4 in iMGs phagocytosis. In iMGs, the CR4 was functionally blocked by the use of the 3.9 Ab, which was also used to show the role of CR4 in monocyte binding of A-Aβ (Fig. 5h) and previously also CR4 binding of aggregated α-synuclein[29]. The

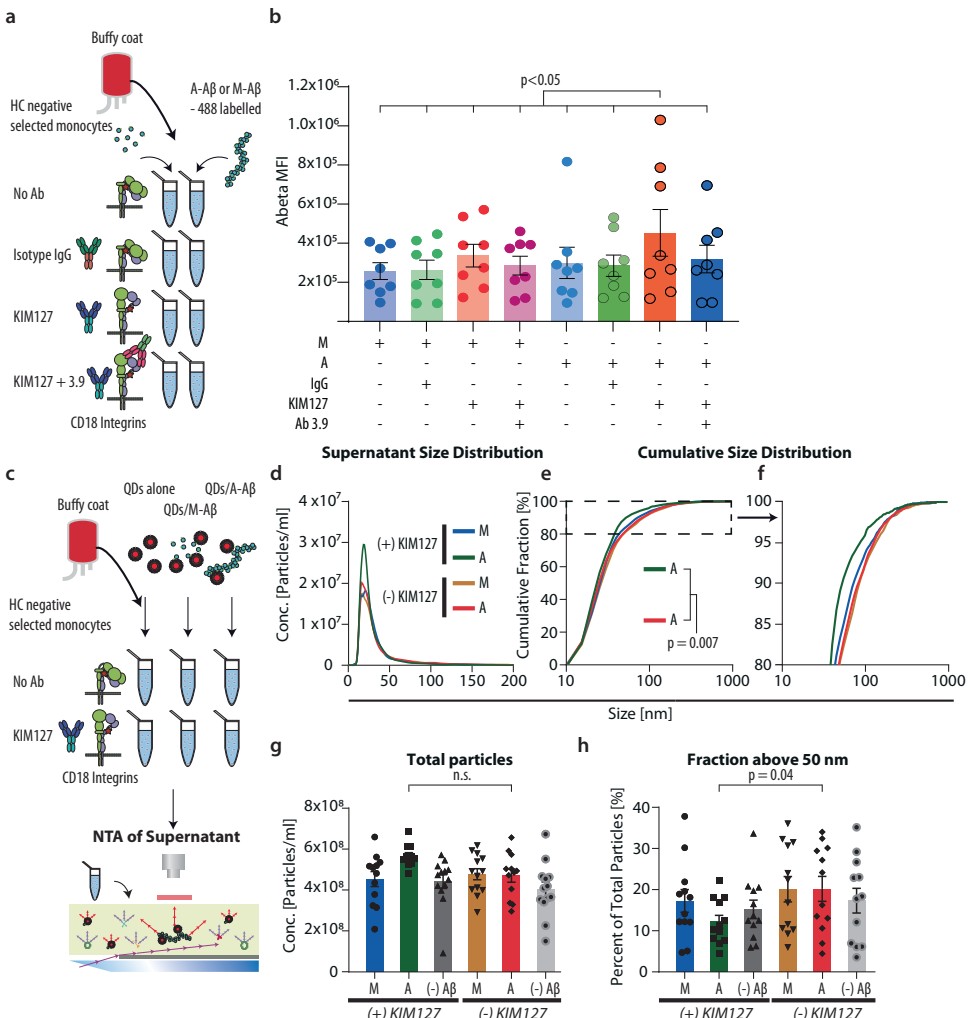

**Fig. 6 | KIM127 integrin activation in monocytes promotes size-selective phagocytosis of Aβ. a** Schematic of experiments with monocyte phagocytosis of Hilyte-labeled aggregated or monomeric Aβ, either in the presence or absence of control IgG, CD18 activating KIM127 Ab and/or CR4 function blocking Ab 3.9. **b** Aβ uptake influenced by Aβ aggregation, KIM127 activation, and CR4 function blocking. **c** Schematic of experiments with monocyte phagocytosis of QD-labeled aggregated or monomeric Aβ, either in the presence or absence of CD18 activating KIM127 Ab and subsequent size-distribution analysis using NTA. **d**–**h** Size distribution in supernatant after phagocytosis of M- or A-Aβ. Distribution of particles between 0–200 nm, with or without KIM127 activation (**d**). Normalized cumulative

size distribution (**e**) with a magnification of the interval reaching 80–100% of particles (**f**). Summary of results for all donors showing total particles detected in supernatants after phagocytosis (**g**). Fraction of particles above 50 nm left in supernatant after phagocytosis (**h**). In (**b**) monocytes from $n = 8$ donors in biologically independent experiments were analyzed with repeated-measure one-way ANOVA with Holm–Sídák's correction for multiple comparisons; mean ± SEM. **d**–**h**, monocytes from $n = 13$ donors in biologically independent experiments were analyzed in a one-sided Kolmogorov–Smirnov test (**e**) or in a repeated-measure one-way ANOVA with Dunnett's correction for multiple comparisons; mean ± SEM. Source data are provided as a Source Data file.

blocking of KIM127 Ab-activated CR4 significantly ($p = 0.005$) reduced phagocytosis (Fig. 7g and Supplementary Fig. 13), comparable to the reduction in monocyte adhesion to A-Aβ (Fig. 5h) and aggregated α-synuclein[29] when a similar functional knock-out was made. These findings suggest a functional coupling of A-Aβ (phagocytosis) and CR4 expression in iMGs.

## Discussion

Here, we combine analyses of clinical samples and paired radiological data from a cohort of MCI patients with hydrodynamic calculations on Aβ aggregate hydrodynamics, and experiments in vitro, and in vivo. From these data, we now propose a link between monocyte recognition of Aβ plasma aggregates and the regulation of AD-like brain pathology in MCI patients.

Blood and CSF of MCI and AD patients contain Aβ aggregates as shown by techniques such as super-resolution microscopy[11] and fluorescence correlation spectroscopy, which are highly sensitive for

the detection of aggregated peptide species[9,46]. In CSF, higher concentrations of aggregates with a size of ~150 nm were found in AD patients compared to MCI or healthy controls[11]. Small Aβ aggregates were reported in the blood of AD patients, but no indication of the particle size or size distribution was reported[8,9]. We used a newly developed FDM-NTA protocol[31], which works directly on liquid biopsies without sample adsorption to glass surfaces as required for super-resolution microscopy[11]. The protocol works well for the detection of highly anisotropic and polydisperse molecular species, such as shown here and in previous studies[31,47] with a reasonable agreement between the apparent $D_H$ and geometric properties of the target particles[31]. With this tool, we now identify a population of large Aβ aggregates with $D_H$s between 600–900 nm in the blood of MCI patients but not in controls. Since recognition was made with Ab specific for Aβ amyloids[48], these aggregates had an amyloid character. The FDM-NTA assay also agreed with findings from other studies, notably the dominance of Aβ aggregates in the range of $D_H$ ~50–150 nm in CSF[11].

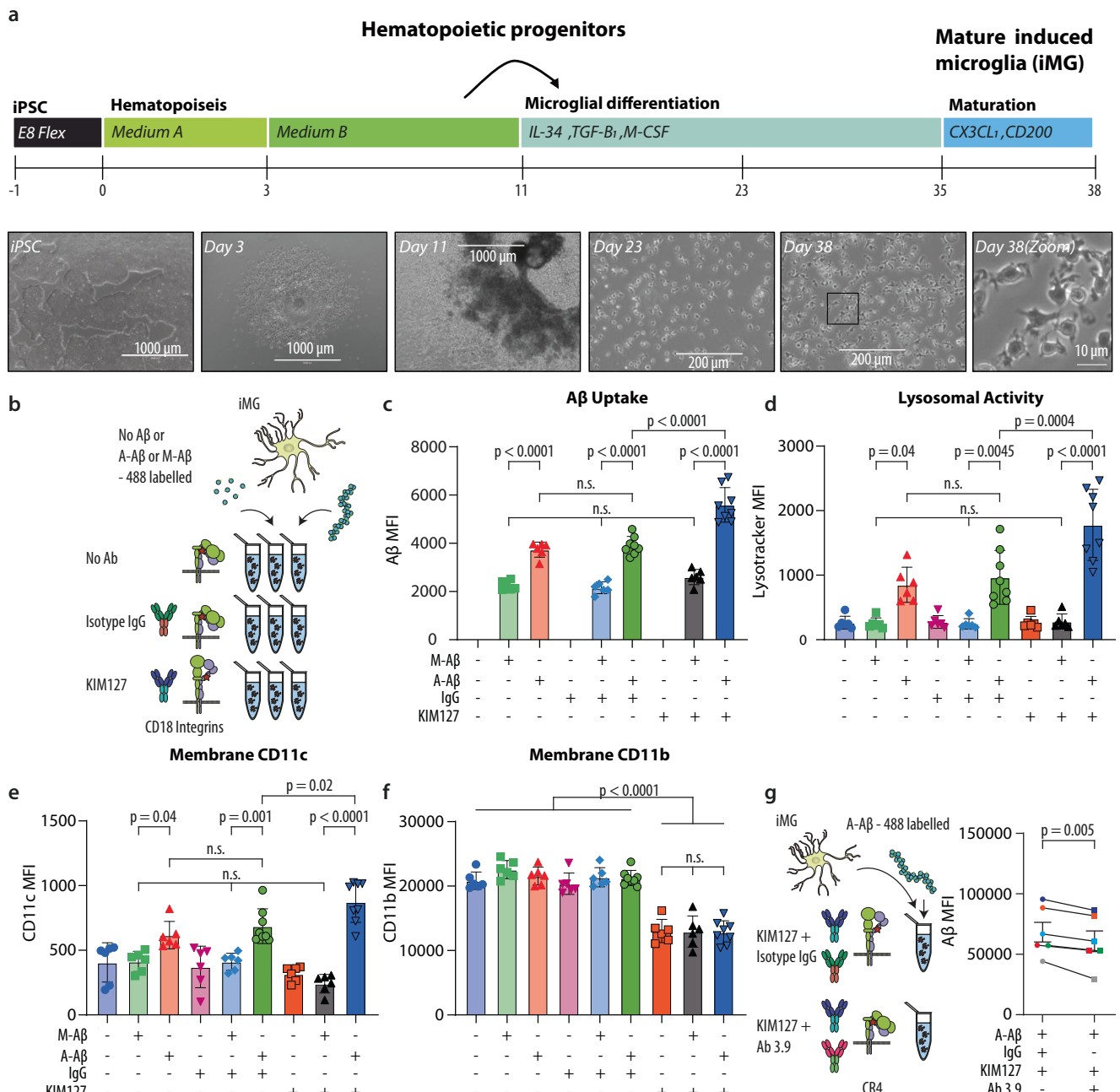

**Fig. 7 | KIM127 integrin activation in iPSC-derived microglia promotes CR4-dependent, size-selective phagocytosis of Aβ and enhances CR4 membrane expression. a** Schematic outline of the iPSC-to-microglia (iMG) differentiation protocol and phase-contrast images of iPSCs, HPs, and iMG at different stages of the microglia differentiation protocol. Scale bar represents 1000 μm (iPSC, Day 3, and Day 11) and 200 μm (Day 23 and Day 38). **b** Schematic of experiments with iPSC-derived microglia phagocytosis using HiLyte™ 488-labeled Aβ42 and KIM127 activation. **c** Aβ uptake influenced by Aβ aggregation and KIM127 activation. **d** Lysosomal activity influenced by Aβ aggregation and KIM127 activation. CR4 (**e**)

or CR3 (**f**) surface expression as influenced by Aβ aggregation and KIM127 activation. **g** Functional knock-out of CR4 using the function-blocking Ab 3.9 and subsequent microglia ability to phagocytose A-Aβ. In c-f, cells iPSC-derived microglial cultures were tested in biologically independent experiments and statistically analyzed using repeated-measure one-way-ANOVA with Turkey's correction for multiple comparisons ($n = 8$ for A (+) IgG (+) and A (+) KIM127 (+). All other conditions $n = 6$; mean ± SD). In (**g**), cells from $n = 6$ cultures were tested in biologically independent experiments and statistically analyzed using a two-sided paired $t$-test; mean ± SD. Source data are provided as a Source Data file.

Supporting the pathophysiological significance, we could stratify monocyte subsets[34] in MCI patients according to the presence or absence of large aggregates. The percentage of non-classical monocytes increased with the presence of aggregates. Compared to classical monocytes, the non-classical subset and their intermediate precursors share a high expression of CD18 integrins[49]. Large aggregates correlated negatively with CD18 expression in intermediate and non-classical subsets. For unclassified, dendritic cell-like monocytes, a

correlation was found with the expression of the KIM127 epitope. Increased KIM127 epitope expression in monocytes was previously reported for disease with a substantial monocyte invasion causing organ inflammation[50]. Consequently, the select depletion of CD18-high expressing monocytes from the blood may reflect extravasation of these monocytes, in good agreement with the central function of these receptors[22] and further supported by the correlation with the KIM127 epitope exposure confirming integrin activation.

The interpretation made above opens three connected questions: where are the adhesive substrates formed, what monocyte-expressed adhesion molecules are central, and how the monocyte adhesion contributes to amyloid regulation in the brain? Of course, the overarching explanation should include why large Aβ plasma aggregates integrate these processes.

First, under flow conditions resembling those in brain blood vessels, we now show by hydrodynamic calculations that large (~700–900 nm), but not small (~100–200 nm), particles enable inertial focusing according to the fundamental principles of the Segrè-Silberberg ring[35]. This effect is pronounced for vessels with diameters 4–8 μm, close to the diameter of human and murine cerebral capillaries of the cortex[17,18]. In our model, when flow directions are altered by vessel morphology, 900 nm particles hit the cell wall with approximately 10,000-fold more kinetic energy than 40 nm particles. A limitation in these calculations is that we cannot establish the absolute kinetic energy required for making particles stick in the endothelial glycocalyx. Nevertheless, the results are theoretical support for how the large aggregates may preferentially form a substrate in thin vessels. Experimental evidence may be extracted also from a recent study of engineered particles with $D_H$s of ~140 nm injected in mice[18]. These particles associate with the 4-μm capillary and ~8–16 μm post-capillary venules but not quantitatively with vessels with larger diameters. It should be noted that engineered particles usually expand with as much as 80% in biological fluids from the formation of a corona of proteins[51]. Although the size requirement in the experimental model seems smaller than predicted in the theoretical model, both analyses support that the large Aβ aggregates will associate with the cortical microvasculature in the brain critical for leukocyte extravasation into the brain, further enhanced by the high vascularization of the cortex and abundance of leukocyte in the adjoining meninges[52]. While our work supports monocytic CD18 integrin-mediated extravasation in MCI patients, further studies in humans remain to establish the broader connection between monocytes in these compartments and the inflammatory response in the patients.

Second, we found that the A-Aβ is a potent ligand for CR4, while the same protein in its monomeric form is not, as also observed for α-synuclein[29]. When considering, on the one hand, the two proteins share practically no sequence homology, and, on the other, the polyanionic motif shared between the Aβ and α-synuclein only exists in their amyloid forms, we propose that CR4 is a receptor for amyloid aggregates through this motif. The motif requires approximately 5 Aβ chains to form a binding site for one integrin I domain. In addition, it is well established that integrins are usually clustered in the cell membrane by multivalent ligands, increasing the binding strength of potent ligands through avidity[41]. In a simple fibril, binding sites with an identical orientation relative to the integrin-presenting cell membrane are separated with a distance of ~43 nm. Even a modest number of 5–6 formed integrin:ligand interactions using the CR4 motif recognition would, in this model, require a fibril of 200 nm, in agreement with our findings from monocyte internalization of A-Aβ and further supported by our earlier work on α-synuclein. Of course, as also shown in our work, the aggregated forms of Aβ are far from the simple structure of a single fibril. Instead, they are aggregates of many fibrils, likely presenting a more disordered array of integrin ligand binding sites. Our work does not include any theoretical or experimental estimation of the number of integrins binding these aggregates. Nevertheless, from the same principles as suggested for fibrils, it may be the case that only larger aggregates will bind with strong avidity to CR4-expressing cells. This points to a requirement of multiple CR4 receptors binding to Aβ as explaining why only large forms of these aggregates are taken up by monocytes and iMGs. Indeed, the formation of strong integrin binding is supported by our cell adhesion assays, where CR4 binding of aggregated Aβ was much better than for monomeric protein and, likewise, in phagocytosis assays with monocytes and iMGs. The

centrifugal force applied in the adhesion assays mimics the shear stress[42] experienced by monocytes in blood vessels. CR4-mediated binding under such forces to large Aβ aggregates shows it can sustain prolonged monocyte adhesion to permit crawling[20] in the capillaries and extravasation in the post-capillary venules under physiological conditions.

Third, to our knowledge, no acceptable methodology is available for directly following leukocyte infiltration into the brain of MCI and AD patients, mainly because of the long duration of the processes leading to disease. On the other hand, PET tracers for staining Aβ and neurofibrillary tau tangle accumulation in the brain are proven effective for following disease-related pathology in living patients. To link brain pathology with a role of the peripheral immune system, we compared the development of brain pathology with the presence or absence of large Aβ plasma aggregates (Fig. 8a–c). Our analysis at the 2-year follow-up showed that large >600 nm Aβ Agg [+] MCI patients had almost no traceable AD-like pathology (Fig. 8b). Large >600 nm Aβ Agg [−] patients presented both Aβ and tau amyloid. This included lower [11]C-PiB retention in amyloid Aβ across the entire cortex of the brain for large Agg [+] compared to large Agg [-]. In this way, the PET scans confirm no build-up of AD-like pathology in the cortex, and other parts of the brain, on the condition of a source of large aggregates in plasma. This may seem counterintuitive, especially since ref. 12 recently reported the presence of large, water soluble aggregates in the brain of AD patients. However, for understanding the relation between such aggregates and brain pathology in the case of MCI patients, we propose that a dynamic perspective on leukocyte trafficking may apply, involving both monocyte diapedesis and export of Aβ aggregates from the CNS to blood. Integrating our flow cytometric data and hydrodynamic calculations, together pointing to the depletion of monocytes from circulation, with the PET scans, a three-step model of events in MCI develop (Fig. 8b). Plasma aggregates (i) preferentially associate with the luminal side of thin cortical vessels, (ii) enable monocyte crawling through CR4-mediated adhesion to amyloid Aβ, and (iii) eventual permit extravasation via post-capillary venules into the brain. In this model, the lack of AD-like pathology in large >600 nm Aβ Agg [+] MCI patients is indirect evidence of enhanced clearance of brain pathology from an increased number of infiltrating phagocytic cells. Both monocytes and iMGs are capable of amyloid clearance in a CR4-involved process. We cannot establish in our model if the infiltrating monocytes in vivo are a significant source of such clearance or if further differentiation into a more microglial-like phenotype is required. Also, CD18 integrins other than CR4 may play a role in this process, notably CR3. Still, none of our other in vitro experiments suggests CR3 to be a strong binder of Aβ in any form.

Why the large Aβ aggregates are absent in plasma from MCI patients with AD-like pathology is not explained by our data. Others have suggested that AD involves dysfunction of the glymphatic system, which empties waste, including aggregated protein, from the brain into the lymphatic vasculature, further exporting their contents into the venous drainage[53]. Indeed, this is supported by the finding of small Aβ aggregates both in CSF and blood[9,11]. In MCI patients, Aβ aggregates are found in the CSF[11], and are emptied into the blood with consequences for cerebral monocyte recruitment as mentioned above. From the finding of large, soluble Aβ aggregate in the brain of AD patients[12], it may be speculated during the progression of MCI to AD that the soluble aggregates have grown to a size, which limits their emptying from the brain into blood (Fig. 8c).

We have previously shown that conformational activation of CD18 integrins induced phagolysosomal activity[29]. In iMGs, aggregated Aβ further added such action to the KIM127-potentiated cells, suggesting a case of integrin ligand-driven, outside-in signaling[54]. The cell-membrane expression of CR4 was increased upon stimulus with aggregated Aβ. At the same time, transcription of the *ITGAX* gene was not altered. This resembles previous observations from other myeloid

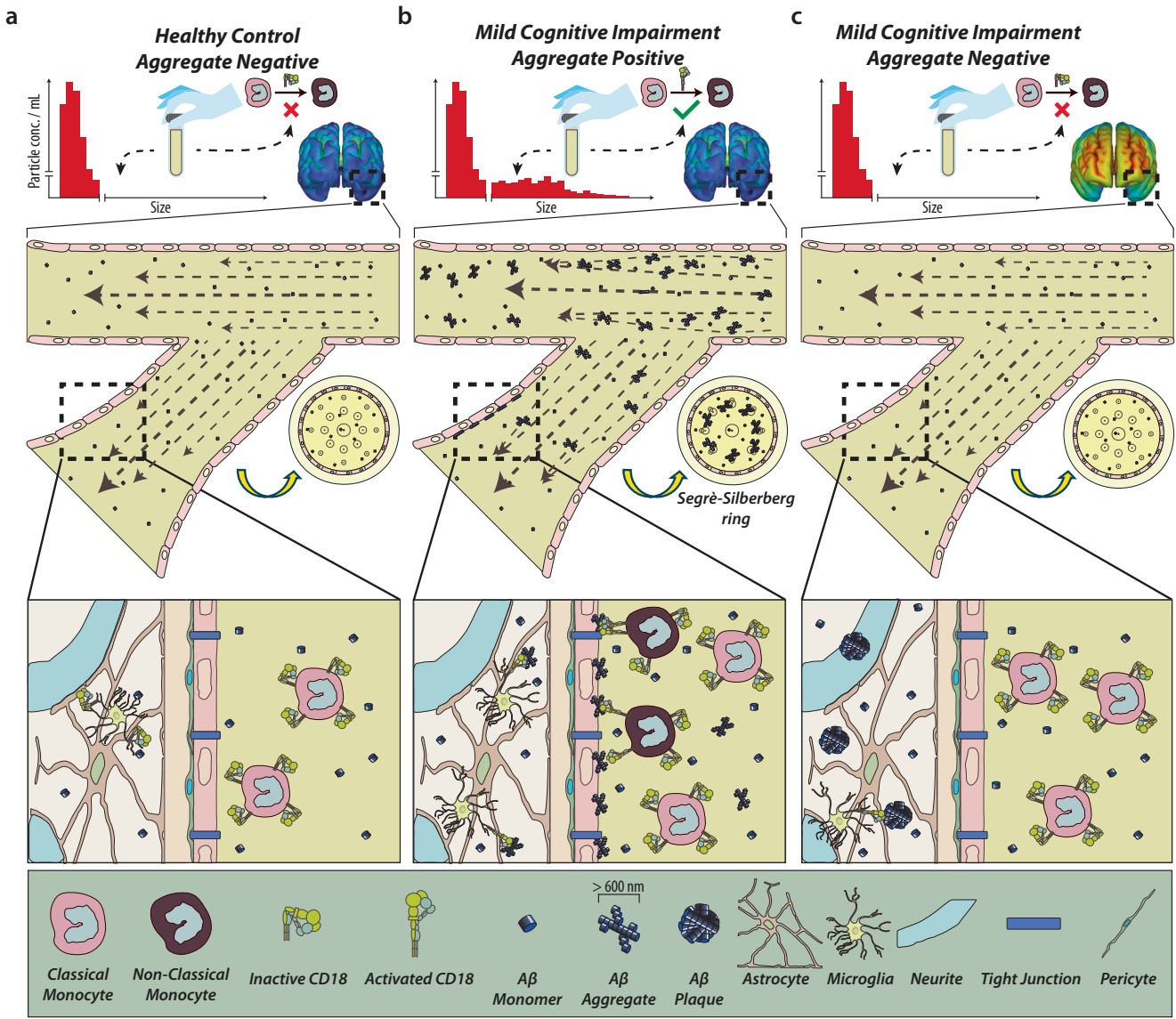

**Fig. 8 | The presence of >600 nm Aβ aggregates in plasma results in Aβ deposition in the cerebral microvasculature, supports monocyte adhesion, and low levels of AD-like pathology in MCI patients. a** HCs have no large Aβ aggregates in plasma and, as a result, no recruitment of CD18-rich, non-classical monocyte subsets to the cerebral vasculature. **b** MCI large Agg. [+] patients present a deposition of the large Aβ aggregates in cerebral capillaries and post-capillary venules, supporting CR4-mediated adhesion of the CD18-rich, non-classical monocytes. The low brain pathology, as recorded in the PET scans, supports a model where extravasated monocytes adds to the microglial clearing of pathologic Aβ aggregates in the cerebral cortex. **c** Large Agg [−] MCI patients lack monocyte depletion from blood, reducing extravasation of these cells. As a result, amyloid aggregates build up in the cortex to form large PET-detectable plaques.

cells, which can release a pool of intracellularly stored CD18 integrins, including CR4, upon appropriate stimulus[55]. By contrast, CR3 membrane expression or *ITGAM* gene transcript was not changed by Aβ.

Taken together, our findings suggest CR4 as part of a finely orchestrated cellular function for clearance of amyloid Aβ aggregates. Among the limitations of our work, it must be stated that the CD18 integrin relevance in MCI was derived from correlation studies. Furthermore, the number of MCI patients analyzed was small and no cohort of AD patients were included. These shortages, together with a need for animal work combining studies of AD development and CR4 function, should encourage external validation, also to explore if CR4 may be a promising target for intervention to reduce AD morbidity.

## Methods

### Ethical oversight of animal and human experiments

All animal experimentation was conducted following approval by local authorities of the Federal State of Berlin (X9007/17). All test animals lived in standardized conditions where temperature, humidity, and hours of light and darkness were maintained at a constant level, all year round. The air was approximately 22 °C and the humidity approximately 55%. Light switched on and off automatically, imitating a day-night rhythm.

The human iPSC line BIHi043A-XM001 was used[56]. All ethical approvals are deposited in the Human Pluripotent Stem Cell Registry (https://hpscreg.eu/) under the accession code #BIHi043-A.

The use of donor blood for isolation of human monocytes was ethically approved by the Aarhus University Blood Bank (Protocol no. 77).

For the analysis and collection of samples from MCI patients, the "Central Denmark Region Committees" approved the study (Protocol No. 1-10-72-191-14) on health research ethics following the Declaration of Helsinki, and all subjects provided written informed consent before participating, as described earlier[5]. Patient demographics are shown in Supplementary Table 1. MCI participants were recruited from

Memory/Dementia clinics in Denmark or by advertisement, and healthy controls were recruited by advertisement. All participants were screened and cognitively assessed as previously described[5].

## Quantum dot coupling of antibodies

QD Ab (Ab) coupling was performed using SiteClick Qdot 655 Ab Labeling Kit (no. S10453; Molecular Probes) according to manufactures instructions. In brief, Ab, either "Aducanumab biosimilar" human IgG1,κ to aggregated Aβ (cat. no. PX-TA1335, lot. no. 040721-A01, ProteoGenix), or "Isotype control" human IgG1,κ (cat.no. HG1K, lot.no. MA14OC2803, SinoBiologicals) was concentrated in Ab preparation buffer to a concentration of 2 mg/mL or above. The buffer was simultaneously changed to 0.05 M Tris, pH 7.0. Next, carbohydrates on the Ab were modified by incubation with β-galactosidase for 4 h at 37 °C. Azide modification was achieved through incubation with uridine diphosphate glucose-GalT enzyme overnight at 30 °C. Ab with modified carbohydrates was purified and concentrated through centrifugation steps using a molecular-weight cut-off membrane concentrator. Finally, 5′-dibenzocyclooctyne-modified QD nanocrystals were coupled overnight at 25 °C and, until further use, stored at 4 °C.

## Synthetic Aβ and protein aggregation

Synthetic human Aβ was purchased from Abcam (cat.no. ab120301; Abcam), solubilized in PBS pH 7.4 at a 1 mg/mL concentration, and stored at −80 °C. To generate A-Aβ, Aβ at 1 mg/mL was incubated at 37 °C in sterile conditions for 24 h. At room temperature, A-Aβ was sonicated for 30 min to break up larger fibrils. The aggregational state was validated using NTA diluted to 1, 10, and 50 μg/ml concentration in PBS pH 7.4. A-Aβ contained, on average, $3 \times 10^6$ particles/mL above 600 nm in size. Unaggregated and mainly M-Aβ had less than $0.5 \times 10^6$ particles/mL above 600 nm as detected by aducanumab biosimilar (Supplementary Fig. 2). For FACS, HiLyte 488-labeled Aβ was purchased (cat.no. AS-60479-01; Anaspec). Aggregation was performed at a concentration of 0.5 mg/mL for 24 h at 37 °C in sterile conditions with subsequent 30 min sonication at room temperature.

## Nanoparticle tracking analysis

Samples for NTA were analyzed using a NanoSight NS300 system (Malvern Panalytical). The system was configured with a 405-nm laser, a high-sensitivity scientific complementary metal−oxide−semiconductor Orca Flash 2.8/Hamamatsu C11440 camera (Malvern Panalytical), a syringe pump, and for fluorescence measurements, a 650-nm long-pass filter was used. Before each measurement, the sample chamber was washed twice with 1 mL PBS with 1 mM EDTA (PBS/EDTA). All samples were thoroughly mixed before measurement and injected into the sample chamber using 1 mL syringes. The measurement script comprised temperature control at 23 °C, followed by a 20 s flush at a flowrate mark of 1000. Next, sample advancement was stabilized over 120 s at flowrate mark 10. Recordings were captured continuously during a steady flow at flowrate mark 10 with five 60-s recordings separated by a 5-s lag time between each sample. The videos were collected and analyzed using NanoSight software (version 3.3 and 3.4 with a concentration upgrade; Malvern). Automatic settings were used for the max jump mode, minimum track length, and blur setting. With these settings, the max jump distances were between 19.0 and 22.7. The camera level and detection threshold was adjusted according to sample composition to accommodate differences in sample heterogenicity and ensure optimal sensitivity. The camera level was set for both scattering and fluorescence mode to a maximum of 16. Detection threshold was set to 2 for validation of antibody-QD binding to synthetic Aβ in PBS, at 3 for detection of Aβ in control and MCI plasma, in mouse plasma, in supernatants from monocyte phagocytosis of Aβ and 5 for validation of antibody-QD binding to synthetic Aβ spiked into plasma. For analysis of synthetic Aβ in scatter detection mode the detection threshold was set to 10. Only samples analyzed with the same detection threshold were compared.

Pure protein was analyzed in PBS/EDTA with and without a 1:20,000 dilution of anti-Aβ (Aducanumab bios.) or isotype-matched Ab-QD reporters. Human plasma samples were analyzed at a 1:10 dilution in PBS/EDTA with a 1:200 dilution of anti-Aβ or isotype Ab-QD reporters. Plasma samples from mice were analyzed in a 1:20 dilution in PBS/EDTA with a 1:20,000 dilution of anti-Aβ or isotype Ab-QD reporters.

## Analytical ultracentrifugation of aggregated Aβ

The Aβ stock at 1 mg/mL was prepared as described above. EDTA solution at 500 mM, pH 8.0, was added to the Aβ stock to make the master Aβ solution with 1 mM of EDTA, further diluted n phosphate-buffered saline (pH 7.40) with 1 mM EDTA. Sedimentation velocity analytical ultracentrifugation (SV-AUC) experiments were performed in an Optima XL-I instrument (Beckman Coulter) calibrated with external temperature and radial calibration tools[57]. To minimize the impact of the systematic noise on the interference (IF) data, all AUC cells were preconditioned by following the standard protocol, followed by acquisition of IF scans of water blanks which were subtracted from the raw data in the subsequent data analysis. To effectively monitor the particles with a large size range, the standard sedimentation velocity protocol was used with three rotor speeds, 3000, 10,000 and 40,000 rpm ($-700 \times g$, $-7800 \times g$ and $-125,000 \times g$, respectively) at 20 °C[58,59]. After a short temperature equilibration (-20 min), AUC cells were filled with samples and initially accelerated to 3000 rpm ($-700 \times g$) and data acquisition was initiated using Rayleigh interference optics and absorbance optics at 280 nm. After 30 min centrifugation, the rotor was accelerated to 10,000 rpm ($-7800 \times g$) and IF and absorbance data were acquired for 20 min, followed by the 3rd step with acceleration to 40,000 rpm ($-125,000 \times g$) and data acquisition for 10 h. Data analysis was carried out using the program, SEDFIT (version #16p47, National Institutes of Health). All the SV scan files were loaded into SEDFIT and the scans were sorted and corrected for the speed dependent rotor stretching[58]. The standard sedimentation coefficient distribution $c(s)$ was calculated for each sample[60].

## Scanning electron microscopy

$8 \times 8$ mm pieces of precut 4-inch silicon wafers were precleaned in oxygen plasma (in an MK II Advanced Vacuum, with settings at 50 W and 0.1 mTorr for 120 s). The wafers were then coated via physical vapor deposition by electron-beam stimulated thermal evaporation in a CryoFox-GLAD (Polyteknik). Wafers were covered with a layer of Ti (20 nm at 0.1 nm/s base pressure $<1 \times 10^{-7}$ Torr) or Au (20 nm at 0.1 nm/s with a 2 nm Ti adhesion layer base pressure $<1 \times 10^{-7}$ Torr) and transferred to 48-well plates within 1 h (cat.no. 150687, Nunclon Delta Surface™, ThermoFisher Scientific) containing 600 μl PBS/1 mM EDTA (PBS/EDTA). A total of 400 μl PBS/EDTA was removed, and 100 μl monomeric or aggregated Aβ was added to the wells to a final concentration of 5 μg/mL. A 1:200 dilution of aggregated Aβ-specific (aducanumab bios.) or isotype Ab-QD conjugate was added, and samples were incubated for 30 min at room temperature. Samples were washed by transfer of wafers into 600 μl PBS/EDTA twice and then into 600 μl distilled $H_2O$ twice before drying with pressurized nitrogen. Images were recorded in a Magellan 400 ultra-high resolution SEM (Field Electron and Ion Company), optimized for contrast using a 3 keV beam with a nominal current of 25 pA (spot size ~1 nm). The particle positions were tracked with ImageJ[61] "Analyze Particles function," corrected by manually marking particles, and then analyzed for nearest neighbor distance. Clusters with a maximum interparticle distance of 100 nm were considered a single cluster. The clustering for experimental images was compared with random two-dimensional distributions of dots generated in silico with 0 to 600 dots.

## APOE genotyping of MCI patients

Anti-coagulated whole blood was collected in EDTA-containing tubes and frozen to allow DNA extraction in batches using the QiaSymphony

SP platform (Qiagen). Polymerase chain reaction (PCR) and Sanger sequencing were used to determine *APOE* genotypes. PCR was performed using primers flanking rs429358 and rs7412 and Q5® Hot Start High-Fidelity DNA Polymerase (New England Biolabs) using 20% Q-solution (Qiagen) in a BioRad C1000 or S1000 thermocycler (BioRad). The amplification reaction was carried out in a volume of 50 µL containing 100 ng of genomic DNA and 10 uM of each primer (Forward primer: 5′-CTC CCA CTG TGC GAC ACC-3′ and Reverse primer: 5′- GGG CTC GAA CCA GCT CTT-3′). The PCR cycling conditions were as follows: Initial denaturation at 98 °C for 30 s, followed by 35 cycles with denaturation at 98 °C for 10 s, annealing at 63 °C for 30 s, extension at 72 °C for 30 s, then a final extension at 72 °C for 2 min. All PCR products were visualized on a 1.2% agarose gel with EtBR and purified using ExoSAP-IT (Thermofisher) and sequenced using BigDye terminator version 1.1 (Thermofisher). The PCR products were sequenced with 10 uM of the forward sequencing primer (Forward primer: 5′- TCG AAC TGG AGG AAC AAC CT-3′) on an ABI 3730xl DNA Analyzer systems by GATC Biotech. Sequence traces were aligned to the *APOE* reference sequence NM_000041 using SeqScape version 2.7 (Thermofisher). This allowed the determination of whether the *APOE* genotype was ε2/ε2, ε2/ε3, ε3/ε3, ε3/ε4, or ε4/ε4. ε1/ε3 and ε2/ε4 were interchangeable and, therefore, impossible to distinguish.

## Measurement of soluble and insoluble Aβ in the PDAPP mouse

All experiments were done with 52 weeks old mice on the C57Bl6N background. For analysis of amyloid precursor protein (APP) processing products, 5 male and 6 female mice hemizygous for the human APP transgene (isoform 770; V717F Indiana mutation) were used. The approximately equal sex balance was chosen to avoid any potential bias from using only sex in the experiments. The mice were sacrificed, and the cortex and hippocampus of the brain were homogenized in ice-cold homogenization buffer (250 mM sucrose, 2 mM MgCl$_2$, 20 mM Tris-HCl, pH 7.5) supplemented with protease inhibitor cocktail (cat. no. 05892791001, Roche). Following incubation for 30 min on ice, nuclei and cellular debris was removed by a centrifugation step (1000 × $g$, 10 min, 4 °C). The protein solution was spun down in an ultracentrifuge (175,000 × $g$, 45 min, 4 °C), and the supernatant was used to detect soluble Aβ species. The pellets were resuspended in lysis buffer (50 mM Tris-HCl, 2 mM MgCl$_2$, 80 mM NaCl, 1% [v/v] Triton X-100, 1% [v/v] NP40, pH 8.0) supplemented with protease inhibitor cocktail. The lysates were again centrifuged (100,000 × $g$, 45 min, 4 °C), and the supernatants were discarded. The remaining pellet was lysed in 70% (v/v) formic acid in TBS (50 mM Tris-HCl, 150 mM NaCl, pH 7.6) and centrifuged again (175,000 × $g$, 45 min, 4 °C). The supernatants were used to detect insoluble Aβ species. Human Aβ species were determined in multiplex biological assay (cat.no. K15199E, Meso Scale Discovery) using SQ120 Imager as a read-out device. All measurements were carried out according to the manufacturer's protocol.

## Flow cytometry of human PBMCs

Samples of PBMCs were thawed in a water bath at 37 °C, washed in PBS/20% (v/v) FCS, counted, and resuspended in stain buffer (PBS/0.5% [w/v] BSA/0.09% [w/v] NaN3) at 10 × 106 cells/mL. Immediately, 1 × 10⁶ cells were stained in 170 µL stain buffer (incl. 50 µL brilliant violet stain buffer) with anti-human CD14 AF488 (clone "63D3", cat. no. 367130, lot.no. B264871, Biolegend), CD16 BV605 (clone "3G8", cat. no. 563172, lot. no. 7299970, BD Bioscience), CD18 BV421 (clone "6.7", cat. no. 562871, lot. no. 7223827, BD Bioscience), and CD163 PE (clone "MAC2-158", cat.no. IQP-570R, lot. no. 170252, IQ Products BV). To recognize conformationally activated CD18, the Ab KIM127 (custom-made from American Tissue Culture Collection's hybridoma CRL2838 by GenScript, lot.no. A214111033) was conjugated to APC using Lightning-Link APC conjugation kit (Innova Biosciences, cat. 705-0010). All Abs were applied at the concentrations indicated in Supplementary Fig. 7. Cells were stained with LIVE/DEAD™ Fixable Near-IR

Dead Cell Stain Kit (cat.no. L10119, Life Technologies) for 30 min at 4 °C. Subsequently, cells were washed twice in cold stain buffer, fixed with PBS/0.9% (v/v) formaldehyde, and analyzed using a Novocyte Analyzer (Acea Biosciences). Compensation was done using single-stained PBMC samples. Data were analyzed using FlowJo 10.4.2 for PC. Monocyte subsets were gated, as shown in Supplementary Fig. 7.

## MRI & PET Neuroimaging

Magnetic resonance imaging (MRI) and PET data was analyzed with MINC (Medical Imaging NetCDF) Toolkit (https://bic-mni.github.io).

MRI was used to obtain individual high-resolution structural images for PET co-registration and brain segmentation. The structural images were acquired with a T1-weighted MP2RAGE sequence with 1 mm isotropic voxels (TR = 5 s, TE = 2.98 s, TI$_1$ = 0.7 s, TI$_2$ = 2.5 s, acquisition matrix 240 × 256 × 176, 4° and 5° flip angles, FOV = 240 × 256 × 176 mm) and preprocessed using a framework suitable for longitudinal data[62]. First, the images were denoised and bias field corrected and then moved to Montreal Neurological Institute (MNI) space. Here, the images were segmented into gray matter (GM), white matter (WM) and cerebrospinal fluid (CSF) and subsequently classified into specific structures, including frontal, temporal, parietal, occipital and cerebellar GM and WM. In addition, the structural images were used to extract the surface corresponding to the midsection of the cerebral cortex using Fast Accurate Cortex Extraction (FACE)[63].

PET imaging was performed on a High-Resolution Research Tomograph (ECAT HRRT; CTI/Siemens). Tau-PET was acquired with ¹⁸F-Flortaucipir (¹⁸F-FTP) for 40 min, 80–120 min post-injection, with a target dose of 370 MBq, and amyloid-PET was acquired with ¹¹C-Pittsburgh Compound B (¹¹C-PiB) for 50 min, 40–90 min post-injection, with a target dose of 400 MBq. Both tracers were administered intravenously as a bolus followed by a 10 mL saline flush. PET acquisition was performed in list mode and each acquisition was subsequentially rebinned into 10-min frames. After reconstruction, the image frames were averaged and divided by the mean uptake in the individual cerebellar GM to obtain a standardized uptake value ratio (SUVR) image. The SUVR images were then smoothed with a 3 × 3 × 3 mm full width at half maximum (FWHM) Gaussian kernel and subsequently mapped to the corresponding cortical surface using trilinear interpolation to produce a parametric surface. This process aims to sample the PET signal where the spill-in signal from neighboring compartments (CSF and WM) is minimal, thus maximizing signal from cortical GM. Individual parametric surfaces were then moved to a common template surface and smoothed with a 20 mm FWHM geodesic Gaussian kernel prior to the statistical analyses. Smoothing on the cortical surface limits spill-in from signal outside the cortex and avoids smoothing of signal across sulci[64]. For each individual, the mean SUVR of ¹⁸F-FTP was calculated in a composite ROI comprising the entorhinal, amygdala, parahippocampal, fusiform, inferior temporal, and middle temporal cortical regions. Likewise, the individual mean ¹¹C-PiB SUVR was calculated in a composite regions of interest (ROI) comprising the prefrontal, orbitofrontal, anterior and posterior cingulate, precuneus, parietal, and temporal cortical regions. These cortical regions have been identified to contain elevated tau and Aβ aggregation, respectively, in AD[65].

## Hydrodynamic simulation of particle movement in blood vessel

The finite element method as implemented in Comsol modeling software (version 6.1, Comsol, Inc.) was used to solve the stationary Navier-Stokes flow problem in a long straight and a branched blood vessel (Supplementary Fig. 14). The dimensions of the vessels follow ref. 17. As boundary conditions, a fully developed flow field, given an average velocity as described[66], was assigned to the surface representing the inlet. A pressure condition was applied to the surfaces representing the outlets[67]. Fluid behavior against the endothelium is described by a slip boundary condition to endothelium surfaces[68]. Values for blood

rheological parameters (viscosity and density) was set as previously described[69]. The solution to Navier-Stokes equations provides velocity and pressure fields. Those fields give rise to drag and lift forces, acting upon particles released into the vessel through the inlet. In this temporal problem, the Comsol algorithm solved the Newtonian force balance of 20,000 particles released and calculated their spatial trajectory. A counter was used to record if particles struck the model endothelium and record particle position and velocity at the instant of impact. This enabled the calculation of kinetic energy carried by the particles and delivered to the endothelium. The straight vessel case use 2D rotational symmetry and we studied how a homogeneous distribution of particles position themselves downstream. In the second case we studied how particles already distributed by inertial focusing behave over a branched vessel in 3D. Governing equations, geometries, and used parameters for both the Navier-Stokes and the Newtonian particle tracing problems are listed in the Supplementary Information.

### Structural modeling of CR4 binding of Aβ

A model of the helical Aβ fibril was generated from the protein data bank (PDB) entry #6SHS by repeated superposition of chain G from one copy onto chain D from a second copy. Under the assumption of rigid models of the Aβ fibril and the ligand binding CR4 ($\alpha_X$) I domain (PDB #4NEH), the I domain was docked manually such that a Glu-3 side chain was near the $Mg^{2+}$ ion in the I domain MIDAS site. Clashes between the I domain and the Ab fibril were minimized. In evaluating Asp-1, Glu-3, and Asp-5, the glutamate was selected as the most likely MIDAS ligand based on surface shape and charge complementarity for the CR4 I domain-fibrillar Aβ interaction. All figures were prepared with the program Pymol (www.pymol.org).

### Cell adhesion assays with K562 cells and monocytes

Immortalized myelogenous K562 cell lines with a recombinant expression of CR3 or CR4 were a kind gift from Dr. T.A. Springer, Harvard Medical School, on a materials transfer agreement. They were made and cultivated as described with the parental K562 cell line[70]. The K562 cells were treated as described below for the monocytes, except that integrin activation was achieved by adding 1 mM $MnCl_2$ rather than KIM127 Ab.

Primary human monocytes were purified and stored as described[29]. Just before use, cells thawed and added to PBS supplemented with 20% (v/v) FCS, collected using centrifugation, and resuspended in RPMI 1640 supplemented with 2% (v/v) FCS. Monocytes were fluorescently labeled using incubation with 2,7-bis(2-carboxyethyl)−5(6)-carboxyfluorescein acetoxymethyl ester (cat. no. 14562; Sigma-Aldrich) at 37 °C and 5% (v/v) $CO_2$ for 15 min, washed twice and resuspended in 150 mM NaCl, 5 mM KCl, 1 mM $MgCl_2$, 1.8 mM $CaCl_2$, 10 mM HEPES (pH 7.4), with 5 mM glucose and 2.5% (v/v) FCS (binding buffer) to a final cell concentration of $6–10 \times 10^5$ cells/ml. The cells were centrifuged for 5 min at $230 \times g$, washed twice, and resuspended in binding buffer with 5 μg/mL CD18 integrin-activating Ab KIM127 (GenScript). To test the contribution of CR3 and CR4 to adhesion, 1 μg/mL dialyzed function-blocking Ab to the CD11b (clone "ICRF44", no. 301302; Biolegend) or CD11c chain (clone "3.9", no. MA1-46052; Thermo Fisher Scientific) was added. Mouse IgG1 (cat.no. M5284, Sigma) was added as an isotype control.

Cell adhesion was tested using a centrifugation-based assay[42]. In brief, polystyrene 96-well microtiter plates with V-shaped bottoms (cat. no. 3896, Costar) were coated for 1 h at 37 °C with aggregated Aβ diluted in 150 mM NaCl and 20 mM Tris [pH 9.4] (coating buffer) at concentrations of 0.940, 1.88, 3.75, 7.5, or 15 μg/mL or left uncoated for reference. Each concentration was prepared in triplicate. Some plates were prepared to denature the Aβ structure before adhesion, and 100 μL 6 M guanidine hydrochloride (Gu·HCl) was added for 1.5 h at room temperature. All plates were washed in 200 μL PBS with 0.05%

(v/v) Tween 20 (cat. no. 9005-64-5, Sigma-Aldrich) and further blocked in this buffer for 1 h at 37 °C. The microtiter plates were emptied and kept at 37 °C in a heating block, the 100-μL cell suspension was then transferred to each well, and plates were incubated at 37 °C with $CO_2$ for 10 min. Plates were then centrifuged at $50 \times g$ for 5 min. The fluorescence count was read in the nadir of the wells using a Victor3™ 1420 multilabel counter (Wallac) with 485-nm excitation and 535-nm emission wavelength. Appropriate centrifugation force was achieved using centrifugation at $10 \times g$ for 5 min and again at $50 \times g$ for 5 min. The fluorescence count was recorded at each centrifugation step.

### Surface plasmon resonance assays and data analysis

SPR assays were performed using CM-4 chips (cat no. 50-105-5323, Cytiva) and run in the BIAcore 3000 instrument (GE Health Care). As described, the chip surfaces were coupled with monomeric Aβ, aggregated Aβ, or iC3b using an amine-coupling chemistry method[71]. A reference surface was prepared by coupling the surfaces with ethanolamine rather than protein. The CR3 and CR4 I domains were stabilized in the activated conformation by mutation of a C-terminal isoleucine residue to glycine and prepared as described earlier[45]. The I domains were diluted in running buffer with 150 mM NaCl, 1 mM $MgCl_2$, 5.0 mM HEPES (pH 7.4) to concentrations of 20, 39, 78, 156, 312, 625, 1250, 2500, 5000, and 10000 nM. The prepared solutions were injected over the surfaces; the contact time ($t_c$) was 180 s, and the following dissociation phase time was 400 s. The surfaces were then regenerated in 50 mM EDTA, 1.5 M NaCl, and 0.1 M HEPES (pH 7.49. The data collection rate was one data point per 0.4 s. For data analysis, the two-dimensional fits were made on the MATLAB 2021a platform (Mathworks) using the fitting tool EVILFIT version 3 software[43]. In brief, input values matched the start and end of injection times. They included sensorgrams for concentrations from 156 nM to 10,000 nM for both I domains. To avoid noise in the data, the association phase was fitted from t = "injection start" plus 1 s to t = "injection end" minus 2–10 s. The dissociation phase was fitted from t = "injection end" plus 7 s to t = "injection end" plus 200–350 s. The operator-set boundaries for the distributions were uniformly set to limit the dissociation equilibrium constant ($K_D$) values in the interval from $10^{-12}$ to $10^{-2}$ M and the dissociation rate ($k_{off}$) in the interval from $10^{-6}$ to $10^0 s^{-1}$ to ensure comparable, and best quality of fits, reflected in a high signal to root-mean-square deviation (RMSD) ratio. The distribution P ($k_d$, $K_D$) was calculated using the discretization of the equation:

$$S_{Total} = \int_{K_{D\min}}^{K_{D\max}} \int_{k_{off\min}}^{k_{off\max}} S\left(k_{off}, K_D, c_{analyte}, t\right) P\left(k_{off}, K_D\right) dk_{off}\, dK_D$$

(1)

where $S$ is the SPR signal. A logarithmic grid of ($k_{off, i}$, $K_{D, i}$) values are distributed on each axis with $18 \times 18$ grid points. This was done through a global fit to the association and dissociation traces at the analyte concentrations ($c_{analyte}$) mentioned above. The algorithm used Tikhonov regularization as described[44] at a confidence level of $p = 0.95$ to determine the most parsimonious distribution consistent with the data, showing only features essential to fit the data.

### Flow cytometry of monocytes with assessment of CR4 phagocytic inhibition

Human primary monocytes, purified and stored as described[29], were thawed in PBS (pH 7.4) supplemented with 20% (v/v) FCS from −135 °C on the day of the experiment and kept on dry ice until use. Cells were resuspended in 150 mM NaCl, 1 mM $MgCl_2$, 1.8 mM $CaCl_2$, 5 mM glucose, 0.1 mg/mL HSA, HEPES 10 mM (pH 7.4) at a concentration of $1 \times 10^6$ cells/mL. Wells in a microtiter plate (cat. no. 82.1583 Sarstedt) received 100 μL of cell suspension (100,000 cells/well). To experiments incubated with mouse isotype IgG (cat. no. M5284, lot.no. 068M4856V), 3.75 μL of Ab solution (0.10 mg/mL) was added. To samples with KIM127

(GeneScript),3.75 µL of 0.5 mg/mL Ab was added. To samples with anti CR4 (clone 3.9) 1.5 µL of 0.25 mg/mL Ab was added. Samples were then incubated for 30 min at 37 °C. Monomeric or aggregated HiLyte 488 Aβ (Anaspec) was then added to a final concentration of 5 µg/mL. Samples were incubated for 60 min at 37 °C with 5% $CO_2$. Samples were then centrifuged for 5 min at $300 \times g$, 4 °C, and the supernatant was discarded. For flow cytometric cell staining, an Ab master mix was used consisting of 2.5 µL anti-CD11b-BV421 (cat. no. 742637, lot.no. 3114956, BD Optibuild), 1 µL 1:200 diluted in PBS (pH 7.4) LysoTracker Red DND-99 (no. L7528; Invitrogen), 10 µL anti-CD11c-PE (cat. no. 333149, lot 34546, BD) and 36.5 µL flow media (PBS/1% [w/v] BSA, 0.5 mM EDTA). Blank and single stain controls were used to set MFI-positive boundaries. 50 µL master mix was added to each sample, followed by incubation for 30 min at 4 °C. Next, 170 µL flow media was added to stop staining. Samples were centrifuged at $300 \times g$ for 5 min and 150 µL flow media was added before Flow analysis using a Novocyte Quanteon Flow Cytometer (Agilent).

## Monocytic phagocytosis of biotinylated Aβ

Human primary monocytes, purified and stored as described[29], were thawed in PBS (pH 7.4) supplemented with 20% (v/v) FCS from −135 °C on the day of the experiment and kept on dry ice until use. Biotinylated monomeric and aggregated Aβ samples were preincubated with 20 nM streptavidin/QD solution (cat.no. Q10123MP, Molecular Probes) for 30 min at 37 °C; the final concentrations of proteins were 20 µg/mL. Cells were resuspended in 1 mL RPMI to a concentration of $40 \times 10^6$ cells/mL. $1 \times 10^6$ cells were added to the Aβ/QD suspensions or QDs without protein (control) together with 5 µg/mL CD18 integrin-activating Ab KIM127 diluted from a stock of 1 mg/ml (GeneScript), followed by incubation for 30 min at 37 °C in 5% $CO_2$. Samples were then washed twice with PBS pH 7.4. After centrifugation at $230 \times g$ for 5 min, the supernatant was saved for NTA and stored at −20 °C. NTA was used to measure phagocytosis by counting aggregated Aβ particles before and after incubation with monocytes. were thawed and then diluted in PBS (1:10) to obtain a particle concentration suitable for analysis. Samples were then analyzed as previously described[29].

## iPSC culture and microglia differentiation

For microglial differentiation, the human induced pluripotent stem cell (iPSC) line BIHi043A-XM001 was used[56]. Cells were cultured on Matrigel™ (cat. no. 356234, Corning)-coated culture plates in Essential 8™ Flex Medium (cat. no. A2858501, Gibco). The culture medium was changed daily, and cells were passaged in clusters every 3–4 days at a density of 80% using 0.5 mM EDTA/PBS. iPSCs were differentiated into microglia using the protocol described[72]. First, iPSCs were differentiated into hematopoietic progenitors (HP) using STEMdiff™ Hematopoietic Kit (cat. No. 05310, Stem Cell Technologies). On Day −1, iPSCs with a 70–80% density were passaged with ReLeSR™ (cat. no. 05872, Stem Cell Technologies) into E8 flex containing Matrigel-coated 6-well plates. Clusters of 100 cells were seeded at the desired density of 50–100 per well. At Day 0, in wells with 40–80 clusters, the E8 flex medium was replaced with 2 mL medium A. On Day 2, 1 mL of medium A was added to the well. On Day 3, the medium was removed and replaced with 2 mL Medium B. On Days 5, 7, and 9, 1 mL of medium B was added to the well. On Day 11, a high density of round bright, floating hematopoietic progenitors was present in the media and attached to the bottom. The adherent cells were gently washed off to increase the yield using a 5 mL serological pipette. The cells were centrifuged at $300 \times g$ for 5 min, resuspended in microglia differentiation medium (DMEM/F12, 2× insulin-transferrin-selenite, 2× B27, 0.5× N2, 1× Glutamax™, 1× non-essential amino acids, 400 µM mono-thioglycerol, 5 µg/mL insulin, 100 ng/mL IL-34, 50 ng/mL TGFβ1, and 25 ng/mL M-CSF) and seeded at a density of 100.000-200.000 cells in Matrigel (cat. no. 354277, Corning)-coated 6-well plates (Corning). On Days 13, 15, 17, 19, and 21, 1 mL of microglia differentiation medium was

added to the well. On Day 23, except for 1 mL, the medium was transferred to a 15-mL tube to spin down the floating cells at $300 \times g$ for 5 min. The cells were resuspended in 1 mL microglia differentiation medium and returned to the well. This was repeated on Days 25–35. On Day 35, the cells were resuspended in a microglia differentiation medium with 100 ng/mL CD200 and 100 ng/mL CX3CL1 to further mature the microglia. On Day 37, 1 mL of the maturation medium was added to the well. On Days 38–42, the microglia were ready for functional studies.

## Immunostaining and Confocal Microscopy

Mature microglia were seeded on non-coated 12 mm glass coverslips, at a density of 50,000 cells pr. coverslips, and allowed to attach overnight. Next day, the cells were washed once in PBS and fixed with 4% (v/v) PFA in PBS for 15 min at RT. The fixed cells were washed twice in PBS and stored in 0.02% (w/v) $NaN_3$ in PBS at 4 °C. For the immunostaining, the cells were incubated with blocking buffer (5% Normal donkey serum, 0.01% [v/v] Triton X-100 in PBS) for 30 min at RT, followed by ON incubation with primary antibody in blocking buffer at 4 °C. Next day, cells were washed in PBS and incubated with secondary antibody in blocking buffer for 2 h at RT. The cells were washed in PBS, incubated with DAPI (cat. no. 10236276001, Roche) diluted 1:3000 for 5 min, and mounted on SuperFrost Ultra Plus Menzel glass slides (Thermo Fischer) in fluorescent mounting medium (cat.no. S3023, DAKO). The coverslip was sealed with transparent nail polish before imaging. The images were acquired on a Zeiss LSM 980 confocal microscope and analyzed by ImageJ (NIH). The primary antibodies used were goat anti-Iba1 (1:500, ab5076, Abcam) or rabbit anti-P2ry12 (1:100, GTX54796, Genetex). Secondary antibodies used were donkey anti-goat Ab (1:500, Alexa Fluor 555, A32816, Thermo Fischer) or donkey anti-rabbit (1:500, Alexa Fluor 488, 711-545-152, Jackson ImmunoResearch).

## FACS of iPSC-derived microglia

iPSC-derived microglia were harvested on the day of the experiment and resuspended in microglia maturation media as single cells at a concentration of $2 \times 10^5$ cells/mL and 250 µL was added to 1.5 mL tubes (50,000 cells/tube). For samples with KIM127 Ab (GeneScript) activation, 1.25 µL of Ab at a concentration of 1.0 mg/mL was added. As a control, 2.5 µL of 0.1 mg/mL mouse isotypic IgG1 (cat. no. X0931, clone "DAK-GO1", lot.no. 20047017, Agilent) was added to the samples. Next, samples were incubated for 30 min at 37 °C. 2.5 µL of 0.5 mg/mL monomeric or aggregated HiLyte 488 Aβ (Anaspec) was then added to a final 5 µg/mL concentration. Samples were incubated for 60 min at 37 °C with 5% $CO_2$. For staining, an Ab master mix was used consisting of 1 µL anti-CD11b-PE (clone "M1/70", cat. no. 101207; lot.no. B357278, Biolegend), 1 µL LIVE/DEAD Fixable Violet Dead Cell (no. L34955; Invitrogen), 1 µL 1:200 diluted in PBS pH 7.4 LysoTracker Red DND-99 (no. L7528; Invitrogen), 2 µL anti-CD11c-PerCP-eFluor 710 (clone "3.9", cat. no. 46-0116-41, lot.no. 2194305, Invitrogen) and 45 µL flow media (PBS/1% [w/v] BSA, 0.5 mM EDTA). Blank and single stain controls were used to set MFI-positive boundaries. 50 µL master mix was added to each sample, followed by incubation for 30 min at 4 °C. Next, 400 µL flow media was added to stop staining. Samples were centrifuged at $230 \times g$ for 5 min and washed twice in flow media before FACS sorting using a BD FACS Aria Fusion Flow Cytometer (Becton, Dickinson, and Company). Untreated cells were collected together, whereas cells treated with Aβ were sorted into high and low Aβ-uptake fractions. Cell lysis and RNA extraction were performed using RNeasy Micro plus kit (cat. no. 74034, Qiagen). cDNA synthesis was performed using High-Capacity RNA-to-cDNA™ Kit (cat. no. 4387406, Applied Biosystems). Taqman qPCR was performed on a QuantStudio 7 Pro Real-Time PCR machine (Applied Biosystems) using TaqMan™ master mix for gene expression (cat. no. 4369016) and the following TaqMan® Assays: *ITGAX* (CD11c; Hs00174217_m1), *ITGAM* (CD11b, Hs00167304), and

*GAPDH* (Hs02758991). *GAPDH* was used as a reference gene for the quantitative analysis with data presented as 2-delta, delta ct fold change relative to untreated cells.

## Flow cytometry of monocytes and iPSC-derived microglia with assessment of CR4 phagocytic inhibition

iPSC-derived microglia were harvested on the day of the experiment and resuspended in microglia maturation media as single cells at a concentration of $1 \times 10^6$ cells/mL and 40 μL was added to a microwell plate (cat. no. 82.1583 Sarstedt) (40,000 cells/well). To samples with mouse isotype IgG (cat. no. M5284, lot.no. 068M4856V), 3.75 μL of 0.04 mg/mL Ab was added. To samples with KIM127 (GeneScript), 3.75 μL of 0.2 mg/mL Ab was added. To samples with anti CR4 (clone 3.9) 1.5 μL of 0.1 mg/mL Ab was added. Samples were then incubated for 30 min at 37 °C. Monomeric or aggregated HiLyte 488 Aβ (Anaspec) was then added to a final concentration of 5 μg/m. Samples were incubated for 60 min at 37 °C with 5% $CO_2$. Samples were then centrifuged for 5 min at $300 \times g$, 4 °C and the supernatant was discarded. For staining, an Ab master mix was used consisting of 2.5 μL anti-CD11b-BV421 (cat. no. 742637, lot.no. 3114956, BD Optibuild), 1 μL 1:200 diluted in PBS pH 7.4 LysoTracker Red DND-99 (no. L7528; Invitrogen), 10 μL anti-CD11c-PE (cat. no. 333149, lot 34546, BD) and 36.5 μL flow media (PBS/1% [w/v] BSA, 0.5 mM EDTA). Blank and single stain controls were used to set MFI-positive boundaries. 50 μL master mix was added to each sample, followed by incubation for 30 min at 4 °C. Next, 170 μL flow media was added to stop staining. Samples were centrifuged at $300 \times g$ for 5 min and 150 μL flow media was added before Flow analysis using a Novocyte Quanteon Flow Cytometer (Agilent).

## Statistics and reproducibility

All statistical analyses were performed using Prism software (GraphPad Software, San Diego, CA). QQ plots were used for testing normality before all parametric tests and ANOVA with adjustment for multiple comparisons were used when comparing multiple groups. A $p < 0.05$ was considered to be statistically significant. All information on experimental replicates, the number of donors tested, and applied statistical tests are stated in figure legends. No statistical method was used to predetermine sample size and no data were excluded from the analyses. The experiments were not randomized. The Investigators were not blinded to allocation during experiments and outcome assessment.

## Reporting summary

Further information on research design is available in the Nature Portfolio Reporting Summary linked to this article.

## Data availability

All data generated in this study are provided in the Supplementary Information and Source Data files. Video recordings for NTA analysis and original files for flow cytometry are available upon request. Source data are provided with this paper.

## Code availability

Codes used for the analysis of the SEM images (in Fig. 1) and hydrodynamic calculations (in Fig. 4) are deposited in the GitHub repository at https://github.com/VingesLDB/2023_NCOMM.

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

## Acknowledgements

We thank Prof. Per Borghammer for advice and Bettina W. Grumsen for excellent technical assistance. We also thank Søren Peder Madsen for maintaining and assisting with optimal usage of the Aarhus University computing cluster "PRIME" and the FACS CORE facility in the Department of Biomedicine for help with all flow cytometric analyses. Funding for this study was received from Aarhus University Research Foundation grant AUFF-E-2015-FLS-9-6 (K.J.-M., T.V.-J.); Aarhus University Research Foundation AU IDEAS 2013 grant NEURODIN (M.R.R., T.V.-J. & D.J.B.); The Danish Alzheimer Association (L.S.M.); Novo Nordisk Foundation grant NNF19OC0058516 (K.J.-M., T.V.-J.); Novo Nordisk Foundation grant NNF21OC0071574 (D.S.S.); Lundbeckfonden grant R380-2021-1326 (K.J.-M., T.E.W., T.V.-J.); Lundbeckfonden (D.J.B.); Independent Research Fund Denmark grant 2030-00002B (K.J.-M., T.E.W., T.V.-J.); Independent Research Fund Denmark grant 3101-00104B (T.V.-J.); Independent Research Fund Denmark grant 4004-00305 (S.F.E.); Danish National Research Foundation. center grant CellPAT - DNRF135 (D.S.S.); EU Horizon 2020 – Fast Track to Innovation, grant 820636 (L.S.M.); This research was supported in part by the Intramural Research Program of the NIH, NIBIB (P.S., Z.H.).

## Author contributions

KJ-M, PP, RI, LSM, SFE, DJB, and TV-J conceptualized the study. KJ-M, PP, RI, PLO, VS, LSM, JT, AL, MNA, MRR, CKH, GRA, HZ, PS, JVN, DSS, SFE, TEW, DJB, and TV-J developed the methodology. KJ-M, PLO, VS, LSM, JT, SG, MB, MNA, GRA, HZ, PS, JVN, DSS, SFE, and TV-J made the investigations. KJ-M, LSM, JVN, DSS, SFE, and TV-J designed the visualization of data. KJ-M, MRR, TEW, DJB, and TV-J secured funding acquisition. TV-J undertook project administration. KJ-M and TV-J performed project-related supervision and wrote the original draft of the manuscript. KJ-M, PP, RI, PLO, LSM, JT, SG, MB, AL, MNA, MRR, CKH, GRA, SFE, HZ, PS, JVN, DSS, TEW, DJB, and TV-J reviewed and edited the manuscript.

## Competing interests

The authors declare no competing interests.

## Additional information

[1]Department of Biomedicine, Aarhus University, The Skou Building, Høegh-Guldbergs Gade 10, DK-8000 Aarhus C, Denmark. [2]Max-Delbrueck-Center for Molecular Medicine, Robert-Rössle-Str. 10, 13125 Berlin, Germany. [3]Department of Nuclear Medicine, Odense University Hospital, J. B. Winsløws Vej 4, DK-5000 Odense C, Denmark. [4]Department of Nuclear medicine and PET, Vejle Hospital, Beriderbakken 4, DK-7100 Vejle, Denmark. [5]Department of Nuclear Medicine & PET Centre, Aarhus University Hospital, Palle Juul-Jensens Boulevard 99, DK-8200 Aarhus N, Denmark. [6]Department of Clinical Medicine, Aarhus University, Palle Juul-Jensens Boulevard 11, DK-8200 Aarhus N, Denmark. [7]Center of Functionally Integrative Neuroscience, Aarhus University and Aarhus University Hospital, Building 1710, Universitetsbyen 3, DK-8200 Aarhus C, Denmark. [8]Department of Hematology, Aarhus University Hospital, Palle Juul-Jensens Boulevard 99, DK-8200 Aarhus N, Denmark. [9]NEURODIN AU IDEAS Center, Department of Biomedicine, Aarhus University, The Skou Building, Høegh-Guldbergs Gade 10, DK-8200 Aarhus C, Denmark. [10]Department of Molecular Biology and Genetics, Aarhus University, Universitetsbyen 81, DK-8000 Aarhus C, Denmark. [11]Laboratory of Dynamics and Macromolecular Assembly, National Institute of Biomedical Imaging and Bioengineering, Building 31, 9000 Rockville Pike, Bethesda, MD 20892, USA. [12]Department of Biological and Chemical Engineering, Aarhus University, Gustav Wieds vej 10 D, DK-8200 Aarhus C, Denmark. [13]Interdisiciplinary Nanoscience Center, Aarhus University, The iNANO House, Gustav Wieds Vej 14, DK-8200 Aarhus C, Denmark. [14]Center for Cellular Signal Patterns, Aarhus University, The iNANO House, Gustav Wieds Vej 14, DK-8200 Aarhus C, Denmark. [15]Department of Brain Sciences, Imperial College London, Burlington Danes, The Hammersmith Hospital, Du Cane Road, London W12 0NN, UK. [16]Institute of Translational and Clinical Research, University of Newcastle, Framlington Place, Newcastle upon Tyne NE2 4HH, UK. ✉e-mail: vorup-jensen@biomed.au.dk

