## [Peer Review File · Nature Communications]

Amyloid- β aggregates activate peripheral monocytes in mild cognitive impairmentReviewers' Comments:

Reviewer #1 (Remarks to the Author):

The manuscript by Juul-Madsen and colleagues studies interactions between monocytes and amyloid beta (abeta) aggregates. They report to have developed an assay which characterizes the size and quantity of amyloid beta aggregates found in the blood plasma of patients with Alzheimer's disease (AD), although the experiments are mainly done in mild cognitive impairment (MCI) samples. This work is an extension of what the group of Vorup-Jensen has published studying interactions between alpha synuclein aggregates and CR4 as well as DNA-protein complexes and B cells (doi: 10.4049/jimmunol.1900494 and doi: 10.1073/pnas.2106647118). The authors combined a commercially available system for nanoparticle tracking analysis (Nanosight) with protein aggregate labelling using a quantum dot-conjugated antibody against abeta (Aducanumab). They further address the role of the peripheral immune system in MCI by reporting the enrichment of intermediate and non-classical monocytes in MCI patients whose plasma abeta levels are increased. They also identify complement receptor 4 (CR4) as a size-selective binding partner of abeta aggregates which they report to be disease-promoting.

Major concerns:

The entire study relies on particle tracking and sizing, obtained by calculation of diffusion coefficients. The analysis would make sense for globular particles, but in the case of fibrils the diffusion is highly anisotropic. There is no information whatsoever regarding the way the sizes were calculated (eg "automatic settings were used for the minimum particle size", or "The detection threshold was set between 2 and 5"...), which makes the starting point of the entire study very weak and completely irreproducible by other laboratories. Assuming that the authors did calculate the size correctly, it should be validated with an orthogonal method (eg comparing sizes of synthetic fibrils obtained with NTA and obtained with either SEM, TEM or super-resolution imaging). The size validation should subsequently be applied to plasma spiked with abeta (i.e. repeat Fig 1D with an orthogonal technique). The labeling specificity should also be validated in biofluids spiked with Abeta (maybe using a fluorescently labeled abeta to co-localise with QD?), as Fig 1C seems to be done only in buffer.

Sizes should be contrasted to the amyloidogenic aggregates reported by other laboratories. In healthy controls and Parkinson's serum and CSF, single-molecule imaging reports that the vast majority of amyloidogenic aggregates are between 40nm and 150nm in size (doi: 10.1093/brain/awab306); while the typical aggregate size of AD and MCI patients in CSF has been reported to be also between 50nm and 150nm (doi: 10.1186/s40478-019-0777-4). The authors should demonstrate that the 600nm threshold (as calculated with NTA) effectively corresponds to 600nm measured with an alternative technique.

Some statistical methods require more robust tests since many of the effects reported as very robust are not particularly well justified by p-values and sample sizes (e.g. Figures 2 and 4). A statistics editor could double check this.

The mechanistic model which summarizes the study is not entirely supported by the reported data. There's no proof about monocytes entering the parenchyma. Most of the reported activity of peripheral immune cells has been reported at the level of the meninges (doi: 10.1016/j.cell.2020.12.040).

Minor concerns:

However the introduction would benefit from improved clarity, language, and overall flow. The part explaining the connection between MCI and AD needs to be rewritten for clarity and accuracy (lines 17-24 of page 4). It would be good to include more information about why it is important to look at different abeta aggregate sizes.

Methods: More experimental detail is required if these results are to be replicated. It needs a thorough rewrite.

Results: Statistics is not reported consistently throughout the manuscript. Formatting is also inconsistent between the figure legends.

Figure 1: The two panels of Figure 1E would be more informative if they are merged into one graph; Figure 1F could be more informative in log scale

Figure 2: Please report correlation coefficients and the strength of the correlation

Figure 3: It is not explained why they added MnCl to the probes in Figure 2B

Figure 4: Please use an ANOVA when you compare more than 2 groups within the same graph and adjust the strength of your posthoc tests. Can you also show the difference between control and A(low) and M(low) in Figure 4K.

Figure 6: Although the figure aims to interpret the results in a mechanistic context, the figure is not an accurate representation of the data provided. The authors claim trans-endothelial migration of monocytes in the brain and there is no direct evidence of such in this article. Also, the figure claims that sCD18 is increased in MCI patients positive for plasma aggregates. However Figure 2I shows that sCD18 levels of these patients are no different from controls. Figure 4 does not support a difference in the microglial response to monomeric and aggregated abeta. The y-axis could have a more accurate label (is "NTA concentration" a thing?).

Discussion: The claims are too strong and not well supported by data.

The formatting is not consistent across figures – including abbreviations (e.g. aggregate: Agg or A?) Referring to the aggregates as "nanoscale", "nanosize" or "nanoparticulate" is not benefitting the description of this work.

Reviewer #2 (Remarks to the Author):

In this paper, the authors analyze levels of nanoscale amyloid aggregates in patients using a novel technique, and they show that the subset of MCI patients with detectable large amyloid-beta aggregates have expanded populations of CD16+ monocytes (intermediate/non-classical), whereas MCI patients without amyloid-beta aggregates had expanded CD16- (classical) monocytes. Monocyte CD18 activation was correlated with levels of large amyloid-beta aggregates, and in vitro assays demonstrated that CR4 (but not CR3), which is a CD18-containing integrin receptor, preferentially binds aggregated amyloid-beta over monomeric amyloid-beta. Thus the CD18-activated monocytes in MCI patients with large amyloid-beta aggregates may be a result of direct interaction between monocyte CR4 and aggregated amyloid-beta. The authors then ask the question whether CR4 is an important receptor for phagocytic uptake of amyloid-beta aggregates by monocytes/other myeloid cells, which is the section that I will focus on, as an iPSC-microglia specialist. The subsection is titled "Phagocytosis of A β in human monocytes and induced pluripotent stem cell-derived microglia is mediated by CR4".

Overall, the methodology for generating iPSC-microglia seems sound, and some very basic validation of the cells identity has been performed using ICC for Iba1 (see Minor Comment 6). The flow cytometry methods appear sound, although the placement of gates is not very transparent (see Minor Comment 7). Amyloid-beta preparations appear to have been well-characterized (Supplementary figure S2). The figures are beautifully presented and indicate statistical significance clearly. However, I have concerns about how these results have been reported in the main body of the text, there are a number of discrepancies that I will detail in the Major Comments below. The data does not support the conclusions for any of the subsection titled "Phagocytosis of A β in human monocytes and induced pluripotent stem cell-derived microglia is mediated by CR4", and most importantly do not provide any evidence that CR4 is directly involved with phagocytic clearance of A β aggregates, which seems to have been the purpose of these investigations.

Major Comments:

1. In page10/line1 the authors state "KIM127 activation increased internalization of aggregated A β ". This is very misleading as Fig4C clearly shows that there is no statistically-significant difference between internalization of aggregated amyloid-beta +/- KIM127 (p value is 0.09, significance is conventionally considered to be below 0.05). This negative result is important and needs to be reported properly. This also means that the subtitle of this part of the Results - "Phagocytosis of A β in human monocytes and induced pluripotent stem cell-derived microglia is mediated by CR4" - is not supported by evidence. The authors have not demonstrated that CR4 mediates phagocytosis of amyloid-beta (aggregates) in human monocytes. I recommend that the subsection title is modified to reflect this.
2. In Figure 4H, iPSC-microglia are incubated with monomeric and aggregated amyloid-beta-42, and flow cytometry performed to evaluate the CR4 expression and amyloid-beta uptake of single cells. The authors selected a subset of cells with higher CR4 expression and showed that higher CR4 expression is associated with significantly more aggregated amyloid-beta uptake than monomeric amyloid-beta. The evidence does support a potential correlation between CR4 expression and aggregated amyloid-beta uptake, but is not sufficient to support the claim that phagocytosis of amyloid-beta is mediated by CR4 in iPSC-microglia (in the subsection title, see Major Comment 1). To evidence this the authors would need to do experiments with CR4 knockout or knockdown. I recommend that the subsection title is modified to avoid claiming that that CR4 mediates phagocytosis of amyloid-beta (aggregates) in human iPSC-

microglia.

3. In page10/line20 the authors state “We found increased CR4 transcription only in iMGs treated with aggregated A β and showing a low A β uptake, while CR4 levels were unchanged in cells with high A β uptake (Fig. 4K)”. In Figure 4K the statistically significant difference in CR4 expression is between iPSC-microglia with high A β uptake versus low A β uptake, however there is no significant difference between high A β uptake versus no A β addition. This should be stated clearly in the main text, since this is the relevant comparison. There is insufficient evidence to support the statement in page11/line1: “These findings suggest a functional coupling of aggregated A β phagocytosis and CR4 expression in iMGs”, or the statement in page12/line17 “our data showing that A β aggregates increase the CR4 expression in human iPSC-derived microglia”. I recommend that these are re-phrased or removed.

4. The Abstract states “[CR4] binds size-selectively to nanoscale amyloid beta and promotes a disease-modifying phenotype also found in human induced pluripotent stem cell-derived microglia”. There is no evidence of a disease-modifying phenotype in the iPSC-microglia, what data supports this claim?

Minor comments:

1. The authors do not explain why adding fluorescent amyloid-beta to primary monocytes and activating the cells with KIM127 is supposed to illuminate the role of CR4. KIM127 is previously described (see page 7 line 19) as a conformation-specific antibody for CD18 that can be used to probe levels of CD18 activation. KIM127 is not explained to be a CD18-activating antibody, although we can infer this indirectly from the result in Fig 3E where addition of KIM127 to monocytes improves their adhesion to aggregated amyloid-beta. It would be helpful if this were spelled out in page 9-line 20, and also it should be acknowledged that both CR3 and CR4 contain CD18, so an assumption has been made that KIM127 activates CR4 (presumably based on the in vitro assays in Figure 2).

2. The authors state in page10/line12 that they used iPSC-microglia to investigate “how CR4 amyloid recognition by microglial cells contributes to the signaling cascade”. Firstly, the signalling cascade of what, and how does this relate to the subsection title? The authors need to be clearer about what they are investigating.

3. There is a typographical error at page10/line22: “This difference was seen for iMGs with low and high uptake of monomeric A β ”, should actually be “This difference was not seen for iMGs with low and high uptake of monomeric A β ”.

4. The data on TREM2 expression in monocytes and iPSC-microglia is completely superfluous for the conclusions of the paper, and does not appear to add anything to the story. The proposed interaction between CD18 activation and TREM2 transcription is not convincing, as the error bars in Fig 4E are very large (for monomeric or aggregated A β + K127), clearly there were 2-3 experimental repeats where no induction of TREM2 transcription occurred. Furthermore the finding in Fig 4L that both monomeric and aggregated A β caused an induction of TREM2 transcription actually indicates that CD18 activation is unlikely to have a major contribution to TREM2 transcription, since CR4 was shown to preferentially bind aggregated A β .

5. Supplementary figure S8- the title of this figure is “NTA detection of A β QD-conjugates from monocyte phagocytosis shows that CD18 activation promotes phagocytosis of aggregated A β ”. However, there are no statistically significant difference and the data is extremely noisy, so this statement is not supported by data shown.

6. Supplementary figure S10- the only validation of cell identity for the iPSC-microglia appears to be immunocytochemistry of Iba1. Iba1 is expressed highly by macrophages as well as microglia. A better

microglial identity marker would be TMEM119.

7. Supplementary figure S11- in S11a and S11b the data that was used for setting the gates should be shown, ideally single-stained controls. Otherwise we are expected to take it on trust that the gates were placed correctly.

Reviewer #3 (Remarks to the Author):

The authors study the relationship of soluble amyloid beta aggregates in the plasma of patients with mild cognitive impairment with CNS measures of brain inflammation. They find that about a third of patients with MCI have circulating Abeta aggregates >600 nm and this relates in creased number of non-classical monocytes and lower CNS inflammation. The also find that a receptor for Ab on monocytes is CR4. This is an interesting study, but there are some questions.

1. The sample size is borderline for the conclusion given that there are multiple comparisons and the best p values are at <0.01 at 10 samples with aggregates.
2. In the analysis in Figure 2, it would be helpful to see data for CD11c and CD11b expression in addition to CD8 and KIM127, w which to an extension dependent epitope on CR3 and CR4. The correlation of ab with KIM127 positivity is helpful.
3. The SPR data for CR4 is characteristically hard to interpret. Compared to the relatively well behaved, but weak, binding for CR3, the response for the CR4 I domain is very weak an noisy. The adhesion result is supportive of CR4 being the receptor, but the SPR result suggests that there is so few CD4 I domain capable to binding Abeta as to be hard to interpret or trust.
4. The idea that Abeta binding to CR4 upregulates CR4 transcription is interesting, but I would not have expected it. Is this well known for iC3b and CR4 expression?
5. The statistical differences are marginal at $p < 0.05$. It has been discussed that this is not really an acceptable standard as it means that 5% of published studies will report erroneous data under the best of circumstances. Can a power analysis be done and the sample size increased to reach $p < 0.005$? Or this issue should be discussed as a marginal call between a trend and significant.

Reviewer #4 (Remarks to the Author):

As per Editor request, I will only review the positron emission tomography (PET) aspects of this study. Overall, the PET quantification methods are poorly described, some results appear to be driven by errors in the quantification pipeline, and the details of the brain-wide statistical analysis are vaguely reported.

1) In Figure 5A and 5B (left and mid panels), the authors report “Mean overall cortical 18F-FTP PET (or 11C-PiB) uptake in each group at baseline, two-year follow-up, and longitudinal change.” The reported mean SUVR values, however, are in the range of 10^9 , a value that is clearly not possible for an average SUVR (doi: 10.1093/brain/awz268). What are they actually reporting in Figure 5A and 5B?

- 2) In Figure 5A and 5B (left and mid panels), there are 3-4 cases in the MCI Agg[-] group with extremely low SUVR values. Again, we don't see cases with such low SUVR values (doi: 10.1093/brain/awz268; doi: 10.1093/brain/aww334). These cases seem to be the result of errors in the quantification pipeline, probably due to errors in the segmentation or coregistration processes.
- 3) In Figure 5A and 5B (right panel), there are subjects with extremely large SUVR changes after two years. Amyloid and tau accumulation are very slow processes, typically 0.1-0.2 SUVR units/year (doi: 10.1093/brain/awy059; doi: 10.1001/jamaneurol.2019.1424). This, again, suggests that there have been errors with the PET quantification pipeline.
- 4) The authors should be clearer with their approach of measuring PET signal in the overall cortex. This approach is certainly less efficient to detect actual accumulation, as there are several regions in the cortex that do not show elevated amyloid or tau-PET signal. Why not using established regions of interest for measuring amyloid and tau (doi: 10.1016/j.jalz.2016.08.005)?
- 5) The authors only vaguely describe the voxel (or vertex)-wise statistical threshold for their cortical maps. Based on Figure 5C-D, I assume they apply an uncorrected $p < 0.05$. However, they also state in Fig. 5's legend that they corrected for multiple comparisons, but they do not describe which method (cluster, FWE, FDR...) they used.
- 6) There are no details on how PET images were reconstructed, in particular whether the acquisition was static or dynamic (and then averaged).
- 7) The authors do not report the software they used to segment the MRI scans, to quantify the PET scans, and to map SUVR values to the cortical surface.
- 8) SUVR stands for standardized uptake value ratio.
- 9) The authors state that mapping SUVR values to the cortical surface "ensures that the PET signal only originates from cortical gray matter." This is not accurate. It is true that measuring SUVR values in the midsection of the cortical mantle can reduce spill-out effects. However, spill-in signal from other regions (e.g. from the white matter) can also occur, and mapping SUVR values to the cortex will not solve this issue.
- 10) It would be useful to categorize MCI patients as amyloid-positive or negative using established cut-points for e.g. Centiloids. Presence of amyloid positivity is strongly associated with tau pathology and overall AD pathology. It would be interesting to see how traditional amyloid positivity correlates with Aggregate-positivity.

Alexis Moscoso Rial

Point-by-point reply to reviewers' comments (NCOMMS-22-53102-T; "Nanoscale amyloid beta activates peripheral monocytes and protects against neuropathology," now entitled "Large amyloid beta aggregates activate peripheral monocytes and protect against neuropathology")

Below, we have outlined the several changes made to our manuscript, all closely following the recommendations by the Reviewers.

All parts of the manuscript have been significantly changed, and the data presentation has been reordered to improve the paper's flow. In the included manuscript, yellow highlight was used to indicate changes of particular relevance for the point-by-point reply and to help the reading of these replies. Figures references with capital letters (e.g., Fig. 1A) refer to the previously submitted paper. In contrast, references with lowercase letters (e.g., Fig. 1a) refer to the present paper.

We sincerely thank the Reviewers for their critical input and efforts to improve our paper.

Reviewer #1 (Remarks to the Author):

The manuscript by Juul-Madsen and colleagues studies interactions between monocytes and amyloid beta (abeta) aggregates. They report to have developed an assay which characterizes the size and quantity of amyloid beta aggregates found in the blood plasma of patients with Alzheimer's disease (AD), although the experiments are mainly done in mild cognitive impairment (MCI) samples. This work is an extension of what the group of Vorup-Jensen has published studying interactions between alpha synuclein aggregates and CR4 as well as DNA-protein complexes and B cells (doi: 10.4049/jimmunol.1900494 and doi: 10.1073/pnas.2106647118). The authors combined a commercially available system for nanoparticle tracking analysis (Nanosight) with protein aggregate labelling using a quantum dot-conjugated antibody against abeta (Aducanumab). They further address the role of the peripheral immune system in MCI by reporting the enrichment of intermediate and non-classical monocytes in MCI patients whose plasma abeta levels are increased. They also identify complement receptor 4 (CR4) as a size-selective binding partner of abeta aggregates which they report to be disease-promoting.

Author reply R1.1: The Reviewer has made an excellent overview of critical points and our study's background. However, we would like to emphasize that the impact of the A β aggregates is not only to promote disease but, at least in our model, also to recruit monocytes to the brain for clearance of AD-like pathology. We understand that it was not clear in our previous manuscript. Descriptions explaining better this part of our data have now been included in the Discussion and the (new) schematic Fig. 8.

Major concerns:

The entire study relies on particle tracking and sizing, obtained by calculation of diffusion coefficients. The analysis would make sense for globular particles, but in the case of fibrils the diffusion is highly anisotropic. There is no information whatsoever regarding the way the sizes were calculated (eg "automatic settings were used for the minimum particle size", or "The detection threshold was set between 2 and 5" ...), which makes the starting point of the entire study very weak and completely irreproducible by other laboratories.

Author reply R1.2: Nanoparticles tracking analysis (NTA) uses the well-known Stokes-Einstein Equation to calculate the hydrodynamic radius (or, in our study, diameter) of a particle from the experimentally found diffusion coefficient¹, determined from the recorded frame-by-frame distance displacement of particles. Others have already used NTA for analysis of highly anisotropic particles, including aggregated A β ^{2,3} and α -synuclein⁴. In this way, our paper does not differ from existing standards in the field. Of course, we agree with the Reviewer that the obtained hydrodynamic radius cannot describe all properties of the particles. Usually, for anisotropic particles, NTA seems to capture well, but not only, the long axis, which is also our conclusion from using some of the orthogonal techniques suggested by the Reviewer (see below in Author Reply R1.3).

Another way to understand our reported values is that they indicate diameters of perfectly spherical particles, which would diffuse/move similarly to the tracked particles. This is useful in at least two ways. First, since the calculation of these diameters from the experimental data is made the same way for all samples, it permits a comparison between samples. Second, we can now use this for hydrodynamic calculations with the apparent (or effective) hydrodynamic size (D_H) established. For

such a purpose, experimental determination of the geometric size of particles would require the subsequent calculation of D_H . By contrast, we can model the movement of the $A\beta$ particles as spheres with the experimentally determined D_H . It greatly facilitates using the mathematically modeled data in Fig. 4 to explain the movements of the experimentally characterized particles.

We agree with the Reviewer that our description in the Methods section of the FDM-NTA protocol was insufficient. We have now expanded the protocol description detailing the most relevant quantitative variables (e.g., "max jump distance"). To facilitate the use by others, we have maintained a definition of critical instrumental settings. At least regarding stating relevant items, we believe our description now compares favorably to other papers in the field^{2, 4}.

Assuming that the authors did calculate the size correctly, it should be validated with an orthogonal method (e.g., comparing sizes of synthetic fibrils obtained with NTA and obtained with either SEM, TEM, or super-resolution imaging). The size validation should subsequently be applied to plasma spiked with abeta (i.e. repeat Fig 1D with an orthogonal technique). The labeling specificity should also be validated in biofluids spiked with Abeta (maybe using a fluorescently labeled abeta to co-localise with QD?), as Fig 1C seems to be done only in buffer.

***Author reply R1.3:** We have carefully considered the options suggested by the Reviewer while also considering some technical limitations imposed by the nature of the samples analyzed in our study.*

As a starting point, the Malvern NTA instrument's performance enables the analysis of particles with diameters up to 1 μm limited by sedimentation. The large $A\beta$ particles at 600-900 nm are consequently well within the operational limits of the instrument⁵.

Our previously submitted manuscript contained SEM micrographs of the QD-Ab bound to synthetic aggregated fibrils (formerly Fig. 1C, now Fig 1h-j), and the micrographs show the presence of fibrils >600 nm in length. This agrees reasonably well with the NTA results for the same material in plain PBS buffer (Suppl. Fig. 3) or

when this material was spiked into plasma and detected with FDM-NTA (Fig. 1e), as suggested by the Reviewer.

As a choice of orthogonal technique that can handle the aggregated A β in colloidal suspension, we have now included data from SV-AUC (Fig. 1f,g). As explained in the manuscript (pp. 7-8), this instrumentation cannot fully return a D_H for large fibrils. However, we used a distinct peak in the sedimentation profile at 25 S (Fig. 1g) for comparison with our earlier data on the likewise highly anisotropic superoligomeric mannan-binding lectin (spMBL). In the sedimentation profile of spMBL, a peak also occurs at 25 S (Fig. 1B in Juul-Madsen et al.⁶). From the small-angle X-ray scattering (SAXS) data presented in Gjelstrup et al.⁷ in Fig. 3 and Table II, D_{max} of the $4 \times 6 \times MBL_3$ complex can be calculated as approximately $0.5 \times 38 \text{ nm} + 3 \times d + 0.5 \times 38 \text{ nm} = 108 \text{ nm}$ (with $d=23.5 \text{ nm}$). Both with NTA in simple light scattering mode and FDM-NTA profiles of aggregated (A)-A β , peaks at $\sim 125 \text{ nm}$ were observable (Fig. 1c & Suppl. Fig. 3), which agrees nicely with the estimated D_H s for the 25-S peak of 108 nm. Taken together, there can be little doubt that FDM-NTA faithfully captures the dimension of the particles in reasonable agreement with some of their geometric properties in solution. We have now collected these results with the other validation experiments in a modified Fig. 1 and discussed the interpretation on pp. 7-8 and p. 18

Concerning the Reviewer's request "to repeat Fig 1D with an orthogonal technique", we do not think this will be effective. SV-AUC does not permit the addition of plasma with sufficient sensitivity to capture very large and quickly sedimenting particles, nor are we aware of any other solution-based technique that will and still return a size measure of particles that are, at the same time, polydisperse, large and rare. Super-resolution microscopy may operate on surface-coated samples, as shown in two studies^{8,9} brought forward by the Reviewer. We thank the Reviewer for guiding our attention to these interesting papers, which are now discussed in our paper and below. However, we are concerned that such an approach will not offer much to our data beyond what can already be extracted from the papers (see Author reply R1.4). Super-resolution microscopy determines the geometric size of the A β aggregates. To compare these results quantitatively with FDM-NTA, some reliable algorithm is

needed to convert the determined 2D geometries into their D_H equivalents of 3D colloidal particles. Frankly, we do not know how to accomplish such conversion, in particular for the larger aggregates of special interest to the present report since they show a complex morphology (Fig. 1h). Further, it is already known that the surface environment may perturb the structure in complex ways¹⁰. Although the topic is not without interest, we feel the required considerable efforts would take away from the central message of our paper, which is to integrate the hydrodynamic properties of the particles with responses by the immune system in MCI.

Concerning the labeling specificity, we can fully follow the reviewer's concern when analyzing complex liquid biopsies such as human plasma¹¹. For this reason, we included an extended number of tests. Specificity of the QD-Aducanumab for aggregated $A\beta$ (in PBS) was shown as done earlier earlier⁶ in Fig. 1e and Fig. 1i,j. Further, all plasma samples were analyzed with the inclusion of QDs conjugated to isotypic control Ab, as shown in Fig. 1b-e, producing negligible responses.

Sizes should be contrasted to the amyloidogenic aggregates reported by other laboratories. In healthy controls and Parkinson's serum and CSF, single-molecule imaging reports that the vast majority of amyloidogenic aggregates are between 40nm and 150nm in size (doi: 10.1093/brain/awab306); while the typical aggregate size of AD and MCI patients in CSF has been reported to be also between 50nm and 150nm (doi: 10.1186/s40478-019-0777-4). The authors should demonstrate that the 600nm threshold (as calculated with NTA) effectively corresponds to 600nm measured with an alternative technique.

Author reply R1.4: *We thank the Reviewer for this guidance and have now followed the suggestion by clarifying the relative D_H distribution. We now show a zoom-in of the D_H distribution in the interval 50-150 nm (Fig. 2b), as defined by De et al.⁸ and the interval of interest in our study (600-900 nm; Fig. 1c). In agreement with De et al.'s findings, it clearly shows that particles in the 50-150 nm interval also in our study is dominating. Unlike what De et al. found for CSF, we could not find any major difference between MCI patients and HC in plasma. In our study, the 600-nm (lower) threshold emerges solely from a comparison between the several controls included in*

our study, most importantly samples from healthy controls and the MCI patient plasma.

As noted in Author Reply R1.3, we have reservations about validating our findings with techniques based on recording 2D geometric properties, especially for large particles. Finding particles in micrographs with a certain, large geometry does not tell the expected D_H . Consequently, using super-resolution microscopy to confirm our data would suffer a lack of theoretical rigor, preventing an unambiguous validation. On the other hand, the comparison with the study by De et al. supports that FDM-NAT and super-resolution microscopy, in relative terms, produce comparable results, at least for the most abundant particles.

Some statistical methods require more robust tests since many of the effects reported as very robust are not particularly well justified by p-values and sample sizes (e.g. Figures 2 and 4). A statistics editor could double check this.

***Author reply R1.5:** The Reviewer's concern has prompted several adjustments to the analyses and made us expand our data set. This seems to have clarified the paper's message as specified further in Author Reply R1.6. We have focused on comparisons with $p < 0.05$ and highlighted non-significant comparisons when this helps the reading of the results. All group comparisons are made with ANOVA with a correction for multiple tests. In the experiment with purified primary human monocytes, the sample size has been expanded from 8 to 13 donors (Fig. 6). For the experiments with human induced pluripotent stem cell-derived microglial cells, a completely revised set of investigations now including 8 separate differentiations (Fig. 7), while the previous experiment included only 3.*

The mechanistic model which summarizes the study is not entirely supported by the reported data. There's no proof about monocytes entering the parenchyma. Most of the reported activity of peripheral immune cells has been reported at the level of the meninges (doi: 10.1016/j.cell.2020.12.040).

***Author reply R1.6:** As should now be clear from the present manuscript, we came to the same conclusion as the Reviewer. We have now revised both the Discussion and*

the schematic Fig. 8. Briefly, we have deleted speculations on the origin of the A β aggregates in the blood via the glymphatic system¹² as we have no data on this process. Our new schematic figure focuses on the deposition A β aggregates in the cortical microvasculature. As noted by the Reviewer, the meninges are an essential site for the accumulation of monocytes, which, in our thinking, is a part of the way to diapedesis into the cerebral cortex. Others have presented credible evidence that the monocytes penetrated even further into AD brains¹³, but this is not essential to the mechanisms we now investigate.

Considering that our work is done with data from living humans, important restrictions on what is experimentally possible prevent us from delivering "proof" of monocyte migration in Agg [+] MCI patients. However, our model integrates a considerable body made by others of clinical observations, work in animal models, and experimental systems^{13, 14, 15}. When combined with our work based on a tightly connected set of patient sample characterization, experiments in vitro, and mathematical modeling of the A β particle hydrodynamics, we now present a three-step model of events relating to activity of myeloid cells in MCI as discussed on p. 21. Plasma aggregates i) preferentially associate with the luminal side of thin cortical vessels, ii) enable monocyte crawling through CR4-mediated adhesion to amyloid A β , and iii) eventual permit extravasation via post-capillary venules into the brain. In this model, the lack of AD-like pathology in Agg [+] MCI patients is indirect evidence of enhanced clearance of brain pathology from increased infiltrating phagocytic cells.

Minor concerns:

However the introduction would benefit from improved clarity, language, and overall flow. The part explaining the connection between MCI and AD needs to be rewritten for clarity and accuracy (lines 17-24 of page 4). It would be good to include more information about why it is important to look at different abeta aggregate sizes.

***Author reply R1.7:** We have now completely rewritten the Introduction, starting with the open questions in understanding the role of the peripheral immune system in MCI. Guided by the Reviewer (Author Reply R1.4), we now use the findings especially by De et al.⁸ to argue that A β aggregate size distribution could be an important addition*

to understanding the pathogenesis in MCI as well as a recent report implicating low monocyte counts in plasma in development of AD¹⁶ (pp. 3-4).

Methods: More experimental detail is required if these results are to be replicated. It needs a thorough rewrite.

Author reply R1.8: We have now thoroughly rewritten the Methods section, especially concerning the FDM-NTA method (as also discussed in Author Reply R1.2)

Results: Statistics is not reported consistently throughout the manuscript. Formatting is also inconsistent between the figure legends.

Author reply R1.9: As described in Author Reply R1.5, we have dealt with the statistical challenges in several ways. In terms of consistency, we now use ANOVA throughout for group comparison with correction for multiple comparison and we only indicate significant ($p < 0.05$) comparisons. The figures have further been revised to make them as consistent as possible.

Figure 1: The two panels of Figure 1E would be more informative if they are merged into one graph; Figure 1F could be more informative in log scale

Author reply R1.10: The previous Fig. 1E is now split into Fig. 2a, c. Likewise, we agree with the Reviewer that (now) Fig. 2d is better presented on a log scale.

Figure 2: Please report correlation coefficients and the strength of the correlation

Author reply R1.11: R^2 is now indicated with all correlations together with the level of significance.

Figure 3: It is not explained why they added MnCl to the probes in Figure 2B

Author reply R1.12: $MnCl_2$ is a classic (and cheap) activator of integrin ligand binding¹⁷. In our hands, it works well with the CR3 and CR4-expressing K562 cells versus parental K562s¹⁸. We have now briefly mentioned its ability to activate on p. 13.

Figure 4: Please use an ANOVA when you compare more than 2 groups within the same graph and adjust the strength of your posthoc tests. Can you also show the difference between control and A(low) and M(low) in Figure 4K.

Author reply R1.13: The corrected and more consistent use of statistical tests are detailed in Author replies R1.5 & R1.9. This is implemented on (now) Fig. 2i-m, Fig. 6c,d,g,h, and Fig. 7c-f with much improved statistics from the increased number of experiments. The q-PCR data (previously Fig. 4K) is now included as Suppl. Fig. 13. As more data and groups are included in the analysis we no longer observe a statistically significant difference between the A β high and low groups for either CD11b (ITGAM) or CD11c (ITGAX).

Figure 6: Although the figure aims to interpret the results in a mechanistic context, the figure is not an accurate representation of the data provided. The authors claim trans-endothelial migration of monocytes in the brain and there is no direct evidence of such in this article. Also, the figure claims that sCD18 is increased in MCI patients positive for plasma aggregates. However Figure 2I shows that sCD18 levels of these patients are no different from controls. Figure 4 does not support a difference in the microglial response to monomeric and aggregated abeta. The y-axis could have a more accurate label (is "NTA concentration" a thing?).

Author reply R1.14: We have modified our schematic figure to summarize (mainly) events on the luminal side of the cerebral microvasculature as detailed in Author Reply R1.6. The data on sCD18 were considered too weak to form a part of the revised manuscript and were deleted. With the increased monocyte donor number (raised from 8 to 13), we have determined a clear and KIM127-dependent difference concerning phagolysosome co-localization with QD-labelled A β (Fig. 6d) and in clearance from the supernatant of primarily aggregated A β (Fig. 6h).

Discussion: The claims are too strong and not well supported by data.

Author reply R1.15: We have completely rewritten the Abstract and Discussion, striving to avoid overstatement of our findings.

The formatting is not consistent across figures – including abbreviations (e.g. aggregate: Agg or A?)

Author reply R1.16: We have now corrected these nomenclature problems.

Referring to the aggregates as "nanoscale", "nanosize" or "nanoparticulate" is not benefitting the description of this work.

Author reply R1.17: We agree with the Reviewer that the "nano designation" is inappropriate except in "nanoparticle tracking analysis," which seems to be the established name for this procedure. We were probably led astray by this name. We have now deleted "nano" in nearly all places in the text, including the title.

Reviewer #2 (Remarks to the Author):

In this paper, the authors analyze levels of nanoscale amyloid aggregates in patients using a novel technique, and they show that the subset of MCI patients with detectable large amyloid-beta aggregates have expanded populations of CD16+ monocytes (intermediate/non-classical), whereas MCI patients without amyloid-beta aggregates had expanded CD16- (classical) monocytes. Monocyte CD18 activation was correlated with levels of large amyloid-beta aggregates, and in vitro assays demonstrated that CR4 (but not CR3), which is a CD18-containing integrin receptor, preferentially binds aggregated amyloid-beta over monomeric amyloid-beta. Thus the CD18-activated monocytes in MCI patients with large amyloid-beta aggregates may be a result of direct interaction between monocyte CR4 and aggregated amyloid-beta. The authors then ask the question whether CR4 is an important receptor for phagocytic uptake of amyloid-beta aggregates by monocytes/other myeloid cells, which is the section that I will focus on, as an iPSC-microglia specialist. The subsection is titled "Phagocytosis of A β in human monocytes and induced pluripotent stem cell-derived microglia is mediated by CR4".

Overall, the methodology for generating iPSC-microglia seems sound, and some very basic validation of the cells identity has been performed using ICC for Iba1 (see Minor Comment 6). The flow cytometry methods appear sound, although the placement of gates is not very transparent (see Minor Comment 7).

Author reply R2.1: We thank the Reviewer for producing this succinct account of our paper's content about cellular biology. The flow cytometric data are now discussed in Author replies R2.6 & R2.7, including the requested gating strategies.

Amyloid-beta preparations appear to have been well-characterized (Supplementary figure S2). The figures are beautifully presented and indicate statistical significance clearly.

***Author reply R2.2:** We thank the Reviewer for appreciating our efforts to make effective figures. In reply to Reviewer #1 comments above (Author reply R1.3), we have now collected the characterization of aggregates of synthetic A β in Fig. 1 with the new addition of data from analytical ultracentrifugation as explained in the reply and the paper (on pp. 7-8).*

However, I have concerns about how these results have been reported in the main body of the text, there are a number of discrepancies that I will detail in the Major Comments below. The data does not support the conclusions for any of the subsection titled "Phagocytosis of A β in human monocytes and induced pluripotent stem cell-derived microglia is mediated by CR4", and most importantly do not provide any evidence that CR4 is directly involved with phagocytic clearance of A β aggregates, which seems to have been the purpose of these investigations.

***Author reply R2.3:** We agree with the Reviewer; this was an important concern about our past work. We have now included several new experiments and expanded the data set to confirm the role of CR4 better (as outlined in Author replies R2.4-R2.6).*

Major Comments:

1. In page10/line1 the authors state "KIM127 activation increased internalization of aggregated A β ". This is very misleading as Fig4C clearly shows that there is no statistically-significant difference between internalization of aggregated amyloid-beta +/- KIM127 (p value is 0.09, significance is conventionally considered to be below 0.05). This negative result is important and needs to be reported properly.

***Author reply R2.4:** We thank the reviewer for making this observation, which, in turn, has led to a more interesting and robust analysis from a total of 13 donors, increased from 8 in the previous study. The finding of no influence of KIM127 remains as shown previously when the total pool of particles is considered (now Fig. 6g). However, when excluding smaller particles (likely to present no or only small amounts of A β), the importance of KIM127 (i.e., integrin activation) is now evident (Fig. 6h). As discussed further below (Author reply R2.5), from the size selectivity would seem to agree with the function of CR4.*

This also means that the subtitle of this part of the Results - "Phagocytosis of A β in human monocytes and induced pluripotent stem cell-derived microglia is mediated by CR4" - is not supported by evidence. The authors have not demonstrated that CR4 mediates phagocytosis of amyloid-beta (aggregates) in human monocytes. I recommend that the subsection title is modified to reflect this.

Author reply R2.5: Cautioned by the Reviewer's point, we have now changed the title to "CD18 integrins support phagolysosome formation and phagocytosis of large, but not small, A β aggregates in human monocytes". However, in considering the data, we can list at least four reasons to suggest CR4 as important in the process leading to size selectivity from data shown in (now) Fig. 6. First, in monocytes, we found a significant binding to aggregated A β by CR4, but not CR3 (Fig. 5h). Second, in K562 cells, CR4 binds well aggregated A β , but not monomeric forms (Fig. 5f). Third, in biochemical assays (SPR) with the CR4 I domain, we make the same observation of a very select binding to aggregated A β (Fig. 5k,l). Fourth, when we analyze the size of the depleted QDs conjugated with A β , we find that the sizes involved are in the 60-200 nm interval (Fig. 6f), which has a correlate with the size requirement for avidity binding predicted from structural analysis of the integrins and the A β fibrils. Taken together, we believe this forms a body of evidence for an important involvement of CR4 in phagocytosis of aggregated A β by monocytes, further supported by our earlier observations with α -synuclein¹⁹. These points are presented in several places in the paper and summarized on pp. 20-21 of the Discussion.

2. In Figure 4H, iPSC-microglia are incubated with monomeric and aggregated amyloid-beta-42, and flow cytometry performed to evaluate the CR4 expression and amyloid-beta uptake of single cells. The authors selected a subset of cells with higher CR4 expression and showed that higher CR4 expression is associated with significantly more aggregated amyloid-beta uptake than monomeric amyloid-beta. The evidence does support a potential correlation between CR4 expression and aggregated amyloid-beta uptake, but is not sufficient to support the claim that phagocytosis of amyloid-beta is mediated by CR4 in iPSC-microglia (in the subsection title, see Major Comment 1). To evidence this the authors would need to do experiments with CR4 knockout or knockdown. I recommend that the subsection title is modified to avoid claiming that that CR4 mediates phagocytosis of amyloid-beta (aggregates) in human iPSC-microglia.

Author reply R2.6: *We thank the Reviewer for these valuable suggestions. The function of CR4 in human microglial cells has escaped much investigation in the literature. Concerning the involvement of CR4 in microglial phagocytosis, we have chosen to make what we term functional knockouts by use of well-characterized function-blocking antibodies to CR4²⁰. This choice was made because we fear genetic ablation of CR4 in the iMGs could have unwanted consequences, at least partly due to the shared/competing signaling pathways by the group of CD18 integrins²¹. Indeed, such experiences were made with gene ablation of integrins in mice, where compensation has been demonstrated on several occasions, probably because totipotent embryonic cells have a high capacity for compensation²². Of course, function-blocking antibodies may also lead to biases, but, at least in the case of CD18 integrins, they are a tried and tested methodology with multiple cell types^{18, 23} and connect well with our investigations of primary human monocytes (Fig. 5h). This approach now confirms that CR4 play a significant role in phagocytosis of aggregated A β by iMGs (Fig. 7g). Of course, the strong preference for aggregated A β in these experiments further invokes CR4 function as the mechanistic explanation as also was the case for monocytes (see Author reply R2.5). Finally, we have analyzed more extensive experiments (n=6-8) than earlier (n=3). Consistent with these data, we have modified the subsection title to "A β aggregates are preferentially phagocytosed and stimulate lysosomal activity through CR4 in stem cell-derived microglia."*

3. In page10/line20 the authors state "We found increased CR4 transcription only in iMGs treated with aggregated A β and showing a low A β uptake, while CR4 levels were unchanged in cells with high A β uptake (Fig. 4K)". In Figure 4K the statistically significant difference in CR4 expression is between iPSC-microglia with high A β uptake versus low A β uptake, however there is no significant difference between high A β uptake versus no A β addition. This should be stated clearly in the main text, since this is the relevant comparison.

Author reply R2.7: *We agree with the Reviewer that, in hindsight, this set of experiments was neither precisely designed nor did they return any major results. We have now made a more straightforward and relevant analysis, namely the connection between ITGAM and ITGAX transcription and the expression of their products in the iMGs membrane. This is inspired by the classic work on monocytes and neutrophils*

(but not microglial cells) by Miller et al.²⁴, which showed that intracellular pools of CD11b (i.e., CR3) and CD11c (CR4) is released to the membrane with appropriate stimulus. Consistent with such a mechanism, we now find that membrane expression of CD11c, but CD11b, is increased (Fig. 7e,f), but with no increase in transcription of the genes (Suppl. Fig. 13).

There is insufficient evidence to support the statement in page11/line1: "These findings suggest a functional coupling of aggregated A β phagocytosis and CR4 expression in iMGs", or the statement in page12/line17 "our data showing that A β aggregates increase the CR4 expression in human iPSCderived microglia". I recommend that these are re-phrased or removed.

***Author reply R2.8:** In our new set of experiments, involving more iMGs culture, we now find significant support for these findings as shown in Fig. 7e. The text has been revised to address the new interpretation made as further explained above in Author reply R2.7.*

4. The Abstract states “[CR4] binds size-selectively to nanoscale amyloid beta and promotes a disease-modifying phenotype also found in human induced pluripotent stem cell-derived microglia”. There is no evidence of a disease-modifying phenotype in the iPSC-microglia, what data supports this claim?

***Author reply R2.9:** These remarks were intended to relate some aspects of the TREM2 expression. As mentioned in more detail in Author reply R2.13, we have removed this part of our work, and the phenotype in question is no longer discussed.*

Minor comments:

1. The authors do not explain why adding fluorescent amyloid-beta to primary monocytes and activating the cells with KIM127 is supposed to illuminate the role of CR4. KIM127 is previously described (see page 7 line 19) as a conformation-specific antibody for CD18 that can be used to probe levels of CD18 activation. KIM127 is not explained to be a CD18-activating antibody, although we can infer this indirectly from the result in Fig 3E where addition of KIM127 to monocytes improves their adhesion to aggregated amyloid-beta. It would be helpful if this were spelled out in page 9-line 20, and also it should be acknowledged that both CR3 and CR4 contain

CD18, so an assumption has been made that KIM127 activates CR4 (presumably based on the in vitro assays in Figure 2).

Author reply R2.10: The use of KIM127 as a CD18 integrin-activating Ab is well established. However, as recommended by the Reviewer, a few details and a helpful reference²³ have now been added to the first description of such use of the Ab on p. 14

2. The authors state in page10/line12 that they used iPSC-microglia to investigate "how CR4 amyloid recognition by microglial cells contributes to the signaling cascade". Firstly, the signalling cascade of what, and how does this relate to the subsection title? The authors need to be clearer about what they are investigating.

Author reply R2.11: This part of our work intended to study transcriptional alterations in, e.g., TREM2 following CR4 ligation with (aggregated) A β . Integrins perform so-called outside-in signaling when binding ligand²⁵, but evidence in our previous work from the transcriptional analyses was weak, as also noted by the Reviewer. We have deleted this work and focused on the signaling leading to increased phagolysosome activity (Fig. 7d) and the consequences for CD11c membrane expression (Fig. 7e)

3. There is a typographical error at page10/line22: "This difference was seen for iMGs with low and high uptake of monomeric A β ", should actually be "This difference was not seen for iMGs with low and high uptake of monomeric A β ".

Author reply R2.12: This work is now deleted from the manuscript.

4. The data on TREM2 expression in monocytes and iPSC-microglia is completely superfluous for the conclusions of the paper, and does not appear to add anything to the story. The proposed interaction between CD18 activation and TREM2 transcription is not convincing, as the error bars in Fig 4E are very large (for monomeric or aggregated A β + K127), clearly there were 2-3 experimental repeats where no induction of TREM2 transcription occurred. Furthermore the finding in Fig 4L that both monomeric and aggregated A β caused an induction of TREM2 transcription actually indicates that CD18 activation is unlikely to have a major contribution to TREM2 transcription, since CR4 was shown to preferentially bind aggregated A β .

Author reply R2.13: We agree with the Reviewer, and our analyses of TREM2 are now deleted from the manuscript.

5. Supplementary figure S8- the title of this figure is "NTA detection of A β QD-conjugates from monocyte phagocytosis shows that CD18 activation promotes phagocytosis of aggregated A β ". However, there are no statistically significant difference and the data is extremely noisy, so this statement is not supported by data shown.

Author reply R2.14: As outlined in several places above, we have now increased the number of experiments with primary monocytes. These data are presented in Fig. 6, where significant findings in Panels d, f, g, and h now support our interpretations.

6. Supplementary figure S10- the only validation of cell identity for the iPSC-microglia appears to be immunocytochemistry of Iba1. Iba1 is expressed highly by macrophages as well as microglia. A better microglial identity marker would be TMEM119.

Author reply R2.15: By the Reviewers request, we expanded or validation of microglia differentiation by the inclusion of the marker P2RY12 using confocal microscopy in Suppl. Fig. 11. Furthermore, we have followed increase in expression of Iba1 and P2RY12 using q-PCR which is also included in Suppl. Fig. 11.

7. Supplementary figure S11- in S11a and S11b the data that was used for setting the gates should be shown, ideally single-stained controls. Otherwise we are expected to take it on trust that the gates were placed correctly.

Author reply R2.16: We thank the reviewer for this suggestion and have now included the suggested single-stain controls in Suppl. Figs. 12 & 14.

Reviewer #3 (Remarks to the Author):

The authors study the relationship of soluble amyloid beta aggregates in the plasma of patients with mild cognitive impairment with CNS measures of brain inflammation. They find that about a third of patients with MCI have circulating Abeta aggregates >600 nm and this relates increased number of non-classical monocytes and lower CNS inflammation. They also find that a receptor for Ab on monocytes is CR4. This is an interesting study, but there are some questions.

1. The sample size is borderline for the conclusion given that there are multiple comparisons and the best p values are at <0.01 at 10 samples with aggregates.

Author reply R3.1: It is correct that some samples may appear low in absolute terms. At least for the clinical side of our experiment, there are significant limitations in establishing an MCI cohort of the size used in our study. However, compared with other interdisciplinary studies, it becomes very clear that our study is not lacking in patient samples. For instance, in one recent study highlighted by Reviewer #1 (De et al.⁸, also discussed in Author replies R1.3, R1.4 & R1.6.), a total of 8 MCI (and 8 AD patients) were studied. Tiiman et al. include 4 MCI patients out of 24 mainly AD patients²⁶. Our study, spanning paired samples of patient plasma, leukocytes, and PET scans, included 38 well-characterized MCI patients^{27, 28}. For this reason, we believe our study is significant progress in unraveling pathogenic mechanisms in MCI patients, especially when multiple advanced techniques are required to stitch together an embracing characterization.

The clinical algorithm for MCI classification has few hard end points²⁹, which undoubtedly explains part of the trouble in raising the significance level in the patient analyses. However, we have now focused on reporting on significance levels $p < 0.05$, according to field standards. In a few places, the lack of significance has been mentioned to help understand our findings' nature. We have used ANOVA across data analysis where appropriate for all data sets. Taken together, we believe the more stringent analyses have improved the clarity and internal consistency of the paper.

The real strength of the cohort is the link between several sets of technically unrelated data, e.g., our characterization plasma A β aggregates, with data on the peripheral leukocytes, and with radiological data. Of course, in the individual analyses, it is difficult to capture by any simple statistics that one set of data efficiently stratifies other analyses with statistical significance. Yet, it is unlikely to be a statistical artifact that, e.g., the A β plasma aggregates in this way stratify analyses of patient leukocyte subsets and the radiological data.

2. In the analysis in Figure 2, it would be helpful to see data for CD11c and CD11b expression in addition to CD8 and KIM127, which to an extent dependent epitope on CR3 and CR4. The correlation of ab with KIM127 positivity is helpful.

***Author reply R3.2:** We thank the Reviewer for highlighting our interesting finding concerning KIM127. We agree that information on CD11b and CD11c expression in the MCI patients would have been helpful. Initially, we chose to follow the CD18 expression as a kind of combined marker for the CD18 integrins, and this also seems to provide helpful information, as shown in Fig. 2n,o. The re-designed in vitro experiments on the combined gene transcription and membrane protein expression of CD11b and CD11c also fill a void on how these proteins may respond to A β exposure. Unfortunately, no more patient samples are left for testing CD11b or CD11c expression.*

3. The SPR data for CR4 is characteristically hard to interpret. Compared to the relatively well behaved, but weak, binding for CR3, the response for the CR4 I domain is very weak and noisy. The adhesion result is supportive of CR4 being the receptor, but the SPR result suggests that there is so few CD4 I domain capable to binding Abeta as to be hard to interpret or trust.

***Author reply R3.3:** We have repeated these experiments and achieved the same results. We agree that the signal in response units is low, but we can analyze them. We have now used the structural analyses more actively to explain that the low signal for CR4 I domain binding compared to CR3 I domain very well matches how the CR4 I domain binding is formed, i.e., the ladder of glutamate residues from multiple A β chains (5a,b). This naturally limits the stoichiometry of the I domain interaction with the A β chains and hence the maximum binding capacity reflected in the low SPR signal. This is now explained on pp. 13 and 14. We have not dwelled on the nature of the CR3 I domain binding epitope. It likely involves a short motif in the primary structure with a basic side chain as found for other CR3 ligands²³. Our findings are strikingly similar to what we reported for α -synuclein¹⁹. We now also used a similar way of thinking to model the interaction of CR4 with a mature fibril. Briefly, as explained in the text (on p. 13), it leads to the hypothesis that high-avidity binding occurs mainly for relatively long segments since the CR4 I domain epitope is repeated in orientation relative to the integrin-presenting cell surface every 43 nm (Fig. 5c).*

This seems to receive experimental support from the data shown in Fig. 6f. Of course, it is understood that less ordered aggregates of A β may not present interactions in such a well-ordered fashion, but the principles may still be valuable for explaining the CR4-mediated phagocytosis.

4. The idea that Abeta binding to CR4 upregulates CR4 transcription is interesting, but I would not have expected it. Is this well known for iC3b and CR4 expression?

***Author reply R3.4:** The Reviewer has addressed an important weakness in our past work, and we apologize for the confusion our previous may have caused. The transcriptional analyses were initially made on a limited set of samples (N=3). In a better-focused effort (see several Author replies to Reviewer #2 above, including R2.7 and R2.13), we have now expanded the number of iMGs culture to N=6-8 (Fig. 7c-g and Suppl. Fig. 13), and find – as one probably would expect – no change in the transcription of ITGAX. However, compared to the earlier manuscript, we have now made a side-by-side comparison with the membrane protein expression of CD11c. It increased with a stimulus with aggregates, but not monomeric, A β . This makes more sense compared to the existing literature since a classic finding was the release of CR4 from intracellular pools to the membrane²⁴, which would fit this pattern.*

5. The statistical differences are marginal at p<0.05. It has been discussed that this is not really an acceptable standard as it means that 5% of published studies will report erroneous data under the best of circumstances. Can a power analysis be done and the sample size increased to reach p<0.005? Or this issue should be discussed as a marginal call between a trend and significant.

***Author reply R3.5:** While we agree with the Reviewer that concerns exist regarding the present use of statistics, we do not believe our paper would be the best place to alter these practices, e.g., with an ad hoc adjustment of level-of-significance to p<0.005. It should also be remembered that, in clinical studies, sample size is determined by what is clinically possible and ethically acceptable. MCI patients are not suffering severe symptoms²⁹, which limits their contact with the clinic and what investigations can be justified on a routine basis. Our analysis, generating multiple linked data sets for each patient, is unusual among the reports in the literature, as we explained in Author Reply R3.1. We cannot perform a statistical test returning a*

significance level for the multiple ways the presence of A β plasma aggregates seems to influence or stratify other technical independent tests. However, it must imply that we are unlikely to report on statistical artifacts. As stated in several places above (e.g., Author reply R3.4), we increased the number of repeats with donor-derived monocytes and iMGs cultures to support the experimental work in vitro with monocytes and iMGs. Also, we have been more consistent about the statistical tests used, e.g., correcting multiple tests when necessary and focusing on results with $p < 0.05$. Following the Reviewers' advice, our paper now meets well-established standards and presents a more unmistakable message.

Reviewer #4 (Remarks to the Author):

As per Editor request, I will only review the positron emission tomography (PET) aspects of this study. Overall, the PET quantification methods are poorly described, some results appear to be driven by errors in the quantification pipeline, and the details of the brain-wide statistical analysis are vaguely reported.

1) In Figure 5A and 5B (left and mid panels), the authors report "Mean overall cortical 18F-FTP PET (or 11C-PiB) uptake in each group at baseline, two-year follow-up, and longitudinal change." The reported mean SUVR values, however, are in the range of 10^9 , a value that is clearly not possible for an average SUVR (doi: 10.1093/brain/awz268). What are they actually reporting in Figure 5A and 5B?

***Author reply R4.1:** The units in Figures 5A and 5B (now Fig. 3a and 3b) have been changed to the correct SUVR values. In the original figure, the error was due to a misinterpretation of "," and "." in the software used to produce the plots, e.g., 2.67617582 was interpreted as 267617582.*

2) In Figure 5A and 5B (left and mid panels), there are 3-4 cases in the MCI Agg[-] group with extremely low SUVR values. Again, we don't see cases with such low SUVR values (doi: 10.1093/brain/awz268; doi: 10.1093/brain/aww334). These cases seem to be the result of errors in the quantification pipeline, probably due to errors in the segmentation or coregistration processes.

***Author reply R4.2:** This is caused by the same error explained above (Author Reply R4.1). The SUVR value of these cases ended with 0, resulting in 1 decimal point less than the rest. Thus, the values were interpreted as a factor 10 smaller than the rest.*

3) In Figure 5A and 5B (right panel), there are subjects with extremely large SUVR changes after two years. Amyloid and tau accumulation are very slow processes, typically 0.1-0.2 SUVR units/year (doi: 10.1093/brain/awy059; doi: 10.1001/jamaneurol.2019.1424). This, again, suggests that there have been errors with the PET quantification pipeline.

Author reply R4.3: This is caused by the same error explained in Author Replies R4.1 and R4.2. Figures have been updated to reflect the correct values.

4) The authors should be clearer with their approach of measuring PET signal in the overall cortex. This approach is certainly less efficient to detect actual accumulation, as there are several regions in the cortex that do not show elevated amyloid or tau-PET signal. Why not using established regions of interest for measuring amyloid and tau (doi: 10.1016/j.jalz.2016.08.005)?

Author reply R4.4: We agree with the reviewer that a composite ROI for each tracer would be more sensitive in differentiating elevated amyloid/tau PET signal. Accordingly, (now) Figures 3a and 3b have been updated to report the values within the tau and amyloid regions, respectively, established in C.R Jack Jr. et al.³⁰. The figure legend and the method description have been updated to reflect this.

5) The authors only vaguely describe the voxel (or vertex)-wise statistical threshold for their cortical maps. Based on Figure 5C-D, I assume they apply an uncorrected $p < 0.05$. However, they also state in Fig. 5's legend that they corrected for multiple comparisons, but they do not describe which method (cluster, FWE, FDR...) they used.

Author reply R4.5: The vertexwise statistical maps were FWE corrected with a cluster-extent-based thresholding method. This has been added to the legend of Fig. 3: "Statistical cortical maps were family-wise error rate corrected ($\alpha = 0.05$) using cluster-extent-based thresholding with a primary cluster-defining threshold of $p < 0.05$."

6) There are no details on how PET images were reconstructed, in particular whether the acquisition was static or dynamic (and then averaged).

Author reply R4.6: A more detailed description of the neuroimaging analysis has been added to the Methods section. This includes details on the PET acquisition.

7) The authors do not report the software they used to segment the MRI scans, to quantify the PET scans, and to map SUVR values to the cortical surface.

Author reply R4.7: In relation to the previous point (Author reply R4.6), a more detailed description of the neuroimaging analysis has been added to the Methods section. This includes information on the MRI and PET analyses, details on the extraction of the cortical surface, and the mapping of SUVR values.

8) SUVR stands for standardized uptake value ratio.

Author reply R4.8: This has been corrected.

9) The authors state that mapping SUVR values to the cortical surface "ensures that the PET signal only originates from cortical gray matter." This is not accurate. It is true that measuring SUVR values in the midsection of the cortical mantle can reduce spill-out effects. However, spill-in signal from other regions (e.g. from the white matter) can also occur, and mapping SUVR values to the cortex will not solve this issue.

Author reply R4.9: The Reviewer is correct. The applied method does not ensure that the PET signal only originates from cortical gray matter. Due to the PET system's wide point spread function and the relatively thin nature of the cerebral cortex, there is undoubtedly a spill-in signal in all reconstructed cortical voxels. The proposed method aims to minimize the spill-in effect by approximating the cortical mantle's midsection. However, we agree that the effect cannot be entirely avoided. The text has been rephrased to reflect this.

10) It would be useful to categorize MCI patients as amyloid-positive or negative using established cut-points for e.g. Centiloids. Presence of amyloid positivity is strongly associated with tau pathology and overall AD pathology. It would be interesting to see how traditional amyloid positivity correlates with Aggregate-positivity.

Author reply R4.10: We agree that reporting amyloid positivity would provide valuable information and make comparing the A β aggregate-positivity with traditional amyloid positivity easier. Accordingly, this has been added to Table S1 in the Supplementary Materials. The mean ¹¹C-PiB SUVR in the composite ROI provided

in C.R Jack Jr. et al.³⁰ defines amyloid positivity with a threshold of 1.5 SUVR. This threshold has previously been used to categorize Agg [+]/Agg [-] individuals in this cohort^{27, 31}.

References

1. Einstein, A. The motion of elements suspended in static liquids as claimed in the molecular kinetic theory of heat. *Ann Phys-Berlin* **17**, 549-560 (1905).
2. Moore, C., Wing, R., Pham, T. & Jokerst, J.V. Multispectral Nanoparticle Tracking Analysis for the Real-Time and Label-Free Characterization of Amyloid-beta Self-Assembly In Vitro. *Anal Chem* **92**, 11590-11599 (2020).
3. Wang, H. *et al.* Somatostatin binds to the human amyloid beta peptide and favors the formation of distinct oligomers. *Elife* **6** (2017).
4. Hoover, B.M. & Murphy, R.M. Evaluation of Nanoparticle Tracking Analysis for the Detection of Rod-Shaped Particles and Protein Aggregates. *J Pharm Sci* **109**, 452-463 (2020).
5. NANOSIGHT NS300 OPERATING MANUAL. Malvern (UK): Malvern Instruments Ltd.; 2015.
6. Juul-Madsen, K. *et al.* Characterization of DNA-protein complexes by nanoparticle tracking analysis and their association with systemic lupus erythematosus. *Proc Natl Acad Sci U S A* **118** (2021).
7. Gjelstrup, L.C. *et al.* The role of nanometer-scaled ligand patterns in polyvalent binding by large mannan-binding lectin oligomers. *J Immunol* **188**, 1292-1306 (2012).
8. De, S. *et al.* Soluble aggregates present in cerebrospinal fluid change in size and mechanism of toxicity during Alzheimer's disease progression. *Acta Neuropathol Commun* **7**, 120 (2019).
9. Lobanova, E. *et al.* Imaging protein aggregates in the serum and cerebrospinal fluid in Parkinson's disease. *Brain* **145**, 632-643 (2022).
10. Cholko, T., Barnum, J. & Chang, C.A. Amyloid-beta (A β 42) Peptide Aggregation Rate and Mechanism on Surfaces with Widely Varied Properties: Insights from Brownian Dynamics Simulations. *J Phys Chem B* **124**, 5549-5558 (2020).
11. Kragstrup, T.W., Vorup-Jensen, T., Deleuran, B. & Hvid, M. A simple set of validation steps identifies and removes false results in a sandwich enzyme-linked immunosorbent assay caused by anti-animal IgG antibodies in plasma from arthritis patients. *Springerplus* **2**, 263 (2013).
12. Rasmussen, M.K., Mestre, H. & Nedergaard, M. Fluid transport in the brain. *Physiol Rev* **102**, 1025-1151 (2022).

13. Munoz-Castro, C. *et al.* Monocyte-derived cells invade brain parenchyma and amyloid plaques in human Alzheimer's disease hippocampus. *Acta Neuropathol Commun* **11**, 31 (2023).
14. Terkelsen, M.H. *et al.* Neuroinflammation and Immune Changes in Prodromal Parkinson's Disease and Other Synucleinopathies. *J Parkinsons Dis* (2022).
15. Kapellos, T.S. *et al.* Human Monocyte Subsets and Phenotypes in Major Chronic Inflammatory Diseases. *Front Immunol* **10**, 2035 (2019).
16. Luo, J., Thomassen, J.Q., Nordestgaard, B.G., Tybjaerg-Hansen, A. & Frikke-Schmidt, R. Blood Leukocyte Counts in Alzheimer Disease. *JAMA Netw Open* **5**, e2235648 (2022).
17. Dransfield, I., Cabanas, C., Craig, A. & Hogg, N. Divalent cation regulation of the function of the leukocyte integrin LFA-1. *J Cell Biol* **116**, 219-226 (1992).
18. Vorup-Jensen, T. *et al.* Exposure of acidic residues as a danger signal for recognition of fibrinogen and other macromolecules by integrin alphaXbeta2. *Proc Natl Acad Sci U S A* **102**, 1614-1619 (2005).
19. Juul-Madsen, K. *et al.* Size-Selective Phagocytic Clearance of Fibrillar alpha-Synuclein through Conformational Activation of Complement Receptor 4. *J Immunol* **204**, 1345-1361 (2020).
20. Hogg, N., Takacs, L., Palmer, D.G., Selvendran, Y. & Allen, C. The p150,95 molecule is a marker of human mononuclear phagocytes: comparison with expression of class II molecules. *Eur J Immunol* **16**, 240-248 (1986).
21. Tan, S.M. The leucocyte beta2 (CD18) integrins: the structure, functional regulation and signalling properties. *Biosci Rep* **32**, 241-269 (2012).
22. Bouvard, D. *et al.* Functional consequences of integrin gene mutations in mice. *Circ Res* **89**, 211-223 (2001).
23. Vorup-Jensen, T. & Jensen, R.K. Structural Immunology of Complement Receptors 3 and 4. *Front Immunol* **9**, 2716 (2018).
24. Miller, L.J., Bainton, D.F., Borregaard, N. & Springer, T.A. Stimulated mobilization of monocyte Mac-1 and p150,95 adhesion proteins from an intracellular vesicular compartment to the cell surface. *J Clin Invest* **80**, 535-544 (1987).
25. Hynes, R.O. Integrins: bidirectional, allosteric signaling machines. *Cell* **110**, 673-687 (2002).
26. Tiiman, A. *et al.* Amyloidogenic Nanoplaques in Blood Serum of Patients with Alzheimer's Disease Revealed by Time-Resolved Thioflavin T Fluorescence Intensity Fluctuation Analysis. *J Alzheimers Dis* **68**, 571-582 (2019).
27. Parbo, P. *et al.* Brain inflammation accompanies amyloid in the majority of mild cognitive impairment cases due to Alzheimer's disease. *Brain* **140**, 2002-2011 (2017).

28. Parbo, P. *et al.* Low plasma neurofilament light levels associated with raised cortical microglial activation suggest inflammation acts to protect prodromal Alzheimer's disease. *Alzheimers Res Ther* **12**, 3 (2020).
29. Petersen, R.C. *et al.* Mild cognitive impairment: clinical characterization and outcome. *Arch Neurol* **56**, 303-308 (1999).
30. Jack, C.R., Jr. *et al.* Defining imaging biomarker cut points for brain aging and Alzheimer's disease. *Alzheimers Dement* **13**, 205-216 (2017).
31. Ismail, R. *et al.* The relationships between neuroinflammation, beta-amyloid and tau deposition in Alzheimer's disease: a longitudinal PET study. *J Neuroinflammation* **17**, 151 (2020).

REVIEWER COMMENTS

Reviewer #1 (Remarks to the Author):

The revised version of the manuscript by Juul-Madsen and colleagues studying interactions between monocytes and amyloid beta aggregates is much improved comparing to the initial submission. I believe that most reviewers' concerns have been successfully addressed. I still think that there should be more clarity regarding the physiological relevance of the "600nm aggregate size" threshold. I'm still not sure if the large size aggregates are formed by clustering several fibrils is genuinely circulating in blood or they form during sample preparation stages (as it's the case in the aggregate show in in Fig1 h (where several well-defined fibrils are bundled together). Maybe it's not relevant from a clinical diagnosis point of view (are there large clusters or not), but it certainly drives some of the conclusions of the paper in terms of the role of aggregates in monocyte activation and the proposed mechanism shown in Figure 8. The language of the manuscript, together with the scheme shown in Figure 8 suggests that the aggregates that cause that monocyte activation are fibrils that grow long, while the only SEM image in Fig1 h shows a bundle of fibrils. Is this just a semantic problem, defining what an aggregate or cluster is?

To sum up, my main concern with the manuscript is that is not clear how aggregate size-dependent interactions with monocytes will have such a dramatic effect in activation. Where is the size-dependent regulation? It has been shown that structural conversion during aggregation happens at sizes way below those 600nm. So if it's just the aggregate getting bigger (elongation of the fibril) that drives the divergence in toxicity, do receptors act in a different way as a response to the size? I understand that fishing those clusters with SEM in human samples would be a tedious experiment if they are in ultra-low abundance, so the conclusions of the paper in the current state require some leap of faith. I believe that the science reported in the manuscript deserves publication, but the meaning of that size threshold needs to be clarified, or toned down.

Reviewer #2 (Remarks to the Author):

In the previous submission of this manuscript I was Reviewer #2, and have been given the opportunity to review the latest submission, which is much improved. As before, I will only comment on the monocyte and microglia work.

The authors present good evidence that CR4 mediates monocyte binding to large A β aggregates, but not very strong evidence that CR4 mediates phagocytosis of large A β aggregates (see point 4 below). In the abstract, the authors suggest that CR4 activation by large A β aggregates leads to CD18-rich monocytes being recruited to brain tissue, to explain the absence of CD18-rich monocytes in human blood circulation where large A β aggregates are present. This is highly speculative and was not investigated in this study, but would be of future interest to explore. The authors also claim in the abstract that CR4 mediates phagocytosis of large aggregates by microglia. These claims are now well-evidenced, as they

showed that in iPSC-microglia preferentially take up large aggregates of A β over monomers, the CD18-activating antibody KIM127 selectively enhances large aggregate (not monomers) phagocytosis, which leads to a corresponding rise in lysosomal activity, and a CR4-blocking antibody attenuates the effect of KIM127 on A-A β phagocytosis. I believe that this is an important finding that deserves publication, and I commend the authors on their experimental design and execution for the iPSC-microglia work, as the data is beautifully consistent and has all of the appropriate controls included. My previous comments have all been satisfactorily addressed, apart from the issue of whether there is sufficient proof to say that monocytes perform CR4-mediated clearance of large A β aggregates (I discuss in points 4 and 6 below).

Specific comments:

1. Lines 40-41: In the abstract the authors state: "In microglial cells, CR4 stimulates phagolysosomal activity". This is inaccurate because KIM127 did not by itself induce phagolysosomal activity, it potentiated the effect of aggregated A β . The authors should reword to make this more clear.
2. Line 41: "CR4 phagocytoses in particular large aggregates", actually it would be more precise to say "Microglia phagocytosis of large aggregates of A β is mediated by CR4" or similar. The phagocytosis is performed by the microglia, rather than by the CR4.
3. Lines 312-313: The subsection title is "CD18 integrins support phagolysosome activity and depletion of large, but not small, A β aggregates in human monocytes". In this section the authors show that CD18 integrin activation has no effect on phagolysosomal activity, therefore the data does not support this claim. A more accurate section title would be something more along the lines of "CD18 integrins support selective depletion of large, but not small, A β aggregates in human monocytes".
4. Lines 320-330 (Figures 6e,f,g,h): It is very interesting that CR4-activation does not increase internalization of A β -coated QDs into monocytes, neither does CR4-activation promote lysosome colocalization with A β -coated QDs. The authors' assertion (in the discussion, see point 6) that CR4 mediates amyloid clearance by monocytes seems to rest only on the data in Fig 6h where the size distribution of particles remaining in the supernatant after phagocytosis was analysed, and it was found that the proportion of "large" (> 50nm) particles is decreased by CR4-activation. However the wording of the text is confusing, it states that "Analyzing particles with DH >50 nm only, more were depleted from the supernatant when they were conjugated with A-A β than if they carried M-A β on the conditions of KIM127 integrin activation (Fig. 6g) implicating CD18 integrins in the process". This implies that the relevant comparison is A-A β versus M-A β , but Fig 6h does not appear to show a significant difference between A-A β and M-A β . What Fig 6h appears to show is a significant difference in uptake of A-A β -coated particles (>50nm only), between KIM127-treated and untreated cells. The authors should edit the text to better communicate this finding.
5. Line 330: Typographical error. The Figure referred to here should be Fig. 6h, not Fig. 6g.
6. Lines 451-454: "In this model, the lack of AD-like pathology in Agg [+] MCI patients is indirect evidence of enhanced clearance of brain pathology from an increased number of infiltrating phagocytic cells. We have shown that both monocytes and iMGs are capable of amyloid clearance in a CR4-involved process..." The evidence presented in this manuscript does not clearly show that monocytes are capable of CR4-mediated amyloid clearance (as I discussed in point 4), so I disagree with this wording. I read the authors' rebuttal. In their rebuttal point R2.5 they claim that their measurements of A β binding to monocytes and depletion of A β from the cell supernatant "forms a body of evidence for an important

involvement of CR4 in phagocytosis of aggregated A β by monocytes". However none of these are direct measures of amyloid (phagocytic) clearance, and they are ignoring their own negative phagocytosis assay data (Fig. 6c,d,g). Unlike the monocytes, it is clear from their data that microglia are capable of CR4-mediated phagocytosis of large A β aggregates. Therefore, why is the lack of AD-like pathology in Agg [+] MCI patients evidence that infiltrating phagocytic cells have cleared the brain pathology? There are many other potential explanations, including that microglia are wholly responsible. I suggest that the highlighted sentences are rephrased to avoid completely unsubstantiated speculation.

7. The authors mention 'ROS' and 'neurotoxic ROS species' in the abstract, introduction and discussion sections, however none of the experiments in this study address 'ROS' or 'neurotoxicity'. It seems very weird to devote a whole paragraph of the discussion to this. It is not a mainstream view in the microglia field that phagocytosis of A β leads to neurotoxicity via ROS. I looked up the review paper (ref 32) referenced to support the assertion that "Usually, ROS production accompanies lysosomal activity", and there is nothing in the review paper to support this statement. I suggest that the mention of ROS is removed.

Reviewer #3 (Remarks to the Author):

The authors have discovered 600 nm A-Beta aggregates in the circulation of patients with mild cognitive impairment and found that these asymmetric aggregates can engage CR4 (CD11c/CD18) leading to adhesion of CD18 high monocytes. The authors have responded thoroughly to the reviewer comments, which has improved the manuscript.

Reviewer #4 (Remarks to the Author):

The authors have addressed all my comments, and now the numerical results they report seem to be correct.

There are two very minor issues that remain:

1) In figure 3a and 3b the authors use right superscripts. The mass number is typically expressed as a left superscript (e.g. ¹⁸F).

2) The authors state "Magnetic resonance imaging (MRI) and PET data was performed using MINC (Medical Imaging 636 NetCDF) Toolkit (<https://bic-mni.github.io>).". I would replace "performed using" by "analyzed with".

Point-by-point reply to reviewers' comments (NCOMMS-22-53102A-Z; "Large amyloid beta aggregates activate peripheral monocytes and protect against neuropathology")

In our reply, we have outlined the several changes made to our manuscript, all closely following the recommendations by the Reviewers.

We are happy to note that the Reviewers were satisfied with our changes to the earlier version and thank them for their valuable input. However, a shared remaining concern by Reviewers 1 and 2 involves the phagocytic response by monocytes to A β aggregates. An additional and novel body of experimental investigation of monocyte A β phagocytosis now addresses this. In this way, we have significantly improved the analysis of these cells' phagocytosis, making the results comparable to similar experiments with iPSC microglia. Other points raised by the reviewers prompted us to make a more careful explanation of the substance of our findings, which we think will help the understanding of our work.

All changes are indicated in the reply below and with yellow highlight in a copy of the manuscript.

In addition to changes based on the reviewers' remarks, we have made minor changes to meet the formatting requirements of Nature Communications. These are briefly indicated at the end of the Point-by-Point reply.

Reviewer #1 (Remarks to the Author):

The revised version of the manuscript by Juul-Madsen and colleagues studying interactions between monocytes and amyloid beta aggregates is much improved comparing to the initial submission. I believe that most reviewers' concerns have been successfully addressed. I still think that there should be more clarity regarding the physiological relevance of the "600nm aggregate size" threshold. I'm still not sure if the large size aggregates are formed by clustering several fibrils is genuinely circulating in blood or they form during sample preparation stages (as it's the case in the aggregate show in in Fig1 h (where several well-defined fibrils are bundled together). Maybe it's not relevant from a clinical diagnosis point of view (are there large clusters or not), but it certainly drives some of the conclusions of the paper in terms of the role of aggregates in monocyte activation and the proposed mechanism shown in Figure 8. The language of the manuscript, together with the scheme shown in Figure 8 suggests that the aggregates that cause that monocyte

activation are fibrils that grow long, while the only SEM image in Fig 1 h shows a bundle of fibrils. Is this just a semantic problem, defining what an aggregate or cluster is?

Author reply RRI.1: We agree with the Reviewer that the size requirements are an important part of our work. However, we want to stress that the question must be considered in at least two ways. First, the limit of "600 nm aggregate size" came from analysis of patient blood. As explained from the data in Fig. 4, the formation of such large aggregates enables a substantial deposition in brain capillaries, thereby forming an adhesive substrate for CR4-expressing cells, including monocytes. Here, the specific size requirement for monocyte depletion and diapedesis is driven by the blood hydrodynamics and, maybe, the propensity of the aggregate to associate with vessel wall glycocalyx, as well as what size of aggregates may tether cells under flow. Second, in our subsequent work on monocyte and iMG phagocytosis, it also seems that aggregate size is important, at least for successful adhesion and phagocytosis. For phagocytosis, aggregates larger than 200-500 nm are more readily internalized than smaller aggregates. This agrees well with our earlier studies on α -synuclein¹. In the Discussion (lines 459-75), we have now emphasized the part of data and structural modeling explaining why such a size requirement originates from the rigid-twisted structure of the A β fibrils, also when they take the form of less ordered clusters of fibrils.

With regard to the morphology of the A β , we agree with the Reviewer that it is easy to overemphasize the fibrillar structure. Indeed, our own data show a more complex structure of the aggregates (Fig. 1h) although the fibrillar structure is a likely subcomponent as judged by the recognition by aducanumab. Some of these issues arise from the availability of high-resolution structures, which only include simple fibrils as now discussed in lines 425-40. Further, we have made this point clearer by referring to the structures as fibrillar aggregates (e.g., lines 277 and 325). The legend to Fig. 8 has also been changed accordingly.

To sum up, my main concern with the manuscript is that it is not clear how aggregate size-dependent interactions with monocytes will have such a dramatic effect in activation. Where is the size-dependent regulation? It has been shown that structural conversion during aggregation happens at

sizes way below those 600nm. So if it's just the aggregate getting bigger (elongation of the fibril) that drives the divergence in toxicity, do receptors act in a different way as a response to the size?

Author reply RRI.2: We have added new data confirming the involvement of CD18 integrins, including CR4, in monocyte phagocytosis of A-A β , but not M-A β . These experiments identify the critical step in the size regulation pertaining to CD18 integrin ligand recognition (Fig. 6). In lines 425-456, we have explained these findings from well-established elements of integrin biology, including the requirement for avidity accompanying the binding of multivalent ligands^{2,3}.

I understand that fishing those clusters with SEM in human samples would be a tedious experiment if they are in ultra-low abundance, so the conclusions of the paper in the current state require some leap of faith. I believe that the science reported in the manuscript deserves publication, but the meaning of that size threshold needs to be clarified, or toned down.

Author reply RRI.2: We thank the Reviewer for the positive evaluation of our work. As described above, we have followed the advice in two ways. First, new, direct evidence of monocytes' phagocytic capacity has been added to support our model (i.e., in Fig. 6). Second, the molecular basis for the size threshold has been more carefully explained in the Discussion aided by available structural information.

Reviewer #2 (Remarks to the Author):

In the previous submission of this manuscript, I was Reviewer #2, and have been given the opportunity to review the latest submission, which is much improved. As before, I will only comment on the monocyte and microglia work.

The authors present good evidence that CR4 mediates monocyte binding to large A β aggregates, but not very strong evidence that CR4 mediates phagocytosis of large A β aggregates (see point 4 below). In the abstract, the authors suggest that CR4 activation by large A β aggregates leads to CD18-rich monocytes being recruited to brain tissue, to explain the absence of CD18-rich monocytes in human blood circulation where large A β aggregates are present. This is highly speculative and was not investigated in this study, but would be of future interest to explore.

Author reply RR2.1: We thank the Reviewer for this digest of our text. Our investigation of the particle behavior in human blood was done by combining in vitro cell adhesion studies and hydrodynamic calculations. We believe this is a necessary first step for highlighting the importance of understanding monocyte diapedesis in MCI. While our methodology cannot provide "proof," it is state-of-the-art as work in patients would require techniques that are hard to come by or, frankly, may not even exist. Following the reviewers' advice, we have modified the abstract and indicated the need for future exploration on line 450. We have collected our comments on monocyte phagocytosis in Author Replies below.

The authors also claim in the abstract that CR4 mediates phagocytosis of large aggregates by microglia. These claims are now well-evidenced, as they showed that in iPSC-microglia preferentially take up large aggregates of A β over monomers, the CD18-activating antibody KIM127 selectively enhances large aggregate (not monomers) phagocytosis, which leads to a corresponding rise in lysosomal activity, and a CR4-blocking antibody attenuates the effect of KIM127 on A-A β phagocytosis. I believe that this is an important finding that deserves publication, and I commend the authors on their experimental design and execution for the iPSC-microglia work, as the data is beautifully consistent and has all of the appropriate controls included. My previous comments have all been satisfactorily addressed, apart from the issue of whether there is sufficient proof to say that monocytes perform CR4-mediated clearance of large A β aggregates (I discuss in points 4 and 6 below).

Author reply RR2.2: We thank the Reviewer for praise of our experimental work. This has also helped resolving the remaining questions on monocyte phagocytosis. We have now employed the protocol for the iMG phagocytosis to the monocyte as further discussed in Author Replies RR2.3, 2.4, and 2.6.

Specific comments:

1. Lines 40-41: In the abstract the authors state: "In microglial cells, CR4 stimulates phagolysosomal activity". This is inaccurate because KIM127 did not by itself induce phagolysosomal activity, it potentiated the effect of aggregated A β . The authors should reword to make this more clear.

Author reply RR2.1: We agree with this reading of our results (Fig. 7d) and thank the Reviewers for the important correction. The relevant text is now changed to use the work "potentiate" (in line 472).

2. Line 41: "CR4 phagocytoses in particular large aggregates", actually it would be more precise to say "Microglia phagocytosis of large aggregates of A β is mediated by CR4" or similar. The phagocytosis is performed by the microglia, rather than by the CR4.

Author reply RR2.2: This is now changed in the revised abstract.

3. Lines 312-313: The subsection title is "CD18 integrins support phagolysosome activity and depletion of large, but not small, A β aggregates in human monocytes". In this section the authors show that CD18 integrin activation has no effect on phagolysosomal activity, therefore the data does not support this claim. A more accurate section title would be something more along the lines of "CD18 integrins support selective depletion of large, but not small, A β aggregates in human monocytes".

Author reply RR2.3: We thank the Reviewer for pointing out this critical deficiency in our work. As also mentioned in the Author reply RR2.1, we have significantly improved the analysis of monocyte A β phagocytosis. This was done by employing the phagocytosis assay used with the microglial cells also for the monocytes. Indeed, Reviewer #2's enthusiasm for our approach greatly supported this choice of methodology. As shown in Fig. 6b, it confirms that the monocytes phagocytose A-A β much better than M-A β only when KIM127 Ab is added. This activity can be blocked by 3.9 Ab, confirming the role of CR4.

4. Lines 320-330 (Figures 6e,f,g,h): It is very interesting that CR4-activation does not increase internalization of A β -coated QDs into monocytes, neither does CR4-activation promote lysosome colocalization with A β -coated QDs. The authors assertion (in the discussion, see point 6) that CR4 mediates amyloid clearance by monocytes seems to rest only on the data in Fig 6h where the size distribution of particles remaining in the supernatant after phagocytosis was analysed, and it was found that the proportion of "large" (> 50nm) particles is decreased by CR4-activation. However the wording of the text is confusing, it states that "Analyzing particles with DH >50 nm only, more were depleted from the supernatant when they were conjugated with A-A β than if they carried M-

A β on the conditions of KIM127 integrin activation (Fig. 6g) implicating CD18 integrins in the process". This implies that the relevant comparison is A-A β versus M-A β , but Fig 6h does not appear to show a significant difference between A-A β and M-A β . What Fig 6h appears to show is a significant difference in uptake of A-A β -coated particles (>50nm only), between KIM127-treated and untreated cells. The authors should edit the text to better communicate this finding.

Author reply RR2.4: As made clear in the text and in the Author Reply RR2.3, our improved assay shows that monocytes and iMGs are very similar in their CR4-mediated phagocytosis of A-A β . It agrees well with our earlier assessment that CD18-integrin function in microglial cells and monocytes is likely comparable as judged by gene transcription of their constituents and cytoplasmic partners¹. The description of the results has now been rewritten to explain better the points raised by the Reviewer (from line 306).

5. Line 330: Typographical error. The Figure referred to here should be Fig. 6h, not Fig. 6g.

Author reply RR2.5: The section has been rewritten, and figure references have been revised.

6. Lines 451-454: "In this model, the lack of AD-like pathology in Agg [+] MCI patients is indirect evidence of enhanced clearance of brain pathology from an increased number of infiltrating phagocytic cells. We have shown that both monocytes and iMGs are capable of amyloid clearance in a CR4-involved process..." The evidence presented in this manuscript does not clearly show that monocytes are capable of CR4-mediated amyloid clearance (as I discussed in point 4), so I disagree with this wording. I read the authors rebuttal. In their rebuttal point R2.5 they claim that their measurements of A β binding to monocytes and depletion of A β from the cell supernatant "forms a body of evidence for an important involvement of CR4 in phagocytosis of aggregated A β by monocytes". However none of these are direct measures of amyloid (phagocytic) clearance, and they are ignoring their own negative phagocytosis assay data (Fig. 6c,d,g).

Author reply RR2.6: The Reviewer addresses a critical weakness in our previous analysis. As discussed in detail above (Author reply RR2.3), we now provide direct evidence (in Fig. 6b) of the monocytic phagocytosis of A β in an assay format corresponding to the analysis of iMGs. As noted in several places, we believe the outcome is similar (with CR4 involvement).

Unlike the monocytes, it is clear from their data that microglia are capable of CR4-mediated phagocytosis of large A β aggregates. Therefore, why is the lack of AD-like pathology in Agg [+] MCI patients evidence that infiltrating phagocytic cells have cleared the brain pathology? There are many other potential explanations, including that microglia are wholly responsible. I suggest that the highlighted sentences are rephrased to avoid completely unsubstantiated speculation.

Author reply RR2.7: We have clarified that monocytes will phagocytose more readily A-A β . However, we agree with the Reviewer that it is uncertain if such phagocytosis in the brain is undertaken by monocytes or rather by a more differentiated, microglial-like state of these cells. This consideration has been added in lines 466-7.

7. The authors mention 'ROS' and 'neurotoxic ROS species' in the abstract, introduction and discussion sections, however none of the experiments in this study address 'ROS' or 'neurotoxicity'. It seems very weird to devote a whole paragraph of the discussion to this. It is not a mainstream view in the microglia field that phagocytosis of A β leads to neurotoxicity via ROS. I looked up the review paper (ref 32) referenced to support the assertion that "Usually, ROS production accompanies lysosomal activity", and there is nothing in the review paper to support this statement. I suggest that the mention of ROS is removed.

Author reply RR2.8: We have followed this good suggestion, and the mentioning of ROS has been removed from the text.

Reviewer #3 (Remarks to the Author):

The authors have discovered 600 nm A-Beta aggregates in the circulation of patients with mild cognitive impairment and found that these asymmetric aggregates can engage CR4 (CD11c/CD18), leading to adhesion of CD18 high monocytes. The authors have responded thoroughly to the reviewer comments, which has improved the manuscript.

Author reply RR3.1: We thank the Reviewer for this positive reception of our changes.

Reviewer #4 (Remarks to the Author):

The authors have addressed all my comments, and now the numerical results they report seem to be correct.

There are two very minor issues that remain:

1) In figure 3a and 3b the authors use right superscripts. The mass number is typically expressed as a left superscript (e.g. ^{18}F).

Author reply RR4.1: We thank the Reviewer for pointing out this error, which is now corrected.

2) The authors state "Magnetic resonance imaging (MRI) and PET data was performed using MINC (Medical Imaging 636 NetCDF) Toolkit (<https://bic-mni.github.io>)." I would replace "performed using" by "analyzed with".

Author reply RR4.2: This is now changed in line 646.

Other minor changes:

As part of the requests by Reviewer #2 to make changes to the Abstract (Author reply RR2.2), it has been trimmed to 150 words according to Nature Comm. guidelines. Likewise, the reference list was shortened to 70 items, partly in consequence of removing the discussion of the role of reactive oxygen species (Author reply RR2.8). In the final paragraph of the Introduction (lines 107-21), verbs have been changed to present tense as per journal standard.

References

1. Juul-Madsen, K. *et al.* Size-Selective Phagocytic Clearance of Fibrillar alpha-Synuclein through Conformational Activation of Complement Receptor 4. *J Immunol* **204**, 1345-1361 (2020).
2. Vorup-Jensen, T. On the roles of polyvalent binding in immune recognition: perspectives in the nanoscience of immunology and the immune response to nanomedicines. *Adv Drug Deliv Rev* **64**, 1759-1781 (2012).
3. Carman, C.V. & Springer, T.A. Integrin avidity regulation: are changes in affinity and conformation underemphasized? *Curr Opin Cell Biol* **15**, 547-556 (2003).

REVIEWERS' COMMENTS

Reviewer #1 (Remarks to the Author):

I'm satisfied by the latest version of the manuscript and answers to questions raised

Reviewer #2 (Remarks to the Author):

I am satisfied that all of my previous points have been addressed appropriately. The authors have provided new data to directly evidence the role of CD18 integrins in monocyte phagocytosis of aggregated amyloid-beta, in Fig6b. The CD18-stimulation seems to have promoted aggregated amyloid-beta uptake in only a minority of the donors (3 out of 8), however the statistics indicate that $p < 0.05$ for the change. On balance, I think that it is acceptable to report an effect when there is additional indirect evidence from other experiments to support a role for CD18 in monocyte phagocytosis of aggregated amyloid-beta. Happy to see this published.

Point-by-point reply to reviewers' comments (NCOMMS-22-53102B; "Large amyloid beta aggregates activate peripheral monocytes and protect against neuropathology")

Below, we have briefly acknowledge the supportive replies from the reviewers.

Reviewer #1 (Remarks to the Author):

I'm satisfied by the latest version of the manuscript and answers to questions raised

Author reply RRR1.1: We thank the Reviewer for the considerable efforts to present the work through the comments made in the reviews of our paper.

Reviewer #2 (Remarks to the Author):

I am satisfied that all of my previous points have been addressed appropriately. The authors have provided new data to directly evidence the role of CD18 integrins in monocyte phagocytosis of aggregated amyloid-beta, in Fig6b. The CD18-stimulation seems to have promoted aggregated amyloid-beta uptake in only a minority of the donors (3 out of 8), however the statistics indicate that $p < 0.05$ for the change. On balance, I think that it is acceptable to report an effect when there is additional indirect evidence from other experiments to support a role for CD18 in monocyte phagocytosis of aggregated amyloid-beta. Happy to see this published.

Author reply RRR2.1: We thank the Reviewer for the considerable efforts to present the work through the comments made in the reviews of our paper. This is include the interest in clarifying the functions of monocytes vis-à-vis aggregated aggregated amyloid-beta.